



**Low methane concentrations in sediment along the continental slope north of**
**Siberia: Inference from pore water geochemistry**
Clint M. Miller[1], Gerald R. Dickens[1], Martin Jakobsson[2], Carina Johansson[2], Andrey
Koshurnikov[3], Matt O'Regan[2], Francesco Muschitiello[2], Christian Stranne[2], Carl-Magnus Mörth[2]
A manuscript submitted to:
*Biogeosciences*
[July 23, 2016]
[1]Department of Earth Science, Rice University, Houston, TX, USA
[2]Department of Geological Sciences, Stockholm University, Stockholm, Sweden
[3]V.I. Il'ichev Pacific Oceanological Institute, RAS





**Abstract:** The Eastern Siberian Margin (ESM), a vast region of the Arctic, potentially holds large amounts of methane in sediments as gas hydrate and free gas. Although this $CH_4$ has become a topic of discussion, primarily because of rapid regional climate change, the ESM remains sparingly explored. Here we present pore water chemistry results from 32 cores taken during Leg 2 of the 2014 SWERUS-C3 expedition. The cores come from depth transects across the continental slope of the ESM between Wrangel Island and the New Siberian Islands. Upward $CH_4$ flux towards the seafloor, as inferred from profiles of dissolved sulfate ($SO_4^{2-}$), alkalinity, and the $\delta^{13}C$-dissolved inorganic Carbon (DIC), is negligible at all stations east of where the Lomonosov Ridge abuts the ESM at about 143°E. In the upper eight meters of these cores, downward sulfate flux never exceeds 9.2 mol/m$^2$-kyr, the upward alkalinity flux never exceeds 6.8 mol/m$^2$-kyr, and $\delta^{13}C$-DIC only slowly decreases with depth (-3.6‰/m on average). Additionally, dissolved $H_2S$ was not detected in these cores, and nutrient and metal profiles reveal that metal oxide reduction by organic carbon dominates the geochemical environment. A single core on Lomonosov Ridge differs, as diffusive fluxes for $SO_4^{2-}$ and alkalinity were 13.9 and 11.3 mol/m$^2$-kyr, respectively, the $\delta^{13}C$-DIC gradient was 5.6‰/m, and $Mn^{2+}$ reduction terminated within 1.3 m of the seafloor. These are among the first pore water results generated from this vast climatically sensitive region, and they imply that significant quantities of $CH_4$, including gas hydrates, do not exist in any of our investigated depth transects spread out along much of the ESM continental slope. This contradicts previous assumptions and hypothetical models and discussion, which generally have assumed the presence of substantial $CH_4$.





## 1. Introduction

The Arctic is especially sensitive to global climate change. Already, over the last century, the region has experienced some of the fastest rates of warming on Earth (Serreze et al., 2000; Peterson et al., 2002; Semiletov et al., 2004). Past and future increases in atmospheric and surface water temperatures should, with time, lead to substantial warming of intermediate to deep waters (Dmitrenko et al., 2008; Spielhagen et al., 2011), as well as sediment beneath the seafloor (Reagan and Moridis, 2009; Phrampus et al., 2014). The latter is both fascinating and worrisome, because pore space within the upper few hundreds of meters of sediment along many continental slopes can contain large amounts of temperature-sensitive methane ($CH_4$) in gas hydrates, free gas, and dissolved gas (Kvenvolden, 1993 and 2001; Beaudoin et al., 2014). Consequently, numerous papers have discussed the potential impact of future warming upon $CH_4$ within slopes of the Arctic Ocean (Paull et al., 1991; Archer, 2007; Reagan and Moridis, 2008; McGuire et al., 2009; Biastoch et al., 2011; Elliott et al., 2011; Ferré et al., 2012; Giustiniani et al., 2013; Thatcher et al., 2013; Stranne et al., 2016).

Globally, the distribution and total amount of $CH_4$ in sediment along continental slopes remains poorly constrained (Beaudoin et al., 2014). This is particularly true for the Arctic Ocean, because ice cover makes accessibility to many regions difficult. Nonetheless, numerous papers have inferred enormous quantities of gas hydrate surrounding the Arctic (Kvenvolden and Grantz, 1990; Max and Lowrie, 1993; Buffett and Archer, 2004; Klauda and Sandler, 2005; Max and Johnson, 2012; Wallmann et al., 2012; Piñero et al., 2013; **Figure 1**). In some sectors, compelling evidence exists for abundant sedimentary $CH_4$ and gas hydrate. Bottom simulating reflectors (BSRs) on seismic profiles generally mark the transition between overlying gas hydrate and underlying free gas (Holbrook et al., 1996; Pecher et al., 2001), and thereby imply





high quantities of CH4 in pore space (Dickens et al., 1997; Pecher et al., 2001). Such BSRs have
been documented along the North Slope of Alaska (Collett, 2002; Collett et al., 2010), within the
Beaufort Sea (Grantz et al., 1976; Grantz et al., 1982; Weaver and Stewart, 1982; Hart et al.,
2011; Phrampus et al., 2014), around Canadian Arctic Islands (Judge, 1982; Hyndman and
Dallimore, 2001; Majorowicz and Osadetz, 2001; Yamamoto and Dallimore, 2008), adjacent to
Svalbard (Eiken and Hinz, 1993; Posewang and Mienert, 1999; Vanneste et al., 2005; Hustoft et
al., 2009; Petersen  et al., 2010), and within the Barents Sea (Andreassen et al. 1990;  Løvø et al.,
1990; Laberg and Andreassen, 1996; Laberg et al., 1998; Chand et al., 2008; Ostanin et al.,
2013). Furthermore, Lorenson and Kvenvolden (1995) observed high CH4 concentrations in
shelf waters of the Beaufort Sea and Shakhova (2010a, 2010b) have documented ample evidence
for methane escape to the water column on the East Siberian Margin (ESM). It generally has
been assumed that sediment on the adjacent ESM slope contains copious CH4 and gas hydrate
(**Figure 1**), although no scientific expedition has investigated the hypothesis.
Regional assessments for the presence of abundant CH4 in marine sediment can be acquired
through two general approaches. The first includes geophysical applications, primarily seismic
reflection profiling and the recognition of BSRs (MacKay et al., 1994; Carcione and Tinivella,
2000; Haacke et al., 2008), which are a common, but not ubiquitous feature, of hydrate bearing
sediments. The second utilizes chemical analyses of pore waters obtained from short sediment
cores (Borowski et el., 1996; Borowski et al., 1999; Kastner et al., 2008b; Dickens and Snyder
2009). In marine sediments with abundant CH4, a general and important process occurs near the
seafloor, typically within the upper 30 m. Upward migrating methane, either through advection
or diffusion, reacts with downward diffusing sulfate ($SO_4^{2-}$):
$^{12}CH_4 + SO_4^{2-} \rightarrow HS^- + H^{12}CO_3^- + H_2O$ (1)



where the superscript $^{12}C$ denotes that methane is depleted in $^{13}C$. This microbially mediated
reaction (Barnes and Goldberg, 1976; Reeburgh, 1976; Devol and Ahmed, 1981; Boetius et al.,
2000), commonly called anaerobic oxidation of methane (AOM), leads to characteristic pore
water chemistry profiles, including a clearly recognizable sulfate methane transition (SMT;
**Figure 2**). The depth of the SMT inversely relates to the flux of $CH_4$ (Dickens, 2001; Bhatnagar,
2011). Where $CH_4$ to the seafloor is high, the SMT is located at shallow depth. Along the
continental shelf and slope of the Beaufort Sea, where seismic profiles indicate gas hydrate,
Coffin et al. (2008, 2013) predictably have documented SMTs in shallow sediment.
The joint Swedish, Russian, U.S. Arctic Ocean Investigation of Climate-Cryosphere-
Carbon interaction (SWERUS-C3) project was initiated to investigate spatial changes in carbon
cycling across the ESM. A central theme of this project was to constrain the amount, distribution,
and fluxes of $CH_4$, and included a two-leg expedition in the boreal summer of 2014 using the
Swedish icebreaker *IB Oden*. Efforts of Leg 2 (8/21-10/3) included retrieval of 60
piston/gravity/multi cores of which six piston, seven gravity, and 17 multicores spanning the
continental slope of the ESM are studied here (**Figure 3**). A total of 446 pore water samples were
collected from these cores to document changes in chemistry associated with expected SMTs.
Here we present and discuss analytical results of these samples. Surprisingly, pore water profiles
strongly indicate that, contrary to general inferences, very little $CH_4$ exists in shallow sediment
along the continental slope north of Siberia, which may preclude the presence of gas hydrate.

**2. Background**
*2.1 East Siberian Margin Geology*



Extensive continental shelves and their associated slopes nearly enclose the central Arctic
Ocean (**Figure 1**). Although it represents only 2.6% of the world's ocean by area (Jakobsson,
2002), the central Arctic Ocean receives approximately 10% of the global freshwater input
(Stein, 2008) as well as corresponding massive discharge of terrigenous material (>249 Mt/yr;
Holmes et al., 2002). Only Fram Strait allows deep-water flow to and from the Arctic Ocean.
This strait, located between Greenland and Svalbard (**Figure 1**), has today a sill depth of about
2540 m (Jakobsson et al., 2003). It opened from early to middle Miocene times (Jakobsson et al.,
2007; Engen et al., 2008; Hustoft et al., 2009). Prior to this, the central Arctic Ocean only
connect to the world oceans through shallow seaways (e.g., Turgay Straight), and deep waters
may have been anoxic for long intervals of the Cretaceous and Paleogene (Clark, 1988; O'Regan
et al., 2011). Sediments with very high total organic carbon (TOC) accumulated on Lomonosov
Ridge during the middle Eocene (Stein et al., 2006), and on Alpha Ridge during the late
Cretaceous (Jenkyns et al., 2004).
The ESM is here defined to comprise the margin of the East Siberian Sea, which stretches
between Wrangel Island to the east and the New Siberian Islands to the west (**Figure 3**). We
include the adjacent continental slope in the ESM. This stretch of continental shelf is the widest
in the world, extending 1500 km north from the coast. The huge swath laying in water depths
less than 100 m (~987 x $10^3$ km$^2$; Jakobsson, 2002) was for the most part, aerially exposed
during glacial periods, resulting in extensive formation of submarine permafrost (Judge, 1982;
Weaver and Stewart, 1982; Løvø et al., 1990; Collett et al., 2010).  The expansive shelf contrasts
with the relative narrow continental slope, which intersects two ridge systems, Mendeleev Ridge
to the east and Lomonosov Ridge to the west (Jakobsson et al., 2008). Bounded by these two



ridge systems, the steep ESM slope leads into the gently sloping Chukchi, Arlis, and Wrangel
perched continental rises (Jakobsson et al., 2003).

*2.2 Regional Oceanography*

Bottom waters impinging the slope of the ESM can generally be divided into three masses:

the Pacific Halocline (~50-200m), the Atlantic Layer (~200-800m), and Canada Basin Bottom
Water (>800m; Timokhov, 1994; Rudels et al., 2000). The Pacific Halocline is a cold (-1.5-0°C),
low salinity (32-33.5 psu) water mass that serves as a boundary (and heat sink) between sea ice
(above) and Atlantic Layer water (below) (Aagaard, 1981; Aagaard and Carmack, 1989). The
underlying Atlantic Layer is warmer (>0°C) but more saline (33.5-34.5 psu; Rudels et al., 2000).
The Atlantic Layer water originates from water arriving to the ESM region partly through Fram
Strait via the West Spitsbergen Current and partly over the Barents Sea through St. Anna
Trough. The inflow from the Atlantic has been observed to vary over time, specifically striking
are observations of warm pulses influencing the core temperature of the Atlantic Layer in the
central Arctic Ocean on decadal time scales (Dmitrenko et al., 2009; Woodgate et al., 2001).
Canada Basin Bottom Water is colder (~-0.5°C) and relatively saline (~34.9 psu), with a
residence time exceeding 300 years (Stein, 2008). The upper halocline shields the lower warmer
waters, which may promote sea ice formation (Aagaard and Carmack, 1989). The aspect
motivating our study is that climate warming could increase bottom water temperatures on the
shelf slope, in the sensitive feather edge of hydrate stability (300-450 m, Stranne et al., 2016),
which would decrease the extent of the gas hydrate stability zone (GHSZ) and possibly release
$CH_4$ to the water column and atmosphere.





*2.3 Current Speculation on Gas Hydrates in the Arctic*
Even during summer months over the last decade, 2-3 m of sea ice covers much of the
Arctic Ocean adjacent to Siberia (Stroeve et al., 2012). This necessitates the use of large ice
breaking vessels to explore the region. Consequently, limited information exists regarding
continental slopes of the ESM. Four icebreaker expeditions, the 1995 Polarstern Expedition
ARK-XI/1 [Rachor, 1995], the 1996 Arctic Ocean Expedition ARK-XII/1 [Augstein et al.,
1997], the 2008 Polarstern Expedition ARK-XXIII/3 [Jokat, 2010], and the 2009 Russian-
American RUSALCA Expedition [Bakhmutov et al., 2009] have retrieved geophysical data and
sediment on or adjacent to the ESM slope.
So far, no drilling has occurred on the ESM slope. However, the 2004 Arctic Coring
Expedition (ACEX; Backman et al., 2009) drilled and cored the central Lomonosov Ridge
(**Figure 1**). There are also land based studies (Gualtier et al., 2005; Sher et al., 2005; Andreev et
al., 2009), and some public Oil and Gas Exploration materials which provide indirect data on the
ESM (Hovland and Svensen, 2006).
Despite the paucity of ground-truth data, as shown by maps of conjectured Arctic gas
hydrate distribution (**Figure 1**), many researchers have predicted widespread and abundant $CH_4$,
including gas hydrate, along the ESM continental slope. This is a logical inference that arose for
two main reasons. First, particulate organic carbon (POC) provides the ultimate source of $CH_4$ in
marine sediments (Kvenvolden and Grantz, 1990), and Arctic slopes may contain high POC
contents, which accumulated prior to the opening of the Fram Strait (Jokat and Ickrath, 2015), or
along with terrigenous material during interglacial intervals of the Quaternary (Danyushevskaya
et al., 1980; Clark, 1988; Darby, 1989; Moran et al., 2006; Archer, 2015). Certainly, organic rich
Eocene sediments have been documented on other Arctic margins and in the ACEX cores on



Lomonosov Ridge (Moran et al., 2006; Backman and Moran, 2009; O'Regan et al., 2011).
Moreover, during Pleistocene glacial periods, extensive portions of the adjacent continental shelf
were subaerially exposed tundra (Gusev et al., 2009; Jakobsson et al., 2014), and the locus of
sediment deposition moved toward the slope (Alekseev, 1997; Naidu et al., 2000; Niessen et al.,
2013). Organic matter burial might be enhanced further by cold seafloor temperatures, which
should reduce bacterial degradation in shallow sediment (Darby et al., 1989; Max and Lowrie,
1993). Second, the thickness of the GHSZ depends on bottom water temperature and the
geothermal gradient (Dickens, 2001), and very low bottom water temperatures along the slope
combined with low geothermal gradients (O'Regan et al., 2016) imply a volumetrically extensive
GHSZ (Miles, 1995; Makogon, 2010). Few environmental considerations point against the
existence of gas hydrates in the ESM slopes although glacial periods dominated by relatively low
sea levels might have kept the sensitive shallow part of the present GHSZ depleted of hydrates
(Stranne et al., 2016).

*2.4 Pore Water Chemistry Above Methane-Charged Sediment Sequences*

Pore water chemistry profiles provide a powerful means to constrain $CH_4$ abundance and

fluxes in marine sediment sequences (Borowski et al., 1996; Berg et al., 1998; Jørgensen et al.,
2001; Torres and Kastner, 2009; Treude et al., 2014). Such profiles are generated by extracting
interstitial water from sediment cores, and measuring the concentrations of dissolved species. In
the absence of significant advection, depth profiles of various analytes relate to Fick's law of
diffusion and chemical reactions (e.g., Berner, 1977; Froelich et al., 1979; Klump and Martens
1981; Schulz, 2000).





The flux ($J$) of a dissolved species through porous marine sediment can be calculated from
the concentration gradient by (Li & Gregory, 1974; Berner, 1975; Lerman, 1977):
$$J = -\varphi D s \frac{\partial C}{\partial Z},\qquad\qquad (2)$$
where $\varphi$ is porosity, $D_s$ is the diffusivity of an ion in sediment at a specified temperature, $C$ is
concentration, and $Z$ is depth. Note that, as generally written, $J$ is positive for upward fluxes and
negative for downward fluxes relative to the seafloor. In many locations, $\varphi$ and $D_s$ change only
moderately (<20%) in the upper few tens of meters below the seafloor. However, abundant $CH_4$
in sediment necessarily leads to a large concentration gradient toward the seafloor and a major
upward flux of $CH_4$. The consequent reaction with $SO_4^{2-}$ via AOM (**Equation 1**) leads to a series
of flux changes in dissolved components (addition or removal), and predictable variations in
corresponding concentration profiles across a SMT (Alperin, 1988; Niewohner et al., 1998;
Ussler and Paull, 2008; Dickens and Snyder, 2009; Regnier et al., 2011).
Typically, the SMT is a thin (<2 m) depth horizon with major inflections in both $CH_4$ and
$SO_4^{2-}$ profiles (**Figure 2**). Sulfate concentrations decrease from seawater values at the seafloor to
zero at the SMT; by contrast, $CH_4$ concentrations rise from zero at the SMT to elevated values at
depth. In regions dominated by diffusion, the depth of the SMT relates to the flux of $CH_4$ from
below (Jørgensen et al., 1990; Dickens, 2001; D'Hondt et al., 2002; Hensen et al., 2003). In part,
this is because $SO_4^{2-}$ concentrations at the seafloor are fixed.
Importantly, as one can infer from **Equations 1 and 2**, AOM affects additional species
dissolved in pore water (Alperin et al., 1988; Jørgensen et al., 1990; Dickens, 2001; Hensen et
al., 2003; Snyder et al., 2007). Dissolved $HS^-$ and $HCO_3^-$ concentrations necessarily increase
across the SMT, so an inflection occurs in their concentration profiles. These two species





contribute to total alkalinity (Gieskes and Rogers, 1973; Haraldsson et al., 1997), which can be
defined as:
$$Alk_T = [HCO_3^-] + 2[CO_3^{2-}] + [HS^-] + [B(OH)_4^-] + [OH^-] + [HPO_4^{2-}] + [NH_3] +$$
$$[X] \, , \tag{3}$$
over the pH range 6.3 to 10.3, where X refers to several minor species. However, in shallow
sediments found above almost all CH$_4$ charged systems, this can be expressed as:
$$Alk_T \approx [HCO_3^-] + [HS^-] \, , \tag{4}$$
Thus, with the production of HS$^-$ and HCO$_3^-$, an inflection in $Alk_T$ occurs across the SMT (Luff
and Wallmann 2003; Dickens and Snyder, 2009; Jørgensen and Parkes, 2010; Smith and Coffin,
2014; Ye et al., 2016).

Marked changes in pore water profiles of other components also typically occur across the

SMT (**Figure 2**). Because CH$_4$ is greatly depleted in [13]C (Paull et al., 2000), the conversion of
CH$_4$ carbon to HCO$_3^-$ carbon (**Equation 1**) induces a decrease in the $\delta^{13}$C values of dissolved
inorganic carbon (DIC) across the SMT (Torres et al., 2007; Holler et al., 2009; Yoshinaga et al.,
2014). However, the magnitude of change becomes complicated because of excess HCO$_3^-$ rising
from below (Snyder et al., 2007; Chatterjee et al., 2011). Dissolved Ba$^{2+}$ concentrations
generally increase significantly just above the SMT. This is because solid barite (BaSO$_4$), a
ubiquitous component of marine sediment on continental slopes (Dehairs et al., 1980; Dymond et
al., 1992; Gingele and Dahmke, 1994), dissolves in the SO$_4^{2-}$-depleted pore water and dissolved
Ba$^{2+}$ then diffuses back across the SMT (Dickens, 2001; Riedinger et al., 2006; Nöthen and
Kasten, 2011). Dissolved Ca$^{2+}$ concentrations usually decrease across the SMT. This is due to
authigenic carbonate precipitation resulting from the production excess HCO$_3^-$ (Greinert et al.,
2001; Luff and Wallmann 2003; Snyder et al., 2007). Importantly, though, dissolved NH$_4^+$



concentrations exhibit no inflection across the SMT. This is because while decomposition of
particulate organic matter generates $NH_4^+$, AOM does not (Borowski et al., 1996). In summary,
pore water analyses at numerous locations demonstrate that characteristic pore water profiles
delineate sites with significant $CH_4$, including gas hydrate, at depth (**Figure 2**).

**3. Materials and Methods**
*3.1 The SWERUS-C3 Expedition, Leg 2*
Between August 21 and October 5, 2014, Leg 2 of the SWERUS-C3 expedition sailed
between Barrow, Alaska and Tromsø, Norway with *IB Oden*. This leg included four transects
that cross the ESM continental slope (**Figure 3**). These transects were along Arlis Spur (TR-1),
north of central East Siberia (TR-2), from close to Henrietta Island to the Makarov Basin (TR-3),
and on the Amerasian side of Lomonosov Ridge (TR-4). Along each transect, scientific
operations involved bathymetric mapping as well as sediment coring a series of stations. One
station also was located on Lomonosov Ridge, near where this long bathymetric high intersects
the ESM. Additionally, three days were spent at Herald Trough, a canyon on the shelf of eastern
Siberia. Data obtained from the northern Lomonosov Ridge and Herald Canyon are not presented
in this manuscript.
An array of coring techniques were used along each transect. In total, 50 sediment cores
were collected at 34 coring stations. These included: multicore sets (22), gravity cores (23),
piston cores (11), and kasten cores (2). The multicorer was an 8-tube corer built by Oktopus
GmbH weighing 500kg. The polycarbonate liners were 60 cm long with a 10 cm diameter. The
piston/gravity coring system was built by Stockholm University with an inner diameter of 10 cm.
Trigger weight cores also were collected during piston coring. The different coring systems



enabled sediment and pore water collection from the seafloor to upwards of eight to nine m
below the seafloor (mbsf).

*3.2 Core material*

Sediment physical properties (piston and gravity cores) were analyzed shipboard using a

Geotek Multi-Sensor Core Logger (MSCL) from Stockholm University. Measurements of the
gamma-ray derived bulk density, compressional wave velocity (p-wave), and magnetic
susceptibility were acquired at a down core resolution of one cm. Discrete samples (2-3 per
section) were collected for sediment index property measurements (bulk density, porosity, water
content and grain density). Grain density was measured using a helium displacement pycnometer
on oven-dried samples. Porosity profiles were generated using the smoothed (3-pt) MSCL-
derived bulk density ($\rho_B$) and the average grain density ($\rho_g$) from each core, where;

$$\varphi = \frac{(\rho_g - \rho_b)}{(\rho_b - \rho_f)},$$    (5)

and a pore fluid density ($\rho_f$) of 1.024 g/cm$^3$ was assumed. In cases where 2 or more distinct
lithologic units existed within a core, the average grain density for each unit was used in this
calculation.
*3.3 Interstitial Water Collection*

Pore waters were collected using Rhizon samplers (Seeberg-Elverfeldt et al., 2005; Dickens

et al., 2007). Cores were cut into ~1.5 m long sections immediately on *Oden*'s deck, brought to
the geochemistry laboratory, and placed on precut racks. Laboratory temperature was a constant
22 °C. Sampling involved drilling holes through the core liner, inserting Rhizons into the
sediment core, and obtaining small volumes of pore water via vacuum and "microfiltration"
(**Figure 4**). An individual Rhizon consists of a hydrophilic membrane composed of a blend of



polyvinylpyrrolidine and polyethersulfone (nominal pore size of 0.12 - 0.18 μm) connected to a
tube. These are pushed into the sediment and, with negative pressure, the filament filters water
into the syringe. The Rhizons were five cm porous flat tip male luer lock (19.21.23) with 12 cm
tubing, purchased from Rhizosphere Research Products (www.rhizosphere.com).

In total, 529 pore water samples were collected in ~10 mL plastic syringes from 32 cores,

which ranged from 0.16 to 8.43 m in length (**Table 1**). Rhizons in gravity and piston cores
typically were spaced every 20 to 30 cm, although occasionally at five cm increments. Of the
total, 456 samples obtained ~10 mL or more of pore water. Rhizon sampling from multicores
took an average of 1.24 hr per sample, and ranged from 0.08 to 4.01 hr; for gravity and piston
cores, the average sampling time was 11.28 hr, and ranged from 1.33 to 23.08 hr. Tabulated
Rhizon flow rates averaged 12.72 mL/hr for multicores and 1.29 mL/hr for piston and gravity
cores (**Table 2**). After considering the time to recover cores from the seafloor, the total time from
core retrieval through sample collection averaged 1.95 hr for multicores and 14.65 hr for piston
and gravity cores.

We highlight the above sampling times due to concerns about the fidelity of chemical

analyses using Rhizon samplers in recent literature (Schrum et al., 2012; Miller et al., 2014).
Since initial implementation of Rhizons in marine sediment cores (Seeberg-Elverfeldt et al.,
2005; Dickens et al., 2007), they increasingly have been used to collect pore waters (e.g.,
Pohlman et al., 2008; Gao et al., 2010; Riedinger et al., 2014). This is for multiple reasons,
including the capability for high-resolution sampling, the ease of sampling, and the minimal
destruction of surrounding sediment (Dickens et al., 2007). However, concerns about using
Rhizon samplers include $CO_2$ degassing during extraction (Schrum et al., 2012) or changes to
pore water composition between core retrieval and water extraction. In the latter case, alteration



of pore water chemistry may occur through reactions induced by elevated temperature, reduced
pressure, evaporation, microbial activity or other processes.

In order to constrain possible changes in pore water chemistry over time, two experiments

were performed onboard *IB Oden*. First, the temperature and pH of a piston core from Station 33
were continuously monitored at five discrete intervals over 24 hours. Probes, inserted into the
sediment by drilling holes in the core liner, recorded data at five minute intervals (**Figure 5**).
Second, for 46 samples (**Table 4**), after collection of the first 10 mL of pore water, the syringe
was removed, and additional pore water was collected in a second (or third) syringe.

While in the shipboard laboratory, Rhizon samples were divided into six aliquots when

sufficient water was available. This sample splitting led to 2465 aliquots of pore water in total,
which then could be examined for different species and at different laboratories. Aliquots 1, 3,
and 6 (below) were collected for all samples.

*3.4 Interstitial Water Analyses*

The first aliquot was used to measure total alkalinity using a Mettler Toledo titrator

onboard *IB Oden*. Immediately after collection, 2 mL of pore water were diluted to 40 mL with
milli-Q water and autotitrated with 0.005M HCl from the original pH to a pH of 5.4. A total of
15 spiked samples and 8 duplicates were analyzed onboard for quality control. Spiked samples
were created by pipetting certified reference material (Batch 135; CRM) into milli-Q water.
Results for spiked samples and duplicates are reported in **Table 3.**

The second aliquot was used to measure the $\delta^{13}C$ composition of DIC ($\delta^{13}C_{DIC}$). Septum

sealed glass vials prepared with 100µL of 85% phosphoric acid and flushed with helium were
prepared before the expedition. The analysis required approximately 40 µg of DIC in each pore





water sample. Onboard alkalinity measurements were used to estimate the correct volume, and
this amount was injected into the vials. Samples were sealed in boxes and refrigerated for the
remainder of the cruise. Four field duplicates, two seawater standards, and a field blank were
collected, stored, and analyzed with the samples. The $\delta^{13}C_{DIC}$ analyses were performed on a
Gasbench II coupled to a MAT 253 mass spectrometer (both Thermo Scientific) at Stockholm
University. The carbon isotope composition of DIC is reported in conventional delta notation
relative to Vienna PeeDee Belemnite (VPDB). Results for field duplicates and standards are
reported in **Table 2**. Standard deviation for the analyses of $\delta^{13}C_{DIC}$ was less than 0.1 per mille.
The results for seawater standards collected onboard are given in **Table 3**.

The third aliquot was used to measure dissolved sulfur and metals. Approximately 3 mL of

pore water were placed into acid washed cryovials. Samples were acid preserved with 10 μL
ultrapure $HNO^3$. Additionally, 11 blind field duplicates and 2 field blanks were collected and
processed in the same manner. Concentrations of Ba, Ca, Fe, Mg, Mn, S, and Sr were determined
on an Agilent Vista Pro Inductively Coupled Atomic Emission Spectrometer (ICP-AES) housed
in the geochemistry facilities at Rice University. Known standard solutions and pore fluid
samples were diluted 1:20 with 18-MΩ water. Scandium was added to both standards and
samples to correct for instrumental drift (emission line 361.383 nm). Wavelengths used for
elemental analysis followed those indicated by Murray et al. (2000). Following initial analysis,
an additional dilution, 1:80 with 18-MΩ water, was analyzed for Ca, Mg, and S. After every 10
analyses, an International Association of Physical Sciences (IAPSO) standard seawater spiked
sample and a blank were examined for quality control. Relative standard deviations (RSD) from
stock solutions are reported in **Table 3**.



The fourth aliquot was used to measure dissolved ammonia ($NH_4^+$). This was carried out
shipboard via a colorimetric method similar to that presented by Gieskes et al. (1991). Set
volumes (100µL) of pore water were pipetted into 1 $cm^3$ plastic cuvettes and diluted with 900 µL
of milli-Q water. Two reagents (100 µL of A and 100 µL of B) were then pipetted into the
cuvettes. Reagent A was prepared by adding 35 g of trisodium citrate ($Na_3C_6H_5O_7$), 2.7 g of
phenol ($C_6H_5OH$), and 0.06 g of sodium nitroprusside ($Na_2[Fe(CN)_5NO]$) to 100 mL of milli-Q
water. Reagent B was prepared by dissolving 1.36 g of sodium hydroxide in 100 mL of milli-Q
water and adding 3 mL sodium hypochlorite (NaClO) solution. After the reagents were added,
solutions were mixed, and allowed to react for at least six but not more than 24 hours. Solutions
turned various shades of blue, which to relate to $NH_4^+$ concentration, and which were measured
by absorbance at 630 nm on a Hitachi U-1100 spectrophotometer. Five point calibration curves
(0 to 200 µM) were measured before each sample set and corrected using VKI standard (QC
RW1; www.eurofins.dk; **Table 3**).
The fifth aliquot was used to measure dissolved phosphate ($PO_4^{3-}$). The method of
preparation also followed that given by Gieskes et al. (1991). The remainder of the pore water
(generally between 1 and 3mL) was added to milli-Q water to a sum of 10 mL. Two reagents
were then added to the solution to react with phosphate (200 µL of A and B). Reagent A was
prepared by first making three solutions: eight grams of ammonium molybdate (($NH_4$)$_2MoO_4$)
were added to 80 mL of milli-Q water, 50 mL of concentrated sulfuric acid were added to 150
mL of milli-Q water, and 0.01 g of potassium antimonyl tartrate hydrate ($C_8H_4K_2O_{12}Sb_2 \cdot$
$XH_2O$) were added to 10 mL of milli-Q water. Then, 30 mL of the ammonium molybdate
solution were added to 90 mL of the sulfuric acid solution, and five mL potassium antimonyl
tartrate solution was slowly added dropwise. Reagent B was created by dissolving 10 g of

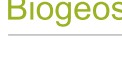
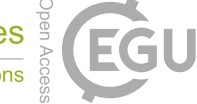


ascorbic acid in 50 mL of milli-Q water. After the samples were prepared, reagent A and B were
added, mixed, and allowed to react for 10 but not more than 30 minutes. Solutions turned various
shades of blue, which to relate to $PO_4^{3-}$ concentration, and which were then measured at an
absorbance of 880 nm on the above spectrophotometer. Five point calibration curves (0 to 50
µM) were measured before each sample set and corrected using VKI standard (QC RW1;
www.eurofins.dk; **Table 3**).

In cases of excess sample, an additional aliquot was collected to test for dissolved hydrogen

sulfide. Approximately 2 mL of pore water was placed into a cryovial, and 200 µL of a 2.5% Zn-
acetate $(Zn(C_2H_3O_2)_2)$ solution was added. Given the extremely low solubility of ZnS, a white
precipitate should form in the presence of even very low $H_2S$ concentrations (Cline, 1969;
Goldhaber, 1974).

**4. Results**
*4.1 Generalities*

With the large number of pore water measurements (**Table 1**) we begin with some

generalities regarding the results. We plot pore water concentration profiles along each transect
collectively (**Figures 6 - 10**), irrespective of coring device or water depth, although clear
variance in pore water chemistry exists between stations for some dissolved species (e.g., Fe).

Most species display "smooth" concentration profiles with respect to sediment depth

(**Figures 6 - 10**). That is, concentrations of successive samples do not display a high degree of
scatter. This is expected for pore water profiles in sediment where diffusion dominates (Froelich
et al., 1979; Klump and Martens 1981; Schulz, 2000). However, as best seen for dissolved
species whose concentrations do not appreciably change over depth (e.g., $Ba^{2+}$ and $Ca^{2+}$) scatter



exists beyond that predicted from analytical precision. This scatter has a weak positive
correlation with increased sampling time, which can be shown by comparing time to a deviation
in concentration (**Figure 11**). The latter is defined by:
$$\Delta X = X_{Measured} - X_{Predicted} ,\qquad(6)$$
where $X$ is the species of interest, and $X_{Predicted}$ is the concentration of $X$ determined from the
linear best fit line of a concentration profile.

A method detection limit (MDL) for each species can be determined by the following

equation:
$$MDL = \left(\frac{C_{High}-C_{Low}}{I_{High}-I_{Low}}\right)3\sigma ,\qquad(7)$$
where C = concentration and I = intensity (counts per second on the ICP-AES). The MDLs were
as follows: Ba = 0.01 µM, Ca = 0.08 µM, Fe = 5.9 µM, Mg = 0.22 µM, Mn = 0.24 µM, S = 1.2
µM, Sr = 0.01 µM. On all plots, for reference, we place dashed lines for values of IAPSO
seawater standard (Alkalinity = 2.33 mM, Ba = 0.00 mM, Ca = 10.28 mM, Fe = 0.00 mM, Mg =
53.06 mM, Mn = 0.00 mM, S = 28.19 mM, Sr = 0.09 mM, $NH_4$ = 0.00 mM, $HPO_4$ = 0.00 mM).
Pore water profiles generated from ACEX cores (Backman et al., 2009) also are shown for
comparison.

*4.2 Porosity and Sampling Time*

Measured porosity values of piston and gravity cores generally decrease with depth from

80% or greater at the mudline to around 60% at eight mbsf (**Figure 12a**). Over the first 0.1 m,
porosity decreases steeply, by an average of 6.8%. From 0.2 to 8.0 m, porosity decreases much
more gradually, by an average of 1.3% every meter. The 1σ deviation in porosity between all
stations typically ranges between 6 and 10% at any given depth.





Sampling time inversely relates to porosity (**Table 2**). Multicore rhizon extraction rates
(**Table 2**) averaged 12.72 mL/hr while gravity and piston cores averaged 1.29 mL/hr. This flow
rate generally decreased with depth. Across all data from all cores, a first-order relationship
between depth (z) and extraction rate (ER) can be expressed as ER = $4.4911z^{-1.512}$ ($R^2$ = 0.789;
**Figure 12b**). The extraction rate correlated with depth more closely than with porosity (**Figure**
**12c**). The porosity ($\phi$)-extraction rate relationship, expressed as ER = $21.718(\phi)^{8.161}$ had an $R^2$ =

0.631.


*4.3 Physiochemical Conditions During Rhizon Sampling*
For the five sections from Station 33 examined for changes in physiochemical conditions,
temperature rose from ~2°C upon initial measurement to between 16.9 and 18.4 °C within 24
hours (**Figure 5**). In general, the shallow sections increased faster than the deeper sections.
Initial pH decreased with depth (0.05 mbsf = 7.79 units, 1.86 mbsf = 7.71 units, 4.80 mbsf = 7.39
units, and 6.30 mbsf = 7.19 units). Over the same time interval, pH decreased significantly in all
core sections, by an average of 0.25 units, with a range between 0.18 and 0.38 units (**Figure 5**).
Note, however, that pH dropped by 0.3 units at ~20 hrs in one of the pH profiles (Section 2, 1.86
mbsf). This may be due to a temporary crack in the sediment core created by removing pore
water through rhizon sampling, although no crack was observed when the core section was split.
In total, 46 of the 68 Rhizon sampling depths at Station 28 enabled collection of multiple
water samples (**Table 4**). This included "second generation" samples, where beyond the first ~10
mL, another 1 to 10 mL were obtained, as well as three "third generation" samples, where
beyond the first ~20 mL, another 1 to 10 mL were obtained. The sample depths which did not
yield enough pore water for a "second generation" tended to be deeper (16 of 22 were in the





deepest section). Relative to the initial 10mL of pore water, alkalinity increased in 43 of the
second generation samples, and in all three of the third generation samples by an average 0.15
mM (4.1% increase). Interestingly, no statistically significant changes in concentrations of
phosphate, ammonia or any dissolved metal were observed.

*4.4 Alkalinity and $\delta^{13}C$*

Alkalinity concentrations increase with depth in all cores (**Figures 6 - 10**). Moreover, in

most cases, the rise is nearly linear. Across all stations on the four transects, alkalinity increases
by an average of 0.51 mM/m, although variance exists between mean gradients for each transect
(Tr1 = 0.46 mM/m, Tr2 = 0.34 mM/m, Tr3 = 0.91 mM/m, and Tr4 = 0.44 mM/m) and by station
along each transect. Overall, the rise in alkalinity at these 15 stations ranges from 0.30 to 0.98
mM/m. The Lomonosov Ridge station differs (**Figure 10**), as alkalinity increases much faster
with depth (1.86 mM/m).

Concave-down $\delta^{13}C$-DIC profiles characterize pore waters at all stations (**Figures 6 - 10**).

The decrease in $\delta^{13}C$-DIC changes most rapidly near the seafloor. Across all stations along the
four transects, pore water $\delta^{13}C$-DIC values decrease from near zero close to the mudline at an
average of -3.6 ‰/m. Again, significant variance in mean gradients occurs according to transect
(Tr1 = -3.3 ‰/m, Tr2 = -3.0 ‰/m, and Tr3 = -4.7 ‰/m) and according to station on each
transect. The range in average $\delta^{13}C$-DIC value gradients across all stations is -2.7 to -4.9 ‰/m.
As with alkalinity, the $\delta^{13}C$-DIC profile at the Lomonosov Ridge station differs, with values
decreasing by 5.6 ‰/m, such that 8 mbsf, $\delta^{13}C$-DIC approaches -45 ‰. In summary, a basic
relationship exists between higher alkalinity and lower $\delta^{13}C$-DIC across all stations.



*4.5 Sulfur and sulfate*
No sulfide was detected by smell or with addition of Zn-acetate in any pore water sample.
Molar concentrations of total dissolved sulfur should, therefore, represent those of dissolved
$SO_4^{2-}$. Along the four transects, dissolved S concentrations decrease with depth at all stations
(**Figures 6 – 9**). The sulfur concentration in the shallowest sample varied from 27.29 to 30.58
mM and averaged 28.70 mM. From these "seafloor" values, concentrations decrease by an
average 0.69 mM/m, again with variance according to transect (Tr1 = -0.58 mM/m, Tr2 = -0.57
mM/m, Tr3 = -1.09 mM/m; and Tr4 = -0.60 mM/m) and station along each transect. The S
gradients across all stations along the ESM slope range from -0.41 to -1.13 mM/m. Total
dissolved S at the Lomonosov Ridge station decreased faster than at any of the other stations (-
1.92 mM/m). Importantly, decreases in dissolved S are similar in magnitude to increases in
alkalinity at each station. Indeed, the molar ratio of alkalinity increase to sulfate decrease (-
ΔAlkalinity/ΔS) is 0.98 (**Figure 13a**).

*4.6 "Nutrients": Phosphate and Ammonia*
Often, in discussions of pore water chemistry, dissolved phosphate ($HPO_4^{2-}$) and ammonia
($NH_4^+$) are classified as "nutrients", although the connotation derives from the fact that these two
species arise through the oxidation of POM in the sediment (Berner, 1977). The C:N:P molar
ratio, known as the "Redfield Ratio", of initial POM is approximately 106:16:1 (Redfield, 1958;
Takahashi, 1985). Therefore, assuming mass balance, dissolved "nutrients" are used as reference
for the amount of POC consumed through microbial oxidation. Importantly, concentrations of
$HPO_4^{2-}$ and $NH_4^+$ are near or below detection in samples immediately below the seafloor
(**Figures 6 -10**).





With depth, concentrations of dissolved $HPO_4^{2-}$ typically increase, reach a subsurface
maximum, and then decrease (**Figure 6 – 10**). With available data, a more pronounced maximum
generally occurs at stations with relatively shallow water depth. For example, and within the
spatial resolution of samples, consider the peak in $HPO_4^{2-}$ concentrations at four stations. At the
two shallow stations, S12 (384 m) and S22 (367 m) the $HPO_4^{2-}$ maxima are, 73 µM (1.91 m) and
18 µM (0.66 m), respectively at the two deeper stations, S17 (977 m) and S14 (733 m), the
$HPO_4^{2-}$ maxima are only 6.7 µM (1.76 m) and 7.1 µM (2.33 m) respectively. The station on
Lomonosov Ridge (S31) has a high in $HPO_4^{2-}$ concentration of 76 µM at 1.02 m below the
mudline. In general, stations with more pronounced $HPO_4^{2-}$ maxima also have greater increases
in alkalinity with depth.
By contrast, dissolved $NH_4^+$ profiles rise almost linearly with depth, but with slight
concave-down curvature. Similar to dissolved $HPO_4^{2-}$ profiles, $NH_4^+$ concentrations increase
with depth fastest at stations with shallower water depth (although we note an exception for Tr2).
Across stations along the four transects, pore water $NH_4^+$ concentrations increase with depth on
average by 38.69 µM/m, with a range from 11.28 to 76.08 µM/m. Along each transect, the
average $NH_4^+$ gradients are as follows: Tr1 = 43.02 µM/m, Tr2 = 17.38 µM/m, Tr3 = 68.97
µM/m, and Tr4 = 29.04 µM/m.
The $HPO_4^{2-}$, $NH_4^+$, and alkalinity profiles relate to one another statistically, although with
distinction. The concentration relationship of alkalinity and ammonium ion can be expressed by
a second order polynomial ($[NH_4^+] = -0.003[Alk]^2 + 0.105 [Alk] – 0.253$; **Figure 13b**) with an
average molar ratio ($\Delta Alk/\Delta NH_4^+$) of 14.69. All stations have a C:N ratio in pore waters more
than the Redfield Ratio of 6.625 (**Figure 14**). The molar ratio of alkalinity and phosphate ion
($\Delta Alk/\Delta HPO_4^{2-}$) averages 55.72 for all stations. This means that all stations have an average C:P





ratio less than 106. Overall, a consistent pattern emerges between changes in $NH_4^+$, and
alkalinity, but one that deviates significantly from Redfield ratio. Interestingly, the C:N ratio
appears to vary significantly across transects. This ratio increases from Tr1 (8.61-11.22), Tr3
(12.5-18.14), Tr2 (17.53-18.55), to the Lomonosov Ridge station (22.62). The C:P ratio followed
a similar pattern, generally increasing from east to west: Tr1 (16.57-74.70), Tr2 (26.32-92.04),
Tr3 (26.29-86.34), and Tr4 (52.18-124.35).

*4.7 Metals*

At most stations, dissolved Ba concentrations increase nonlinearly from values at or below

detection limit (0.01 μM) near the seafloor to generally constant values (0.6 – 0.7 μM) within 0.8
m below the seafloor. However, at several stations, dissolved Ba concentrations remained at or
below the detection limit for all samples.

Overall, dissolved Ca, Mg, and Sr concentrations decrease slightly with depth (**Figures 6 -**

**10**). Across stations along the four transects, Ca concentrations drop on average between -0.094
and -0.122 mM/m (Tr1), between -0.092  and -0.093 mM/m (Tr2), between -0.092  and -0.101
(Tr3), and -0.075 mM/m (Tr4). Magnesium concentrations also drop, the average change being
between -0.430 and -0.481 mM/m (Tr1), between -0.274 and -1.319 (Tr2), between -0.863 and -
0.942 mM/m (Tr3), and -0.467 mM/m (Tr4). Strontium concentrations decrease by an average
amount of 0.3 μM/m, considering all stations along the four transect stations (Tr1 = 0.5 μM/m,
Tr2 = 0.3 μM/m, Tr3 = 0.1 μM/m, and Tr4 = 0.1 μM/m). The station on Lomonosov Ridge again
stands apart. At this location, the decreases in dissolved Ca, Mg, and Sr are 0.27 mM/m, 1.24
mM/m, and 0.50 μM/m, respectively.





The profiles of dissolved Mn and Fe are spatially complicated. Generally, profiles show a
broad rise in concentration and subsequent fall at deeper depth. Some stations have a maxima in
dissolved Mn (Stations S12 (135 µM at 5 m), S28 (66 µM at 3.1 m), and Lomonosov Ridge (86
µM at 1.3 m), where concentrations decrease below. At other stations, Mn concentrations are still
increasing at the lowest depth. Iron concentrations are generally below the detection limit at or
near the mudline, and begin increasing around 2.5 – 3.5 m reaching concentrations upward of 20
µM.

**5. Discussion**
*5.1 Flow Rates from Rhizons*
Pore water flow rate drops quasi-exponentially with depth (**Figure 12b**), similar to what
was documented on ACEX (Dickens, 2007). This probably results from the decrease in porosity
(and presumably permeability) with depth (**Figure 12c**). Given that individual Rhizons have
similar vacuum to pull the water, a decrease in porosity and permeability means a slower flow
(Domenico and Schwartz, 1998).

*5.2 Fidelity of Rhizon Pore Water Measurements*
Researchers have employed multiple methods to extract pore waters from marine
sediments over the last few decades (Seeberg-Elverfeldt et al., 2005). As the rhizon technique
remains relatively novel, the accuracy and precision of analyses obtained through this approach
warrant consideration before discussing the results. This issue arises particularly because of the
two aforementioned papers questioning the fidelity of pore water records generated through
rhizon sampling.





Schrum et al. (2012) compared dissolved species collected by whole round squeezing and
rhizons. They observed very subtle but consistent (0.06 to 0.8 mM) offsets to lower alkalinity in
Rhizon samples, and hypothesized that this reflected $CO_2$ degassing during extraction. For
example, the release of gas during filtering under vacuum conditions might increase, leading to
precipitation of $CaCO_3$, and ultimately a drop in alkalinity. They noted, though, that rhizons
seemed to provide accurate measurements for nutrients and metals.
Miller at al. (2014) compared chloride concentrations, oxygen isotopes, and hydrogen
isotopes in pore waters collected from whole round squeezing and rhizons. The rhizon samples
appeared to have higher $[Cl^-]$ and greater enrichments in heavier isotopes ($^{18}O$ and D). The
authors suggested some combination of water absorption onto the hydrophilic membrane, ion
exclusion and isotope fractionation due to clay ultrafiltration, and water evaporation during
degassing as possible sources for these offsets.
Rather than an issue with Rhizon sampling per se, an alternative explanation for analytical
discrepancies lies with collection time. A lengthy time between core retrieval and final pore
water collection could allow for changes in physiochemical conditions, which might relate to
evaporation and carbonate precipitation. Our experiments show that significant differences in the
chemical environment of cores occur during rhizon sampling. Consider the temperature (**Figure
5a**) and pH (**Figure 5b**) evolution over 24 hours for the five core sections from station S33 that
were analyzed. Note that the time to recover, to cut, and to transport these sections from the ship
deck to the geochemistry laboratory (total 1.71 hrs) was similar to that involved for other
samples (**Table 1**). Thus, we consider results from these cores representative.
Many authors have observed variations in pore water pH, DIC, alkalinity, and $Ca^{2+}$ values
over time (e.g., Gieskes, 1974; Paull et al., 1996; Wang et al., 2010; Sauvage, 2013). The



changes in sections from S33 clearly indicate that physiochemical conditions within the core
change significantly within 24 hours. The ~15 °C increase will alter inorganic solid-liquid
equilibrium conditions (de Lange et al., 1992), and should increase microbial respiration (Sander
and Kalff, 1993). The nominal ~0.25 drop in pH implies a reduction in alkalinity. Interestingly,
though, this appears opposite of results from sequential sampling, where each progressive
"generation" of pore water had greater alkalinity.

One issue is location. The pH sondes were always more than 10 cm from the nearest

rhizon. Although it is possible that the Rhizon's negative pressure in the sediment is
compensated by $O_2$/air increasing respiration, previous experiments on rhizon flow (Seeberg-
Elverfeldt et al., 2005; Dickens et al., 2007) indicate that rhizons generally pull water from <3
cm along the core. Thus, water masses adjacent to pH meters were likely "out-of
communication" with those being sampled by the rhizons. We suggest that at least two factors
effect chemistry: (1) temperature and pH (and pressure) of pore waters change with time after
core retrieval; and, (2) pore water chemistry evolves during water removal.

The observed evolution of pore water chemistry may be related to increasing temperature

and possible introduction of atmospheric air via the Rhizon drill hole each time the syringe was
removed. As temperature increases, greater microbial activity may drive pH down by increasing
$CO_2$ concentration. Additionally, removing the syringe may have provided opportunity for
atmospheric air to enter the sediment through the filament. As the pH decreased, carbonate
dissolved, increasing $HCO_3^-$ concentration in the pore water. The Rhizons continually applied
additional negative pressure. However, as stated previously, the pH sondes were sufficiently far
from the Rhizons to be affected by pore water extraction.





As clearly documented here and in other works (Seeberg-Elverfeldt et al., 2005; Dickens et
al., 2007; Pohlman et al., 2008), rhizon sampling can lead to "smooth" concentration profiles for
multiple dissolved species, including alkalinity (**Figures 6 – 10**).  The concerns raised about
rhizon sampling may be valid for dissolved components when concentration gradients are low.
For example, Schrum et al. stressed alkalinity differences of 0.06 to 0.8 mM, but the total
alkalinity range in this study was 1.80 and 14.58 mM. A similar finding occurs in the dissolved
$Ca^{2+}$ and $Ba^{2+}$ profiles of this study, where adjacent samples deviate by amount greater than
analytical precision (**Table 3, Figure 11**). However, when the signal to noise ratio become high,
as true with most dissolved components at most stations (**Figures 6 – 10**), the rhizon sampling
renders pore water profiles with well defined concentration gradients that can be interpreted in
terms of chemical reactions and fluxes.

*5.3 Reading the Pore Water Profiles*
Pore water profiles in most marine sediment express solute fluxes resulting from
chemical reactions, sediment properties, and diffusion (Berner, 1980; Berg et al., 1998). Within
10 m of the seafloor, where temperature and the diffusion coefficient change minimally, depth
intervals having inflections in the concentration gradient (dC/dz) generally represent zones
where production or consumption of dissolved components occur ($\Delta$J), or where porosity ($\phi$)
changes significantly (**Equation 2**). Importantly, excepting areas of the seafloor with strong fluid
flow (e.g., mud volcanoes, cold seeps), methane charged sediments along continental margins
have very predictable pore water profiles.
As previously emphasized, numerous studies demonstrate that a prominent SMT
characterizes shallow sediment in locations with high methane concentrations in underlying





strata. Moreover, inflections in pore water $SO_4^{2-}$, alkalinity, $\delta^{13}$C-DIC values, and hydrogen
sulfide consistently occur across this geochemical horizon (**Figure 2**). This is because AOM
consumes $SO_4^{2-}$ and produces $^{13}$C-depleted $HCO_3^-$ and $HS^-$ (**Equation 1**). The overall
geochemistry is best understood by considering fluxes (Borowski et al., 1996; Berg et al., 1998;
Chatterjee et al., 2011). Across the SMT, upward migrating methane of some flux ($J$CH$_4$) reacts
with downward diffusing $SO_4^{2-}$ of equal flux but opposite sign (-$J$SO$_4^{2-}$). This leads to a sharp
concave-down inflection in $SO_4^{2-}$ concentrations (i.e. the SMT), with the depth driven by $J$CH$_4$.
Fluxes of $HCO_3^-$ ($J$HCO$_3^-$) and $HS^-$ ($J$HS$^-$) of similar magnitude enter pore water, but are
expressed differently in pore water profiles. In general, the input of $^{13}$C-depleted $HCO_3^-$
contributes to already $^{13}$C-enriched $HCO_3^-$ concentrations, produced during methanogenesis
deeper in the sediment column. The consequence is a steep rise in $HCO_3^-$ concentrations with
depth, but having a positive kink across the SMT, where a coincident drop in the $\delta^{13}$C-DIC
values occur. The input of $HS^-$ diffuses upward and downward, where it reacts with dissolved Fe
or sedimentary phases. The consequence is a "bell shaped" $HS^-$ pore water profile with the
maxima at the SMT.

Good examples of where such pore water chemistry is documented include: Baltic Sea

(Jørgensen et al, 1990), Black Sea (Jørgensen et al, 2004), Blake Ridge (Paull et al., 2000;
Borowski et al., 2001), Cariaco Trench (Reeburgh, 1976), Cascadia Margin (Torres and Kastner,
2009), Gulf of Mexico (Kastner et al., 2008a; Hu et al., 2010; Smith and Coffin, 2014), Hydrate
Ridge (Claypool et al., 2006), offshore Namibia (Niewohner et al., 1998), offshore Peru
(Donohue et al., 2006), South China Sea (Luo et al., 2013; Hu et al., 2015), and Sea of Japan
(Expedition Scientists, 2014). In any case, through use of **Equation 2**, fluxes of dissolved ions,
and by inference dissolved CH$_4$, can be calculated from measured pore water concentration




profiles with knowledge of porosity and sedimentary diffusion constants (e.g., Niewohner et al.,
1998). At sites with abundant methane in the upper few hundred meters, notably including sites
with gas hydrate, estimated values for $J\text{CH}_4$ and $-J\text{SO}_4^{2-}$ are universally high (**Table 4**).This
includes sites in the Beaufort Sea, 154.8 mol/m$^2$-kyr (Coffin et al., 2013), 102 mol/m$^2$-kyr
(Umitaka Spur; Snyder et al., 2007), 86.2 mol/m$^2$-kyr (Hikurangi Margin; Coffin et al., 2007),
362.0 mol/m$^2$-kyr (Chilean Margin; Coffin et al., 2006), 162.5 mol/m$^2$-kyr (Argentine Basin;
Hensen et al., 2003; **Figure 15**). Methane above gas hydrates can migrate upward even faster
through advective bubble ebullition at cold seeps (Joye et al., 2004).

*5.4 General Absence of Methane*
Direct measurements of dissolved $\text{CH}_4$ in deep-sea sediment are complicated (Claypool
and Kvenvolden 1983). During core retrieval and depressurization, gas ebullition occurs, which
leads to significant $\text{CH}_4$ loss from pore space. Interestingly, however, in sediments containing
high $\text{CH}_4$ concentrations and recovered through piston coring, gas release typically generates
obvious sub-horizontal cracks that span the core between the liner. No such cracks were
documented in any of the cores.
Excluding Station St31 on the southern Lomonosov Ridge (discussed below), there is no
indication of a shallow SMT. Interstitial water sulfur concentrations do not drop below 22.78
mM within the upper 8 m. In fact, calculated downward $\text{SO}_4^{2-}$ fluxes, as inferred from sulfur
concentration gradients (**Table 4**) range from -1.8 to -9.2 mol/m$^2$-kyr for all stations except
Station S31. For comparison, with a temperature of 2 °C (**Figure 5a**) and measured porosities
(**Figure 12a**), even an SMT at six mbsf would imply $\text{SO}_4^{2-}$ flux of -40 mol/m$^2$-kyr.



Given the lack of HS⁻ and the measured pH at Station S33 (**Figure 5**), alkalinity should
closely approximate $HCO_3^-$ concentrations (**Equation 4**). Estimated $HCO_3^-$ fluxes do not exceed
6.8 mol/m2-kyr at any station east of the Lomonosov Ridge (**Table 4**). For comparison, when
alkalinity gradients are used to estimate $JHCO_3^-$ at sites with abundant $CH_4$ at depth, values
generally exceed 30 mol/m$^2$-kyr above the SMT (**Table 4**). These extreme fluxes arise because
methanogenesis in deeper sediment drives an upward flux of $HCO_3^-$ (**Figure 2**), and because
AOM also contributes HS⁻ to pore water at the SMT (**Equation 1**).
The $\delta^{13}$C-DIC values of pore water decrease with depth at all stations, almost in concert
with the rise in alkalinity. However, other than Station S31, the lowest value of $\delta^{13}$C-DIC is -
25.23 ‰ at 5.5 m at Station S22 (**Figure 8**). This is interesting because a series of microbial
reactions utilizing particulate organic matter (POM) can lead to higher alkalinity and lower $\delta^{13}$C-
DIC values in pore water (Chatterjee et al., 2001). The most important of these reactions is
organoclastic sulfate reduction, which can be expressed as (Berner, 1980; Boudreau and
Westrich, 1984):
$$2^{12}CH_2O + SO_4^{2-} \rightarrow H_2S + 2H^{12}CO_3^-, \tag{8}$$
where again the $^{12}$C superscript indicates depletion in $^{13}$C. Notably, this reaction has a 2:1
relationship between C and S fluxes, rather than the 1:1 ratio of AOM (**Equation 1**).
As emphasized previously, methane-charged sediment sequences do occur on continental
slopes in the Arctic. Of particular interest to this study are locations in the Beaufort Sea, where
indications for gas hydrate manifest on seismic profiles (Grantz et al., 1976; Grantz et al., 1982;
Weaver and Stewart, 1982; Hart et al., 2011; Phrampus et al., 2014), and pore water profiles
have been generated using shallow piston cores (Coffin et al., 2013). Striking contrasts exist
between pore water profiles of the Beaufort Sea and those of the ESM (**Table 4; Figure 15**). In



the Beaufort Sea, there are moderate to high downward sulfate and upward methane fluxes (1.9
to 154.8 mol/m²-kyr), shallow SMTs (6.29 to 1.06 mbsf), high DIC fluxes between the SMT and
the mudline (46.3 to 242.6), and negative $\delta^{13}$C-DIC values at SMT's (≈ -20‰).

*5.5 Special Case "Lomonosov Ridge Station"*

Station 31 on the Lomonosov Ridge (**Figure 10**) differs from all other stations examined

in this study. Here, pore water chemistry profiles hint at $CH_4$ in pore space within shallow
sediment. Extrapolation of the dissolved sulfur profile suggests an SMT at approximately 13.9
m. Such a depth lies within the range common for locations with AOM (D'Hondt et al., 2002),
notably including well studied sites on Blake Ridge (Borowski et al., 1999). Similar to some sites
with $CH_4$, the $\delta^{13}$C-DIC values become very "light"; indeed, the value at the base of the core, -
43.54, almost necessarily implies $CH_4$ oxidation and a shallow SMT. Comparably steep
alkalinity (1.6 mM/m) and $NH_4$ gradients (60.4 μM/m) also characterize other sites with $CH_4$
near the seafloor. However, an issue concerns reduced sulfur produced via AOM (**Equation 1**).
One might expect evidence of $HS^-$ migrating from below (**Figure 2**), but none was detected.

A comparison of published DIC fluxes, $SO_4^{2-}$ fluxes, and SMT depths (**Table 4**) reveals

fluxes decrease exponentially with SMT depth (**Figure 15**). A fundamental relationship exists
when one considers that $CH_4$ flux controls SMT depth (**Equation 1**; **Figure 2**). The modest
$SO_4^{2-}$ flux (-13.9 mol/m²-kyr) and alkalinity flux (11.3 mol/m²-kyr) of the Lomonosov Ridge
station fits quite well with literature values of similar SMT depth. For example, Hensen et al.
(2003) calculated a -14.69 mol/m²-kyr $SO_4^{2-}$ flux for a site with an SMT at 14 m in the Argentine
Basin. Berg (2008) calculated a $SO_4^{2-}$ flux of -8.05 mol/m²-kyr for a site with an SMT at 16 m at
the Costa Rican Margin.




*5.6 Other Chemistry*

A well-documented sequence of reactions characterize shallow marine sediment
(Froelich et al., 1979; Berner, 1980). Microbial communities preferentially utilize the most
energetically favorable oxidant available (Froelich et al., 1979; D'Hondt et al., 2002). Thus, with
increasing depth below the seafloor, a near universal order of oxidation/reduction reactions arise:
aerobic respiration, denitrification, manganese reduction, nitrate reduction, iron reduction, sulfate
reduction, and finally methanogenesis. Importantly, these reactions impact pore water chemistry
and the depths of zones dominated by these reactions generally depend on the supply of POM to
the seafloor.

Many of the cores collected along the slope of the ESM appear to terminate in the zone of
metal oxide reduction. This is because, at most stations, Mn and Fe profiles are still increasing at
the bottom of the sampled interval (**Figure 6-10**). The relatively deep depths of metal oxide
reduction are consistent with a relatively low input of POM to the seafloor, and moreover
generally contrast with sites of high $CH_4$ concentrations in shallow sediment. From a simple
perspective, there may be insufficient POC to drive methanogenesis near the seafloor.

The station on the Lomonosov Ridge again stands apart. Here, Mn and Fe concentrations
reach maxima at 1.3 mbsf and 0.5 mbsf, respectively, and decrease below. Thus, complete
consumption of Mn and Fe occurs in the upper few meters, and methanogenesis could be
occurring below 13.9 mbsf.

*5.7 Signatures of AOM and Organoclastic Sulfate Reduction*





Some authors have used changes in DIC and $SO_4^{2-}$ concentrations between the seafloor

and the SMT to infer the relative importance of AOM and organoclastic sulfate reduction (OSR)
in marine sediments (Kastner et al. 2008b; Luo et al. 2013; Hu et al. 2015). The idea is can be
expressed by comparing $\Delta(DIC+Ca^{2+}+Mg^{2+})$ and $\Delta SO_4^{2-}$, where $Ca^{2+}$ and $Mg^{2+}$ are included to
account for loss of DIC via carbonate precipitation. The rationale lies in the fact that the C:S
ratio for AOM is 1:1 (**Equation 1**), whereas the C:S ratio for OSR is 2:1 (**Equation 8**).
However, this approach neglects two considerations: (1) changes in concentration do not directly
relate to fluxes, because of differences in diffusivities of various ionic species, and, (2) a flux of
$HCO_3^-$ from below the SMT can augment the DIC produced from AOM or OSR at or above the
SMT (Dickens and Snyder, 2009). Thus, changes in alkalinity relative to $SO_4^{2-}$ often exceed 1:1,
even at locations completely dominated by AOM (Chatterjee et al., 2011).

Rather than just comparing changes in C:S molar ratios, to interrogate the importance of

the two reactions, one might also incorporate $\delta^{13}$C-DIC value. This is because $\delta^{13}$C-DIC values
and the depth of DIC production differ considerably for AOM, OSR and methanogenesis at
many locations. We generate a figure expressing these relationships at multiple sites (**Figure 16**),
where the y-axis is:

$$\frac{\Delta(DIC + Ca^{2+} + Mg^{2+})}{\Delta(SO_4^{2-})}, \tag{9}$$

and the x-axis is: DIC*$\delta^{13}$C-DIC. The C:S ratios of dissolved species lie above 1:1 at most
locations, regardless of whether $CH_4$ exists in shallow sediment However, sites with $CH_4$ have
considerably more negative DIC*$\delta^{13}$C-DIC values Notably, all stations from the ESM, except
S31 on the Lomonosov Ridge, have modest DIC*$\delta^{13}$C-DIC values.

Two basic models help to explain the relationships in **Figure 16**. The first model assumes

all $SO_4^{2-}$ consumption occurs through OSR; whereas the second model assumes that $SO_4^{2-}$



consumption occurs via AOM and OSR, but DIC from methanogenesis also migrates upward
from below the SMT. The details of both models are included in **Appendix 1**. For the "OSR
only" model a C:S ratio of 2:1 at the mudline slowly increases as $^{13}$C-depleted carbon is
produced. The ESM stations plot near to this model. In the AOM model a C:S ratio of 2.5:1 at
the mudline decreases rapidly to an asymptotic value of 1.6:1. The additional flux of DIC from
below the SMT prevents the second model from approaching 1:1. Although the height and slope
of this model can be changed by altering the fluxes, it shows that CH$_4$ charged locations with
upward migrating DIC must have C:S molar ratios in excess of 1:1. It is possible that this upward
flux is a necessary characteristic of all sites with methanogenesis.

In summary, from general pore water considerations as well as from comparisons to pore

water profiles at other locations, sediments along the ESM continental slope do not contain
significant CH$_4$ in shallow sediment. Implicit in this finding is that sediment sequences along the
ESM lack gas hydrate. As models for gas hydrate occurrence in the Arctic (**Figure 1**) correctly
predict gas hydrate in several regions (e.g., Kvenvolden and Grantz, 1990; Max and Lowrie,
1993; Max and Johnson, 2012), our findings prompt an interesting question: why are predictions
so markedly wrong for the ESM?

*5.7 Explanations*

To understand the absence of gas hydrates on the ESM, one needs to consider the

generalities of gas hydrate occurrence in marine sediment. There are two basic conditions for gas
hydrate on continental slopes (Kvenvolden, 1993; Dickens, 2001). The first is the "potential
volume", or the pore space where physiochemical conditions (e.g., temperature, pressure,
salinity, sediment porosity) are amenable to gas hydrate formation. As stressed in previous





works, the ESM, with cold bottom water and a low geothermal gradient, has a relatively large
volume of sediment with appropriate gas hydrate stability conditions (Stranne et al., 2016). The
second is the "occupancy", or the fraction of sediment pore space with sufficient $CH_4$ to
precipitate gas hydrate. The short answer is that environmental conditions on the ESM are highly
conducive for gas hydrate, but there is little $CH_4$.

It is also important to recognize how diffusive systems operate in marine sediment.

Hundreds of pore water profiles have been generated during scientific ocean drilling expeditions,
including scores into $CH_4$ charged sediment sequences. These profiles almost universally show
connectivity of pore water chemistry over hundreds of meters (**Figure 2**). This occurs because,
given sufficient permeability and time, diffusive fluxes transport species from intervals of high
concentration to intervals of low concentration. Hence, unless some impermeable layer exists in
the sediment sequence, even $CH_4$ at depth impacts near seafloor concentrations. Indeed, ODP
Leg 164 on the outer Blake Ridge wonderfully shows this phenomenon. The uppermost gas
hydrate in sediment in this region probably lies at about 190 mbsf; nonetheless, its presence can
be observed in shallow pore water profiles, because the flux of $CH_4$ from depth drives AOM near
the seafloor (Borowski et al., 1999; Dickens, 2001). Assuming that an impermeable layer does
not exist in the upper few hundreds of meters of sediment on slopes of the ESM, the lack of gas
hydrates and $CH_4$ suggests either insufficient POC to generate $CH_4$, or substantial loss of $CH_4$
over time.

The accumulation of POC on slopes of the ESM may be relatively low over the Plio-

Pleistocene, an amount too small to drive methanogenesis. With low POC inputs, other microbial
reactions can exhaust the organic matter needed for methanogenesis. This may, in fact, explain
why the pore water chemistry suggests that metal-oxide reduction dominates the geochemical

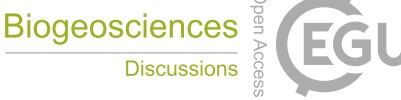

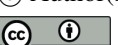

environment at most stations on the ESM. Without further investigation, we can offer three
possibilities as to why this might occur: (1) significant sea-ice concentrations, both at present-
day and during past glacial intervals, greatly diminishes primary production within the water
column, (2) the extremely broad continental shelf prevents large accumulations of terrestrial
organic rich sediment from reaching the slope, or (3) highly variable sediment accumulation,
perhaps corresponding to glacial-interglacial oscillations, creates a situation where organic
matter can be consumed during intervals of low deposition. In the latter case, large glaciers in the
past may have physically removed sediment (and organic matter) from the slope (Jakobsson et
al., 2014)

There is also the issue of POC that likely accumulated in the Cretaceous through early

Eocene. In theory, organic-rich sediment accumulated around the Arctic during this time, which
should have generated $CH_4$. This $CH_4$ could either be too deeply buried to migrate into the
modern GHSZ or have been lost in the intervening time.

**822    6. Conclusions**

Leg 2 of the SWERUS-C3 expedition recovered sediments and pore waters from

numerous stations across the ESM continental slope. These stations extend from Wrangel Island
to the New Siberian Islands, and give information from a climatically sensitive but highly
inaccessible area.

In an effort to understand $CH_4$ cycling on the ESM continental slope, we generated

detailed pore water profiles of multiple dissolved constituents at the stations. The pore water
profiles are coherent and interpretable, and give a general view: most stations have low $SO_4^{2-}$ and
$HCO_3^-$ fluxes (<9.2 and 6.8 mol/m$^2$-kyr respectively), a moderate decrease in $\delta^{13}$C-DIC values





with depth (-3.6‰/m average), no dissolved $H_2S$, moderate rise in $HPO_4^{2-}$ and $NH_4$
concentrations, and slightly decreasing $Ca^{2+}$, $Mg^{2+}$, and $Sr^{2+}$ concentrations. Except for one
station on the Lomonosov Ridge, metal oxide reduction appears to be the dominant geochemical
environment affecting shallow sediment, and there is no evidence for upward diffusing $CH_4$.
These results strongly suggest that gas hydrates do not occur on slopes of the ESM. This directly
conflicts with multiple publications, which have assumed large quantities of $CH_4$ and gas hydrate
in the region. It is possible that $CH_4$ and gas hydrate occur where the Lomonosov Ridge
intersects the ESM.

The contradiction between models for gas hydrate in the region and actual data may arise

for two basic reasons. First, in relatively recent geological times, insufficient POC accumulates
along the slope to form $CH_4$ and gas hydrates; second, $CH_4$ generated from POC deposited in
older geological times is too deeply buried or has been lost.


**Acknowledgments**. The authors would like to thank the SWERUS-C3 Leg 2 crew as well as
reviewers.



**Table List**

**Table 1 -** Rhizon Efficacy

**Table 2 -** Rhizon Flow Rates

**Table 3 -** QA/QC

**Table 4 -** Published and Calculated Fluxes

a = Coffin et al., 2013; b = Personal Communication; c = Coffin et al., 2007; d = Coffin et al., 2006; e = Coffin et al., 2008; f = Hamdan et al., 2011 and Coffin et al., 2014; g = Dickens and Snyder, 2009; h= Snyder et al., 2007; i = Mountain et al., 1994; j = Lin et al., 2006; k = Berelson et al., 2005; l = Hensen et al., 2003; m = Dickens, 2001; n = Geprags et al., 2016; o = Claypool et al., 2006; p = Keigwin et al., 1998; q = Berg, 2008; r = Borowski et al., 2000; s = D'Hondt et al., 2002; t = D'Hondt et al., 2004; u = Torres et al., 2009; v = Burns, 1998; w = Kastner et al., 2008; x = Paull et al., 1996; y = Flood et al., 1995; z = Wefer et al., 1998; 1 = Prell et al., 1998; 2 = Takahashi et al., 2011; 3 = Riedel et al., 2006; 4 = Tamaki et al., 1990; 5 = Lyle et al., 1997; 6 = Moore et al., 2001; 7 = Kimura et al., 1997; 8 = Suess et al., 1988; 9 = D'Hondt et al., 2003. [‡] = Calculated from published material.

**Table S1 -** All Results

**Figure Captions**

**Figure 1.** Generalized Arctic map with background from GeoMapApp (http://www.geomapapp.org; Ryan et al., 2009). Inserted gas hydrate models based on Max and Lowrie, 1993; Max and Johnson, 2012; and Soloviev, 2002.

**Figure 2.** Idealized pore water concentration profiles for high and low upward methane flux. Discrete data points for sites 722 (Arabian Sea; Seifert and Michaelis, 1991; D'Hondt et al., 2002) and 1230 (offshore Peru; Donohue et al., 2006) are given as reference.

**Figure 3**. Bathymetric map of the Eurasian Arctic showing the overall cruise track of Leg 2, along with the four transects and coring locations.





878 **Figure 4**. Rhizon sampling of S28 (a) overall core with Rhizon samples inserted and attached to

879 syringes; (b) close-up showing pore water filling syringes.


881 **Figure 5.** Measured temperature and pH of Station 33 over 24 hours showing temperature

882 increase and concomitant decrease in pH. Only three pH profiles were collected due to pH meter

883 failure.


885 **Figure 6.** Transect 1 results. ACEX results (grey triangles; Backman et al., 2009) and IAPSO

886 standard seawater (black dotted line) shown for comparison.


888 **Figure 7.** Transect 2 results. ACEX results (grey triangles; Backman et al., 2009) and IAPSO

889 standard seawater (black dotted line) shown for comparison.


891 **Figure 8.** Transect 3 results. ACEX results (grey triangles; Backman et al., 2009) and IAPSO

892 standard seawater (black dotted line) shown for comparison.


894 **Figure 9.** Transect 4 results. ACEX results (grey triangles; Backman et al., 2009) and IAPSO

895 standard seawater (black dotted line) shown for comparison.


897 **Figure 10.** Lomonosov Ridge Station results. ACEX results (grey triangles; Backman et al.,

898 2009), IAPSO standard seawater (black dotted line), and representative stations from the four

899 transects shown for comparison.


901 **Figure 11.** Calcium "error" with sampling time. X-Axis equal to duration of time between core

902 retrieval and rhizon pore water completion.


904 **Figure 12.** Relationship of (a) porosity and (b) rhizon extraction rate revealing the (c)

905 exponential correlation in flow rate with porosities commonly observed in piston, gravity, and

906 multicores.






**Figure 13.** Relationship of (a) sulfate change ($\Delta SO_4^{2-}$) and carbonate corrected alkalinity change ($\Delta Alk + Ca^{2+} + Mg^{2+}$) following 2:1 ratio; (b) the second order polynomial association of $NH_4^+$ to Alkalinity; and (c) decreasing $\delta^{13}C$-DIC values with alkalinity increase. Methane charged sites (1230, 1426, and 1427; 1230, Shipboard Scientific Party, 2003; 1426 and 1427, Expedition Scientists, 2014) given for comparison.

**Figure 14.** C:N:P ratio indirectly shown with $\Delta Alk/\Delta NH_4^+$ and $\Delta Alk/\Delta HPO_4^{2-}$. Several global sites, 994, 995, 997, 1059, 1225, 1230, 1426, 1427, and 1319 (994-997, 1059, Borowski et al., 2000; 1225 and 1230, Shipboard Scientific Party, 2003; 1426 and 1427, Expedition Scientists, 2014) given for comparison. Blue marginal distribution curves show global distribution while red gives ESM stations (this project). ESM pore waters have higher C:N and lower C:P than comparative sites.

**Figure 15.** Bicarbonate ($HCO_3^-$) and sulfate ($SO_4^{2-}$) flux exponential relationship with SMT depth for all sites listed in Table 4.

**Figure 16.** Ratio of carbonate corrected alkalinity change ($\Delta Alk + Ca^{2+} + Mg^{2+}$) and sulfate change ($\Delta SO_4^{2-}$) to the product of DIC and $\delta_{13}C$-DIC value (AT13-2 and KC151, Kastner et al., 2008a; PC02-PC14, Coffin et al., 2008; 994-997, 1059, Borowski et al., 2000; Paull et al., 2000; 1326 and 1329, Torres and Kastner, 2009; GC233 and GB425, Hu et al., 2010; D-5 – D-8 and D-F, Hu et al., 2015; C9-C19, Luo et al., 2013; PC-07, Smith and Coffin, 2014; 1230, Shipboard Scientific Party, 2003; 1244 and 1247, Claypool et al., 2006; 1305 and 1306, Party, 2005) including global sites for comparison) showing the paucity of methane charged sites actually reaching 1:1 C:S ratio. Two simple models of OSR and OSR + AOM (following Chatterjee et al., 2011; and Malinverno and Pohlman, 2011); given as dotted lines. When an additional flux of $HCO_3^-$ is added from below the SMT the C:S ratio is unlikely to reach 1:1.

Error bars are one sigma. ESM plotted pore waters substitute alkalinity for DIC. With the absence of sulfide, DIC and alkalinity should be roughly equivalent in these pore waters. ESM locations use the same symbols as previous figures.



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

Expedition 346 Scientists: Asian Monsoon: onset and evolution of millennial-scale variability of
Asian monsoon and its possible relation with Himalaya and Tibetan Plateau uplift, IODP
Preliminary Report, 346, 2014.
Eiken, O. and Hinz, K.: Contourites in the Fram Strait, Sediment Geol., 82, 15–32, 1993.



Ferré, B., Mienert, J., and Feseker, T.: Ocean temperature variability for the past 60 years on the

Norwegian-Svalbard margin influences gas hydrate stability on human time scales, J.

Geophys. Res., 117, 1-14, 2012.

Flood, R. D., Piper, D. J. W., Klaus, A., and Scientific Research Party: Proceedings of the Ocean

Drilling Program, Initial Reports, 155: College Station, TX (Ocean Drilling Program),

1995.

Froelich, P., Klinkhammer, G. P., Bender, M. A. A., Luedtke, N. A., Heath, G. R., Cullen, D.,

Dauphin, P., Hammond, D., Hartman, B., and Maynard, V.: Early oxidation of organic

matter in pelagic sediments of the eastern equatorial Atlantic: suboxic diagenesis,

Geochim. Cosmochim. Ac., 43, 1075-1090, 1979.

Gao, H., Schreiber, F., Collins, G., Jensen, M. M., Kostka, J. E., Lavik, G., de Beer, D., Zhou, H.

Y., and Kuypers, M. M.: Aerobic denitrification in permeable Wadden Sea sediments,

The ISME J., 4, 417-426, 2010.

Geprägs, P., Torres, M. E., Mau, S., Kasten, S., Römer, M., and Bohrmann, G.: Carbon cycling

fed by methane seepage at the shallow Cumberland Bay, South Georgia, sub-Antarctic,

Geochem. Geophy. Geosy., 17, 1401-1418, 2016.

Gieskes, J. M. and Rogers, W. C.: Alkalinity determination in interstitial waters of marine

sediments, J. Sediment. Res., 43(1), 272-277, 1973.

Gieskes J. M.: The alkalinity-total carbon dioxide system in seawater, In The Sea, vol. 5 (ed. E.

D. Goldberg). John Wiley and Sons, New York, 123–151, 1974.

Gieskes, J. M., Gamo, T., and Brumsack, H.: Chemical methods for interstitial water analysis

aboard JOIDES Resolution, 1991.



Gingele, F. and Dahmke, A.: Discrete barite particles and barium as tracers of paleoproductivity

in South Atlantic sediments, Paleoceanography, 9, 151–168, 1994.

Giustiniani, M., Tinivella, U., Jakobosson, M., and Rebesco, M.: Arctic Ocean gas hydrate

stability in a changing climate, J. Geol. Res., 1-10, 2013.

Goldhaber, M.: Kinetic Models of Sulfur Diagenesis in Recent Marine Sediments, In

Transactions-AGU, 55, 696-697, 1974.

Grantz, A., Boucher, G., and Whitney, O. T.: Possible solid gas hydrate and natural gas deposits

beneath the continental slope of the Beaufort Sea, U.S. Geological Survey Circulation

Number 733, 1976.

Grantz, A., Mann, D. M., and May, S. D.: Tracklines of multichannel seismic-reflection data

collected by the U.S. Geological Survey in the Beaufort and Chukchi Seas in 1977 for

which profiles and stack tapes are available, U.S. Geological Survey Open-File Report

1982, 82-735, 1982.

Greinert, J., Bohrmann, G., and Suess, E.: Gas hydrate-associated carbonates and methane-

venting at Hydrate Ridge: classification, distribution, and origin of authigenic lithologies,

Natural gas hydrates: Occurrence, distribution, and detection, 99-113, 2001.

Gualtieri, L., Vartanyan, S., Brigham-Grette, J., and Anderson, P. M.: Evidence for an ice-free

Wrangel Island, Northeast Siberia during the Last Glacial Maximum, Boreas 34, 264–

273, 2005.

Gusev, E., Anikina, N., Andreeva, I., Bondarenko, S., Derevyanko, L., Iosifidi, A., Klyuvitkina,

1159       T., Litvinenko, I., Petrova, V., Polyakova, E., Popov, V., and Stepanova, A.: Stratigraphy

of Late Cenozoic sediments of the western Chukchi Sea: New results of shallow drilling

and seismic reflection profiling, Global Planet. Change, 68, 115–131, 2009.





Haacke, R. R., Westbrook, G. K., and Riley, M.: Controls on the formation and stability of gas

hydrate-related bottom-simulating reflectors (BSRs): a case study from the west Svalbard

continental slope, J. Geophys. Res., 113, 1-17, 2008.

Hamdan, L. J., Gillevet, P. M., Pohlman, J. W., Sikaroodi, M., Greinert, J., and Coffin, R.B.:

Diversity and biogeochemical structuring of bacterial communities across the Porangahau

ridge accretionary prism, New Zealand, FEMS Microbiol. Ecol., 77, 518-532, 2011.

Haraldsson, C., Anderson, L. G., Hassellöv, M., Hulth, S., and Olsson, K.: Rapid, high-precision

potentiometric titration of alkalinity in ocean and sediment pore waters, Deep Sea

Research Part I: Oceanographic Research Papers, 44, 2031-2044, 1997.

Hart, P. E., Pohlman, J. W., Lorenson, T. D., and Edwards, B. D.: Beaufort Sea Deep-water gas

hydrate recovery from a seafloor mound in a region of widespread BSR occurence, in

Proceedings of the 7th International Conference on Gas Hydrates (ICGH 2011),

Edinburgh, Scotland, 2011.

Hensen, C., Zabel, M., Pfiefer, K., Schwenk, T., Kasten, S., Riedinger, N., Schulz, H. D.,

Boetius, A.: Control of sulfate porewater profiles by sedimentary events and the

significance of anaerobic oxidation of methane for the burial of sulfur in marine

sediments, Geochimica Cosmochimica Acta 67, 2631-2647, 2003.

Holbrook, W. S., Hoskins, H., Wood, W. T., Stephen, R. A., and Lizarralde, D.: Methane hydrate

and free gas on the Blake ridge from vertical seismic profiling, Science, 273, 1840–1843,

1996.

Holmes, R. M., McClelland, J. W., Peterson, B. J., Shiklomanov, I. A., Shiklomanov, A. I.,

Zhulidov, A. V., Gordeev, V. V., and Bobrovitskaya, N. N.: A circumpolar perspective

on fluvial sediment flux to the Arctic Ocean, Global Biogeochem. Cy., 16, 1-14, 2002.



Holler, T., Wegener, G., Knittel, K., Boetius, A., Brunner, B., Kuypers, M. M., and Widdel, F.:
Substantial 13C/12C and D/H fractionation during anaerobic oxidation of methane by
marine consortia enriched in vitro, Environ. Microbiol. Rep., 1, 370-376, 2009.

Hovland, M. and Svensen, H.: Submarine pingoes: Indicators of shallow gas hydrates in a
pockmark at Nyegga, Norwegian Sea, Mar. Geol., 228, 15–23, 2006.

Hu, X., Cai, W. -J, Wang, Y., Luo, S., and Guo X.: Pore-water geochemistry of two contrasting
brine-charged seep stations in the northern Gulf of Mexico continental slope, Mar.
Geochem., 118, 99–107, 2010.

Hu, Y., Feng, D., Liang, Q., Xia, Z., Chen, L., and Chen, D.: Impact of anaerobic oxidation of
methane on the geochemical cycle of redox-sensitive elements at cold-seep stations of the
northern South China Sea. Deep Sea Research Part II: Topical Studies in Oceanography,
2015.

Hustoft, S., Bünz, S., Mienert, J., and Chand, S.: Gas hydrate reservoir and active methane-
venting province in sediments on <20Ma young oceanic crust in the Fram Strait, offshore
NW-Svalbard, Earth Planet. Sc. Lett., 284, 12-24, 2009.

Hyndman, R. D. and Dallimore, S. R.: Natural gas hydrates studies, Canada Recorder, 26, 11–20,
2001.

Jakobsson, M.: Hypsometry and volume of the Arctic Ocean and its constituent seas:
Geochemistry, Geophysics, Geosystems, 3, 5, 1-18, 2002

Jakobsson, M., Grantz, A., Kristoffersen, Y., and Macnab, R.: Physiographic provinces of the
Arctic Ocean seafloor, GSA Bull., 115, 1443-1455, 2003.

Jakobsson, M., Backman, J., Rudels, B., Nycander, J., Frank, M., Mayer, L., Jokat, W.,
Sangiorgi, F., O'Regan, M., Brinkhuis, H., King, J., and Moran, K.: The early Miocene



Onset of a Ventilated Circulation Regime in the Arctic Ocean: Nature, 447, 21, 986-990,

2007.

Jakobsson, M., Polyak, L., Edwards, M., Kleman, J., and Coakley, B.: Glacial geomorphology of
the Central Arctic Ocean: the Chukchi Borderland and the Lomonosov Ridge, Earth Surf.
Proc. Land., 33, 526-545, 2008.
Jakobsson, M., Andreassen, K., Bjarnadóttir, L. R., Dove, D., Dowdeswell, J. A., England, J. H.,
Funder, S., Hogan, K., Ingólfsson, Ó., Jennings, A., Krog-Larsen, N., Kirchner, N.,
Landvik, J. Y., Mayer, L., Möller, P., Niessen, F., Nilsson, J., O'Regan, M., Polyak, L.,
Nørgaard-Pedersen, N., and Stein, R.: Arctic Ocean glacial history, Quaternary Sci. Rev.,

92, 40-67, 2014.

Jørgensen, B. B., Bang, M., and Blackburn, T. H.: Anaerobic mineralization in marine sediments
from the Baltic Sea-North Sea transition, Mar. Ecol. Progress Series, 59, 39-54, 1990.
Jørgensen, B. B., Weber, A. and Zopfi, J.: Sulfate reduction and anaerobic methane oxidation in
Black Sea sediments. Deep Sea Research Part I: Oceanographic Research Papers, 48 (9),

2097-2120, 2001.

Jørgensen, B. B., Böttcher, M. E., Lüschen, H., Neretin, L. N., and Volkov, I. I.: Anaerobic
methane oxidation and the deep H2S sink generate isotopically heavy sulfides in Black
Sea sediments, Geochim. Cosmochim. Ac., 68, 2095–2118, 2004.
Jørgensen, B. B. and Parkes, R. J.: Role of sulfate reduction and methane production by organic
carbon degradation in eutrophic fjord sediments (Limfjorden, Denmark), Limnol.
Oceanogr, 55, 1338-1352, 2010.



Jokat, W.: The expedition of the Research Vessel" Polarstern" to the Arctic in 2009 (ARK-
XXIV/3). Berichte zur Polar-und Meeresforschung (Reports on Polar and Marine
Research), 615, 2010.
Jokat, W. and Ickrath, M.: Structure of ridges and basins off East Siberia along 81° N, Arctic
Ocean, Mar. Petrol. Geol., 64, 222-232, 2015.
Joye, S. B., Boetius, A., Orcutt, B. N., Montoya, J. P., Schulz, H. N., Erickson, M. J. and Lugo,
S.K.: The anaerobic oxidation of methane and sulfate reduction in sediments from Gulf
of Mexico cold seeps. Chemical Geology, 205 (3), 219-238, 2004.
Jenkyns, H. C., Forster, A., Schouten, S., and Sinninghe Damsté, J. S.: High temperatures in the
Late Cretaceous Arctic Ocean, Nature, 432, 888–892, 2004.
Judge, A. S.: Natural gas hydrates in Canada, In: M.H. French (Editor), Proceedings of the
Fourth Canadian Permafrost Conference 1981, Roger J.E. Brown Memorial Volume,
National Research Council of Canada, Ottawa, Ont., 320-328, 1982.
Kastner, M., Claypool, G., and Robertson, G.: Geochemical constraints on the origin of the pore
fluids and gas hydrate distribution at Atwater Valley and Keathley Canyon, northern Gulf
of Mexico, Mar. Petr. Geol., 25, 860–872, 2008a.
Kastner, M., Torres, M., Solomon, E., and Spivack, A. J.: Marine pore fluid profiles of dissolved
sulfate; do they reflect in situ methane fluxes?, In: Fire in the Ice, NETL Methane
Hydrate Newsletter, Summer, 2008b.
Keigwin, L. D., Rio, D., Acton, G. D., and Shipboard Scientific Party: Proceedings of the Ocean
Drilling Program, Initial Reports, 172: College Station, TX (Ocean Drilling Program),

1998.





Kimura, G., Silver, E. A., Blum, P., Shipboard Scientific Party: Proceedings of the Ocean

Drilling Program, Initial Reports, Vol. 170, 1997.

Klauda, J. B. and Sandler, S. I.: Global distribution of methane hydrate in ocean sediment,

Energy Fuel, 19, 469–78, 2005.

Klump, J. V. and Martens, C. S.: Biogeochemical cycling in an organic rich coastal marine

basin—II. Nutrient sediment-water exchange processes, Geochim. Cosmochim. Ac., 45,

101-121, 1981.

Kvenvolden, K. A., and Grantz, A.: Gas hydrates in the Arctic Ocean region, in The Arctic

Ocean Region, Geology of North America, Geol. Soc. of Am., Boulder, Colo., 539-549,

1990.

Kvenvolden, K. A.: Gas hydrates: Geological perspective and global change, Rev. Geophys., 31,

173–187, 1993.

Kvenvolden, K. A. and Lorenson, T. D.: The global occurrence of natural gas hydrate. In: Paull,

C. K., Dillon, W. P. (Eds.), Natural Gas Hydrates: Occurrence, Distribution, and

Detection, AGU Geophy. Monograph Ser., 124, 3–18, 2001.

Laberg, J. S. and Andreassen, K.: Gas hydrate and free gas indications within the Cenozoic

succession of the Bojornya Basin, western Barents Sea, Mar. Petrol. Geol., 13, 921-940,

1996.

Laberg, J. S., Andreassen, K., and Knutsen, S. M.: Inferred gas hydrate on the Barents Sea shelf

a model for its formation and a volume estimate, Geo-Mar. Lett., 18, 26–33, 1998.

Lerman, A.: Migrational processes and chemical reactions in Sulfate profiles and barium fronts

in sediment 539 interstitial waters, In The Sea, Volume VI (ed. E. D. Goldberg), Wiley,

New York, 695-738, 1977.

Li, Y-H. and Gregory, S.: Diffusion of ions in sea sediments. Geochim. Cosmochim. Ac., 38,

703-714, 1974.

Lin, S., Hsieh, W. C., Lim, Y. C., Yang, T. F., Liu, C. S., and Wang, Y.: Methane migration and

its influence on sulfate reduction in the Good Weather Ridge region, South China Sea

continental margin sediments, Terr. Atmos. Ocean. Sci., 17, 883-902, 2006.

Lorenson, T. D. and Kvenvolden, K. A.: Methane in coastal seawater, sea ice, and bottom

sediments, Beaufort Sea, Alaska, USGS Open-File Report 95–70, 1995.

Løvø, V., Elverhøi, A., Antonsen, P., Solheim, A., Butenko, G., Gregersen, O., and Liestøl O.:

Submarine permafrost and gas hydrates in the northern Barents Sea, Norsk Polarinstitutt

Rapportserie, 56, 1-171, 1990.

Luff, R. and Wallmann, K.: Fluid flow, methane fluxes, carbonate precipitation and

biogeochemical turnover in gas hydrate-bearing sediments at hydrate ridge, Cascadia

margin: numerical modeling and mass balances, Geochim. Cosmochim. Ac., 67, 3403–

3421, 2003.

Luo, M., Chen, L., Wang, S., Yan, W., Wang, H., and Chen, D.: Pockmark activity inferred from

pore water geochemistry in shallow sediments of the pockmark field in southwestern

Xisha Uplift, northwestern South China Sea, Mar. Pet. Geol., 2013, 48, 247–259, 2013.

Lyle, M., Koizumi, I., Richter, C., and Shipboard Scientific Party: Proceedings of the Ocean

Drilling Program, Initial Reports, Vol. 167, 1997.

MacKay, M. E., Jarrard, R. D., Westbrook, G. K., Hyndman, R. D., and Shipboard Scientific

Party: Origin of bottom-simulating reflectors: geophysical evidence from the Cascadia

accretionary prism, Geol., 22, 459–462, 1994.



Majorowicz, J. A. and Osadetz, K. G.: Gas hydrate distribution and volume in Canada. AAPG

bulletin, 85 (7), 1211-1230, 2001.

Makogon, Y. F.: Natural gas hydrates–A promising source of energy, J. Nat. Gas Sc. Eng., 2, 49-

59, 2010.

Malinverno, A. and Pohlman, J. W.: Modeling sulfate reduction in methane hydrate-bearing

continental margin sediments: does a sulfate-methane transition require anaerobic

oxidation of methane, Geochem. Geophy. Geosy., 12, 1-18, 2011.

Max, M. D. and Lowrie, A.: Natural gas hydrates: Arctic and Nordic Sea potential. Arctic

geology and petroleum potential, proceedings of the Norwegian Petroleum Society

conference, 15-17 August 1990, Tromsø, Norway. No. 2. Elsevier Science Ltd., 1993.

Max, M. D. and Johnson, A. H.: Natural Gas Hydrate (NGH) Arctic Ocean potential prospects

and resource base, OTC Arctic Technology Conference, 27-53, 2012.

McGuire, A. D., Anderson, L. G., Christensen, T. R., Dallimore, S., Guo, L. D., Hayes, D. J.,

Heimann, M., Lorenson, D. D., MacDonald, R. W., and Roulet, N.: Sensitivity of the

carbon cycle in the Arctic to climate change, Ecol. Monogr., 79, 523–555, 2009.

Miles, P. R.: Potential distribution of methane hydrate beneath the European continental margins,

Geophys. Res. Lett., 22, 3179-3182, 1995.

Miller, M. D., Adkins, J. F., and Hodell, D. A.: Rhizon sampler alteration of deep ocean

sediment interstitial water samples, as indicated by chloride concentration and oxygen

and hydrogen isotopes, Geochem. Geophy. Geosy., 15, 2401-2413, 2014.

Moore, G. F., Taira, A., and Klaus, A., and Shipboard Scientific Party: Proceedings of the Ocean

Drilling Program, Initial Reports Volume 190, 2001.



Moran, K., Backman, J., Brinkhuis, H., Clemens, S. C., Cronin, T., Dickens, G. R., Eynaud, F.,

Gattacceca, J., Jakobsson, M., Jordan, R.W., and Kaminski, M.: The Cenozoic

palaeoenvironment of the arctic ocean, Nature, 441, 601-605, 2006.

Mountain, G. S., Miller, K. G., Blum, P., and Shipboard Scientific Party: Proc. ODP, Initial

Reports, 150: College Station, TX (Ocean Drilling Program), 1994.

Murray, R. W., Miller, D. J., and Kryc, K. A.: Analysis of major and trace elements in rocks,

sediments, and interstitial waters by inductively coupled plasma–atomic emission

spectrometry (ICP-AES), ODP Technical Note 29, 2000.

Naidu, A. S., Cooper, L. W., Finney, B. P., Macdonald, R. W., Alexander, C., and Semiletov, I.

P.: Organic carbon isotope ratios ($\delta$13C) of Arctic Amerasian continental shelf sediments,

Int. J. Earth Sci., 89, 522–532, 2000.

Niessen, F., Hong, J. K., Hegewald, A., Matthiessen, J., Stein, R., Kim, H., Kim, S., Jensen, L.,

Jokat, W., Nam, S.-I., and Kang, S.-H.: Repeated Pleistocene glaciation of the East

Siberian Continental Margin, Nat. Geosci., 6, 842-846, 2013.

Niewohner, C., Hensen, C., Kasten, S., Zabel, M., and Schulz, H. D.: Deep sulfate reduction

completely mediated by anaerobic methane oxidation in sediments of the upwelling area

off Namibia, Geochim. Cosmochim. Ac., 62, 455–464, 1998.

Nöthen, K. and Kasten, S.: Reconstructing changes in seep activity by means of pore water and

solid phase Sr/Ca and Mg/Ca ratios in pockmark sediments of the Northern Congo Fan,

Mar. Geol., 287, 1-13, 2011.

O'Regan, M., Williams, C. J., Frey, K. E., Jakobsson, M.: A synthesis of the long-term

paleoclimatic evolution of the Arctic. Oceanography 24(3), 66–80, 2011



O'Regan, M., Preto, P., Stranne, C., Jakobsson, M., and Koshurnikov, A.: Surface heat flow

measurements from the East Siberian continental slope and southern Lomonosov Ridge,

Arctic Ocean, Geochem. Geophy. Geosy., 17, 1-15, 2016.

Ostanin, I., Anka, Z., di Primio, R. and Bernal, A.: Hydrocarbon plumbing systems above the

Snøhvit gas field: structural control and implications for thermogenic methane leakage in

the Hammerfest Basin, SW Barents Sea. Marine and Petroleum Geology, 43, 127-146,

2013.

Party, S. S.: Integrated Ocean Drilling Program Expedition 303 Preliminary Report North

Atlantic Climate Ice sheet–ocean atmosphere interactions on millennial timescales during

the late Neogene-Quaternary using a paleointensity-assisted chronology for the North

Atlantic, 1-51, 2005.

Paull, C. K., Ussler, W. III, and Dillon, W. P.: Is the extent of glaciation limited by marine gas-

hydrates?, Geophys. Res. Lett., 18 432–434, 1991.

Paull, C. K., Matsumoto, R., Wallace, P. J., and Shipboard Scientific Party: Proceedings of the

IODP, Initial Reports. Volume 164: College Station, TX, USA, 1996.

Paull, C. K, Lorenson, T. D., Borowski, W. S., Ussler, III W., Olsen, K., and Rodriguez, N. M.:

Isotopic composition of $CH_4$, $CO_2$ species, and sedimentary organic matter within

samples from the Blake Ridge: gas source implications, In: Paull C. K., Matsumoto, R,

Wallace, P. J., and Dillon, W. P., (Eds) Proceedings of the ODP, Sci. Res., Vol. 164 67–

78, (ODP), 2000.

Pecher, I. A., Kukowski, N., Huebscher, C., Greinert, J., and Bialas, J.: The link between bottom-

simulating reflections and methane flux into the gas hydrate stability zone - new evidence

from Lima Basin, Peru Margin, Earth Planet. Sc. Lett., 185, 343-354, 2001.



Peterson, B. J., Holmes, R. M., McClelland, J. W., Vorosmarty, C. J., Lammers, R. B.,
Shiklomanov, A. I., Shiklomanov, I. A., and Rahmstorf, S.: Increasing river discharge to
the Arctic Ocean, Science, 298, 2171–2173, 2002.
Petersen, C. J., Bünz, S., Hustoft, S., Mienert, J., and Klaeschen, D.: High-resolution P-Cable 3D
seismic imaging of gas chimney structures in gas hydrated sediments of an Arctic
sediment drift, Mar. Petrol. Geol., 1–14, 2010.
Phrampus, B. J., Hornbach, M. J., Ruppel, C. D., and Hart P. E.: Widespread gas hydrate
instability on the upper US Beaufort margin, J. Geophys. Res-Sol. Ea., 119, 8594–8609,

2014

Piñero, E., Marquardt, M., Hensen, C., Haeckel, M., and Wallmann K.: Estimation of the global
inventory of methane hydrates in marine sediments using transfer functions,
Biogeosciences 10, 959–975, 2013.
Pohlman, J. W., Riedel, M., Waite, W., Rose, K., and Lapham, L.: Application of Rhizon
samplers to obtain high-resolution pore-fluid records during geo-chemical investigations
of gas hydrate systems, Fire in the Ice: Methane Hydrate Newsletter, US Department of
Energy/National Energy Technology Laboratory, Fall, 2008.
Posewang, J. and Mienert, J.: High-resolution seismic studies of gas hydrates west of Svalbard,
Geo-Mar. Lett., 19, 150–156, 1999.
Prell, W. L., Niitsuma, N., and Shipboard Scientific Party: Proceedings of the Ocean Drilling
Program, Initial Reports, 117: College Station, TX (Ocean Drilling Program), 1998.
Rachor, E.: The expedition ARK-XI/1 of RV" Polarstern" in 1995: [ARK XI/1, Bremerhaven-
Tromsø-, 07.07. 1995-20.09. 1995]. Berichte zur Polarforschung (Reports on Polar
Research), 226, 1995.





Reagan, M. T. and Moridis, G. J.: Dynamic response of oceanic hydrate deposits to ocean

temperature change, J. Geophys. Res., 113, 1-21, 2008.

Reagan, M. T. and Moridis, G. J.: Large-scale simulation of methane hydrate dissociation along

the West Spitsbergen margin, Geophys. Res. Lett., 36, 1-5, 2009.

Redfield, A. C.: The biological control of chemical factors in the environment, Am. Sci., 46,

221-230, 1958.

Regnier, P., Dale, A. W., Arndt, S., LaRowe, D. E., Mogollón, J., and Van Cappellen, P.:

Quantitative analysis of anaerobic oxidation of methane (AOM) in marine sediments: a

modeling perspective, Earth-Sci. Rev., 106, 105-130, 2011.

Reeburgh, W. S.: Methane consumption in Cariaco Trench waters and sediments, Earth Planet

Sci. Lett., 28, 337–344, 1976.

Riedel, M., Collett, T.S., Malone, M.J., and the Expedition 311 Scientists Proceedings of the

Integrated Ocean Drilling Program, Volume 311, 2006.

Riedinger, N., Kasten, S., Gröger, J., Franke, C. and Pfeifer, K.: Active and buried authigenic

barite fronts in sediments from the Eastern Cape Basin, Earth Planet. Sc. Lett., 241, 876-

887, 2006.

Riedinger, N., Formolo, M. J., Lyons, T. W., Henkel, S., Beck, A., and Kasten, S.: An inorganic

geochemical argument for coupled anaerobic oxidation of methane and iron reduction in

marine sediments, Geobiology, 12, 172-181, 2014.

Rudels, B., Muench, R. D., Gunn, J., Schauer, U., and Friedrich, H. J.: Evolution of the Arctic

Ocean boundary current north of the Siberian shelves, Journal of Marine Systems, 25, 1,

77-99, 2000.





Ryan, W. B. F., Carbotte, S. M., Coplan, J. O., O'Hara, S., Melkonian, A., Arko, R., Weissel, R.

1409        A., Ferrini, V., Goodwillie, A., Nitsche, F., Bonczkowski, J., and Zemsky, R.: Global

Multi-Resolution Topography synthesis, Geochem. Geophys. Geosyst., 10, 1-9, 2009.

Sander, B. C. and Kalff, J.: Factors controlling bacterial production in marine and freshwater

sediments, Microb. Ecol., 26, 79-99, 1993.

Sauvage, J.: Dissolved Inorganic Carbon and Alkalinity in Marine Sedimentary Interstitial

Water, Master Thesis, University of Rhode Island, 2013.

Schrum, H. S., Murray, R. S., and Gribsholt B.: Comparison of rhizon sampling and whole round

squeezing for marine sediment porewater, Sci. Drill., 13, 47–50, 2012.

Schulz, H. D.: Quantification of early diagenesis: dissolved constituents in marine pore water, In

Mar. Geochem., Springer Berlin Heidelberg, 85-128, 2000.

Seeberg-Elverfeldt, J., Schlüter, M., Feseker, T., and Kölling, M.: Rhizon sampling of pore

waters near the sediment/water interface of aquatic systems, Limnol. Oceanogr. Methods,

3, 361-371, 2005.

Seifert, R. and Michaelis, W.: Organic compounds in sediments and pore waters of Sites 723 and

724, In Prell, W.L., Niitsuma, N., et al., Proc. ODP, Sci. Results, 117: College Station,

TX (ODP), 529–545, 1991.

Serreze, M. C., Walsh, J. E., Chapin, III F. S., Osterkamp, T., Dyurgerov, M., Romanovsky, V.,

Oechel, W. C., Morison, J., Zhang T., and Barry, R. B.: Observational evidence of recent

change in the northern high-latitude environment, Climatic Change, 46, 159–207, 2000.

Semiletov, I., Makshtas, O. A., Akasofu, S.I., and Andreas, E. L.: Atmospheric CO2 balance: the

role of Arctic sea ice, Geophys. Res. Lett., 31, 1-4, 2004.



Shakhova, N, Semiletov, I, Salyuk, A, Yusupov, V, Kosmach, D, and Gustafsson, Ö.: Extensive

Methane venting to the atmosphere from sediments of the East Siberian arctic shelf,

Science, 327, 1246–1250, 2010a.

Shakhova, N., Semiletov, I., Leifer, I., Rekant, P., Salyuk, A., and Kosmach, D.: Geochemical

and geophysical evidence of methane release from the inner East Siberian Shelf, J.

Geophys. Res., 115, 1-14, 2010b.

Sher, A. V., Kuzmina, S. A., Kuznetsova, T. V., and Sulerzhitsky, L. D.: New insights into the

Weichselian environment and climate of the East Siberian Arctic, derived from fossil

insects, plants, and mammals, Quaternary Sci. Rev., 24, 533 – 569, 2005.

Shipboard Scientific Party: Leg 201 summary, In D'Hondt, S.L., Jørgensen, B.B., Miller, D.J., et

al., Proc. ODP, Init. Repts., 201: College Station TX (Ocean Drilling Program), 1–81,

2003.

Smith, J. P. and Coffin, R. B.: Methane Flux and Authigenic Carbonate in Shallow Sediments

Overlying Methane Hydrate Bearing Strata in Alaminos Canyon, Gulf of Mexico,

Energies, 7, 6118-6141, 2014.

Snyder, G. T., Hiruta, A., Matsumoto, R., Dickens, G. R., Tomaru, H., Takeuchi, R.,

Komatsubara, J., Ishida, Y., and Yu, H.: Porewater profiles and authigenic mineralization

in shallow marine sediments above the methane-charged system on Umitaka Spur, Japan

Sea. Deep-Sea Research II 54, 1216–1239, 2007.

Soloviev, V. A.: Gas-hydrate-prone areas of the ocean and gas-hydrate accumulations, Journal of

the Conference Abstracts, 6, 158, 2002.



Spielhagen, R. F., Werner, K., Sorensen, S. A., Zamelczyk, K., Kandiano, E., Budeus, G., Husum, K., Marchitto, T. M., and Hald, M.: Enhanced modern heat transfer to the Arctic by warm Atlantic water, Science 331, 450-453, 2011.

Stein, R., Boucsein, B., and Meyer, H.: Anoxia and high primary production in the Paleogene central Arctic Ocean: First detailed records from Lomonosov Ridge, Geophys. Res. Lett., 33, 1-6, 2006.

Stein, R.: Arctic Ocean sediments: processes, proxies, and paleoenvironment, Developments in Mar. Geol., 2. Elsevier, Amsterdam. 592 pp., 2008.

Stranne, C., O'Regan, M., Dickens, G. R., Crill, P., Miller, C., Preto, P., and Jakobsson, M.: Dynamic simulations of potential methane release from East Siberian continental slope sediments, Geochem. Geophy. Geosy., 17, 872-886, 2016.

Stroeve, J. C., Serreze, M. C., Holland, M. M., Kay, J. E., Malanik, J., and Barrett, A. P.: The Arctic's rapidly shrinking sea ice cover: a research synthesis, Climatic Change, 110, 1005-1027, 2012.

Suess, E., von Huene, R., and Shipboard Scientific Party: Proceedings of the Ocean Drilling Program, Initial Reports, 112: College Station, TX (Ocean Drilling Program), 1988.

Takahashi, T. Broecker, V. S., and Langer, S.: Redfield ratio based on chemical data from isopycnal surfaces, J. Geophys. Res-Oceans, 90, 6907-6924, 1985.

Takahashi, K., Ravelo, A.C., Alvarez Zarikian, C.A., and the Expedition 323 Scientists Proceedings of the Integrated Ocean Drilling Program, Volume 323, 2011.

Tamaki, K., Pisciotto, K., Allan, J., and Shipboard Scientific Party: Proceedings of the Ocean Drilling Program, Initial Reports, Vol. 127, 1990.

Thatcher, K. E., Westbrook, G. K., Sarkar, S., and Minshull, T. A.: Methane release from

warming-induced hydrate dissociation in the West Svalbard continental margin: Timing,

rates, and geological controls, J. Geophys. Res-Sol. Ea., 118, 22–38, 2013.

Timokhov L. A.: Regional characteristics of the Laptev and the East Siberian seas: climate,

topography, ice phases, thermohaline regime, and circulation, In: Russian-German

cooperation in the Siberian Shelf seas: geo-system Laptev Sea, H. Kassens, H. W.

Hubberten, S. Priamikov and R. Stein, editors. Berichte zur Polarforschung, 144, 15-31,

1994.

Torres, M. E., Kastner, M., Wortmann, U. G., Colwell, F., and Kim, J.: Estimates of methane

production rates based on δ13C of the residual DIC in pore fluids from the Cascadia

margin, EOS, 8(52), Fall Meet. Suppl., Abstract GC14A04, 2007.

Torres, M. E. and Kastner M.: Data report: Clues about carbon cycling in methane-bearing

sediments using stable isotopes of the dissolved inorganic carbon, IODP Expedition 311,

Proceedings of the IODP, 311, 2009.

Treude, T., Krause, S., Maltby, J., Dale, A.W., Coffin, R. and Hamdan, L.J.: Sulfate reduction

and methane oxidation activity below the sulfate-methane transition zone in Alaskan

Beaufort Sea continental margin sediments: Implications for deep sulfur cycling.

Geochimica et Cosmochimica Acta, 144, 217-237, 2014

Troup, B. N., Bricker, O. P., and Bray, J. T.: Oxidation effect on the analysis of iron in the

interstitial water of recent anoxic sediments, Nature 249, 237-239, 1974.

Ussler, W. and Paull, C.K.: Rates of anaerobic oxidation of methane and authigenic carbonate

mineralization in methane-rich deep-sea sediments inferred from models and

geochemical profiles. Earth and Planetary Science Letters, 266(3), 271-287, 2008.





Vanneste, M., Guidard, S., and Mienert, J.: Bottom-simulating reflections and geothermal

gradients across the western Svalbard margin, Terra Nova, 17, 510–516, 2005.

Wallmann, K., Pinero, E., Burwicz, E., Haeckel, M., Hensen, C., Dale, A. W., and Ruepke, L.:

The global inventory of methane hydrate in marine sediments: A theoretical approach,

Energies, 5, 2449–2498, 2012.

Wang, G., Spivack, A. J., and D'Hondt, S.: Gibbs energies of reaction and microbial mutualism

in anaerobic deep subseafloor sediments of ODP Site 1226, Geochim. Cosmochim. Ac.,

74, 3938-3947, 2010.

Weaver, J. S. and Stewart, J. M.: In-situ hydrates under the Beaufort Sea Shelf, In: M.H. French

(Editor), Proceedings of the Fourth Canadian Permafrost Conference 1981, Roger J. E.

Brown Memorial Volume, Nat. Res. Council Can., Ottawa, Ont., 312-319, 1982.

Wefer, G., Berger, W. H., Richter, C., and Shipboard Scientific Party: Proceedings of the Ocean

Drilling Program, Initial Reports, 175: College Station, TX (Ocean Drilling Program),

1998.

Woodgate R. A., Aaagard, K., Muench, R. D., Gunn, J, Björk, G., Rudels, B., Roach, A. T., and

Schauer, U.: The Arctic Ocean boundary current along the Eurasian slope and the

adjacent Lomonosov Ridge: Water mass properties, transports and transformation from

moored instruments, Deep Sea Research, 48, 1757-1792, 2001.

Yamamoto, K. and Dallimore, S.: Aurora-JOGMEC-NRCan Mallik 2006–2008 Gas Hydrate

Research Project progress. In: Fire in the Ice, NETL Methane Hydrate Newsletter,

Summer 2008, 1–5, 2008.





Ye, H., Yang, T., Zhu, G., Jiang, S., and Wu, L.: Pore water geochemistry in shallow sediments

from the northeastern continental slope of the South China Sea, Mar. Petrol. Geol., 75,

68-82, 2016.

Yoshinaga, M. Y., Holler, T., Goldhammer, T., Wegener, G., Pohlman, J. W., Brunner, B.,

Kuypers, M. M., Hinrichs, K. U. and Elvert, M.: Carbon isotope equilibration during

sulphate-limited anaerobic oxidation of methane. Nature Geoscience, 7(3), 190-194,

2014.






**Figures**
**Figure 1.**

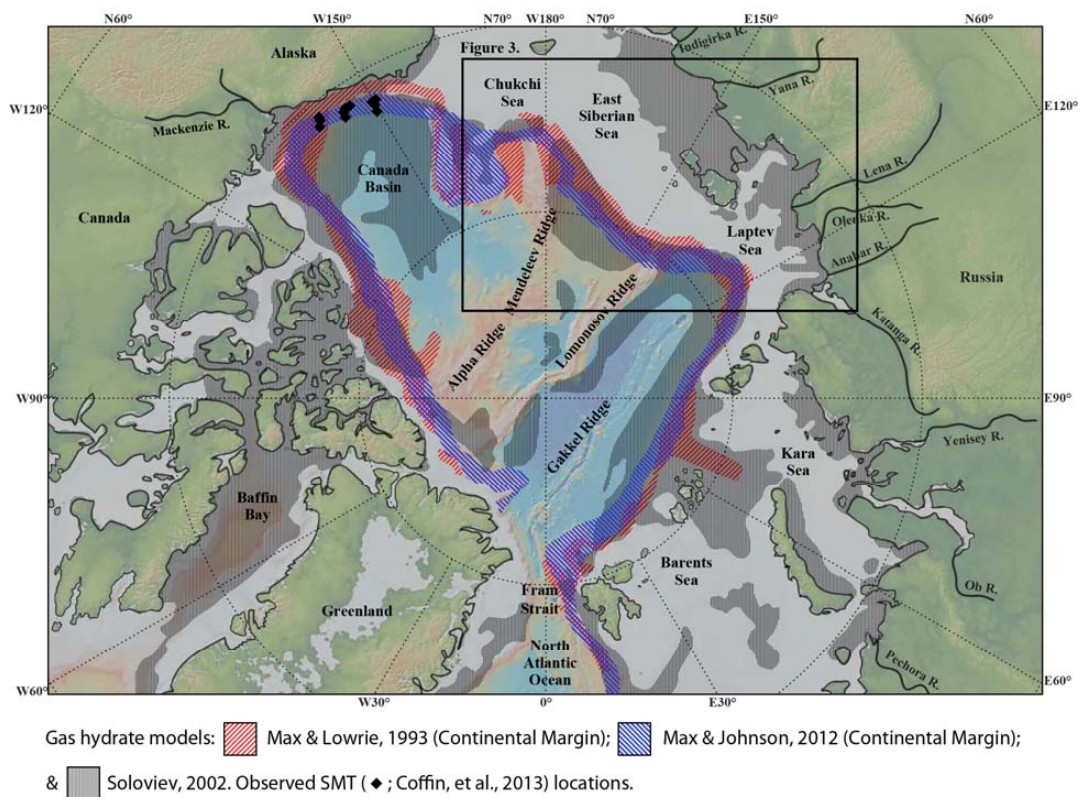






**Figure 2.**

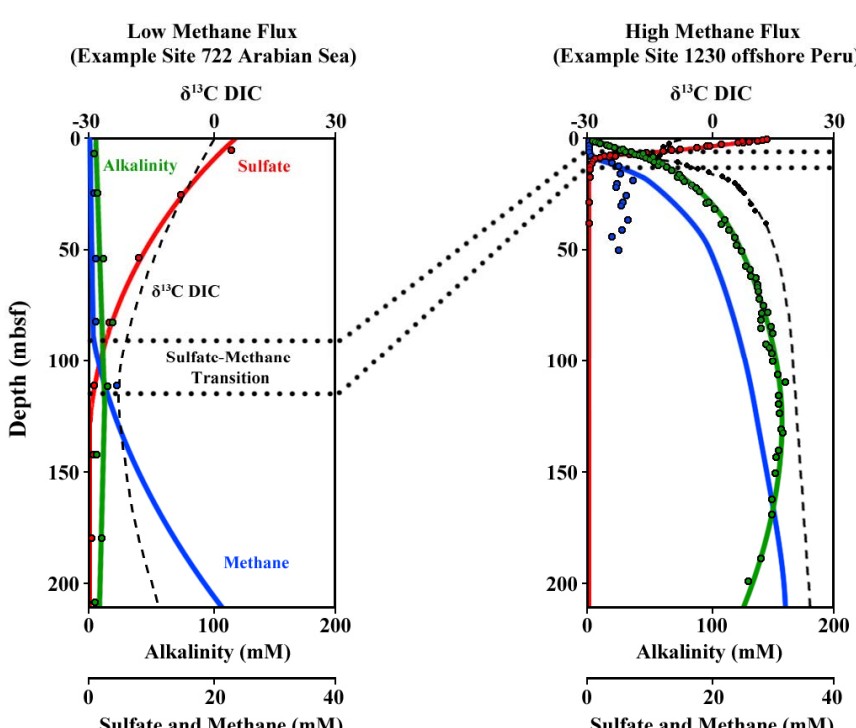


















**Figure 3.**

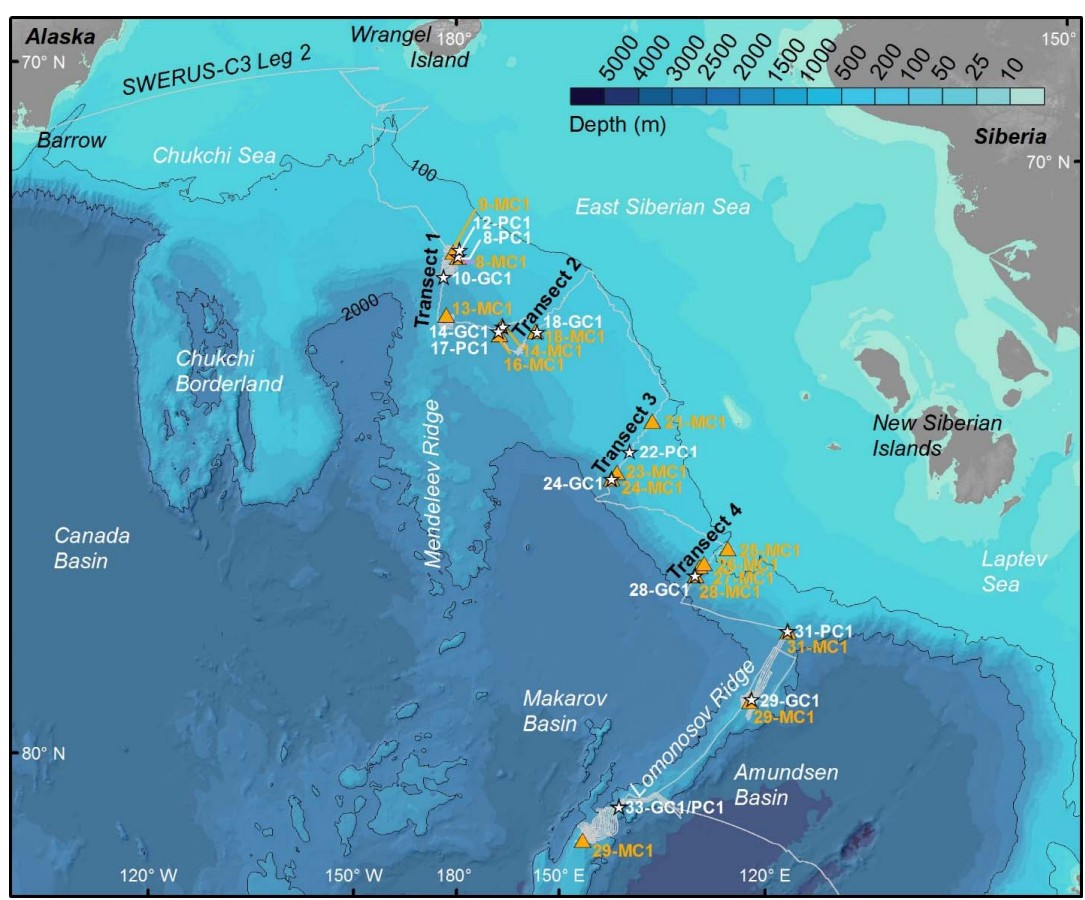



**Figure 4.**

(a.)

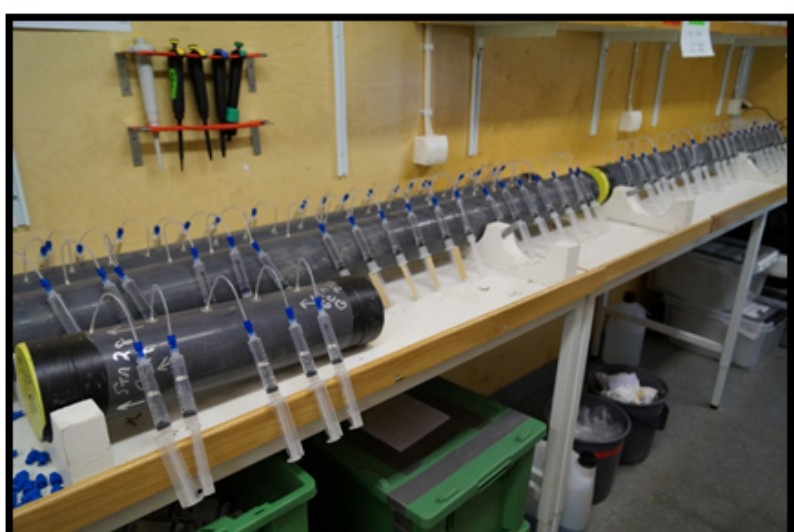

(b.)

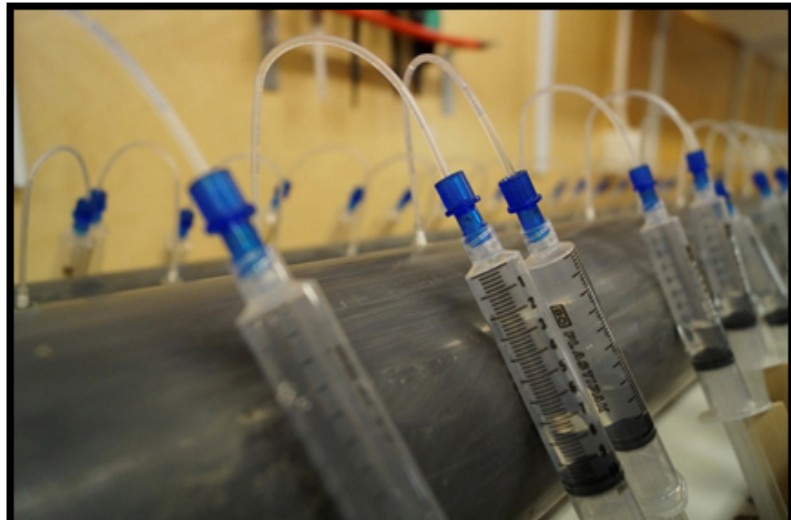







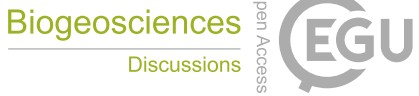



**Figure 5.**

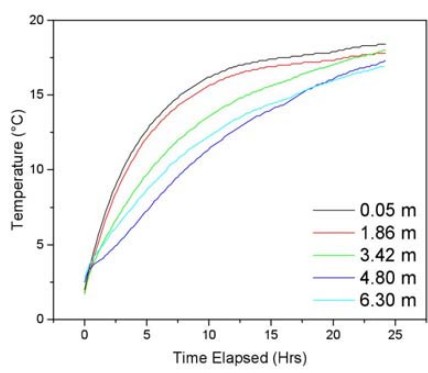

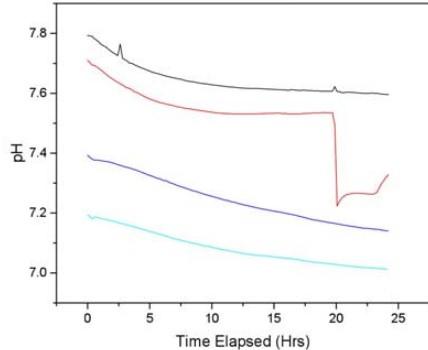




**Figure 6.**






**Figure 7.**









**Figure 8.**






**Figure 9.**







**Figure 10.**






**Figure 11.**

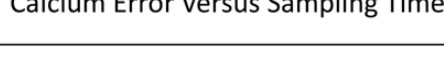

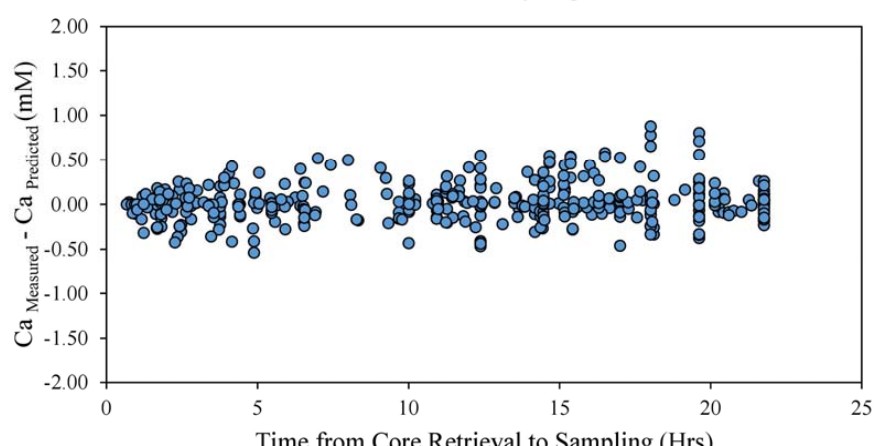





















**Figure 12.**

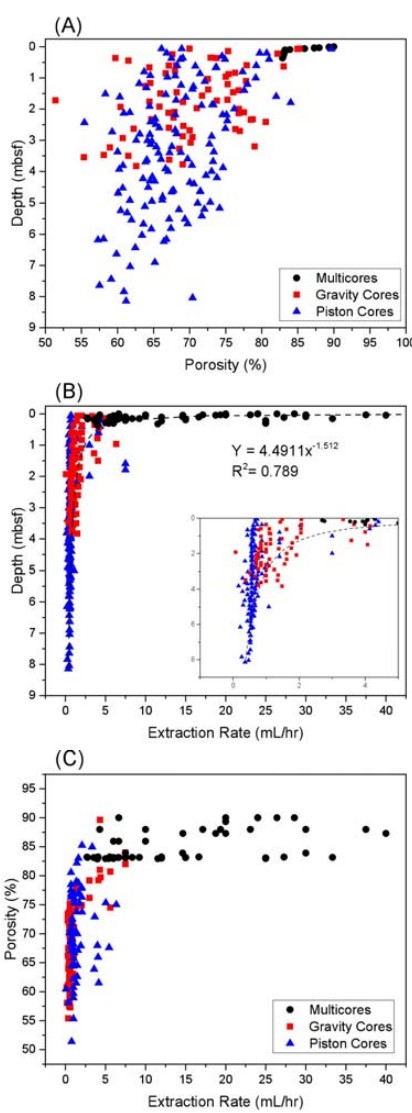











**Figure 13.**

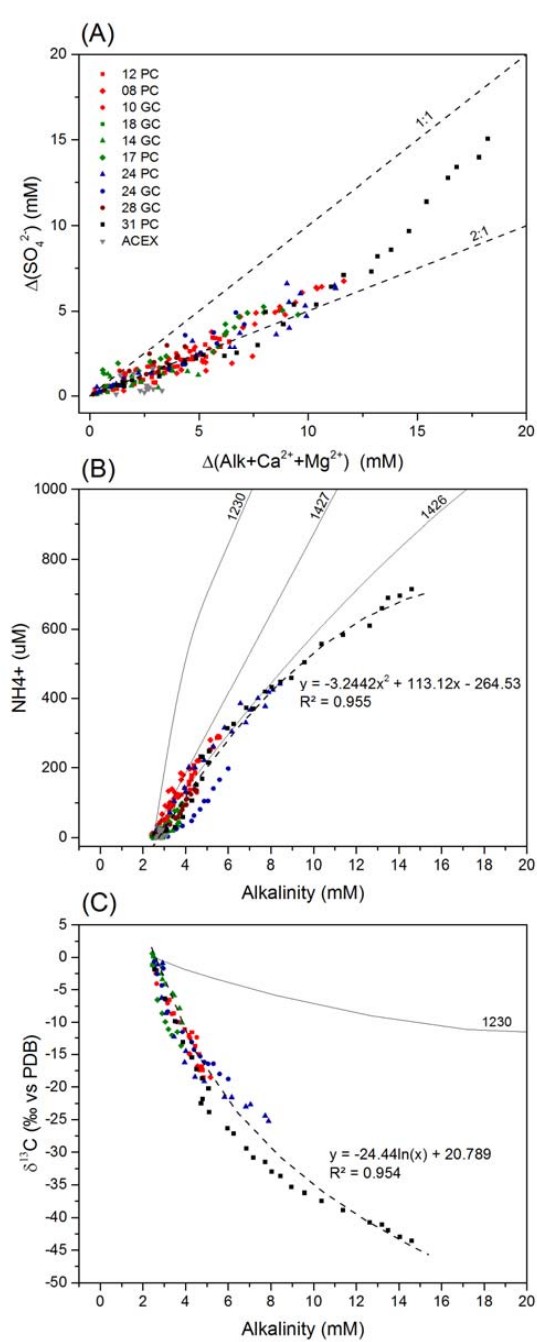




**Figure 14.**

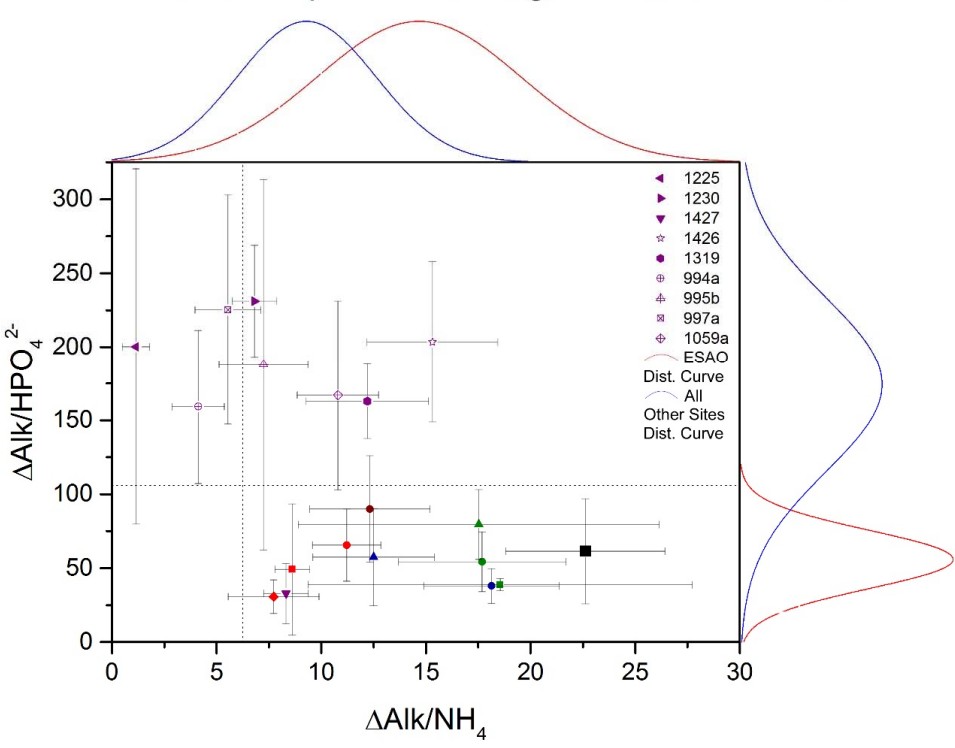






**Figure 15.**

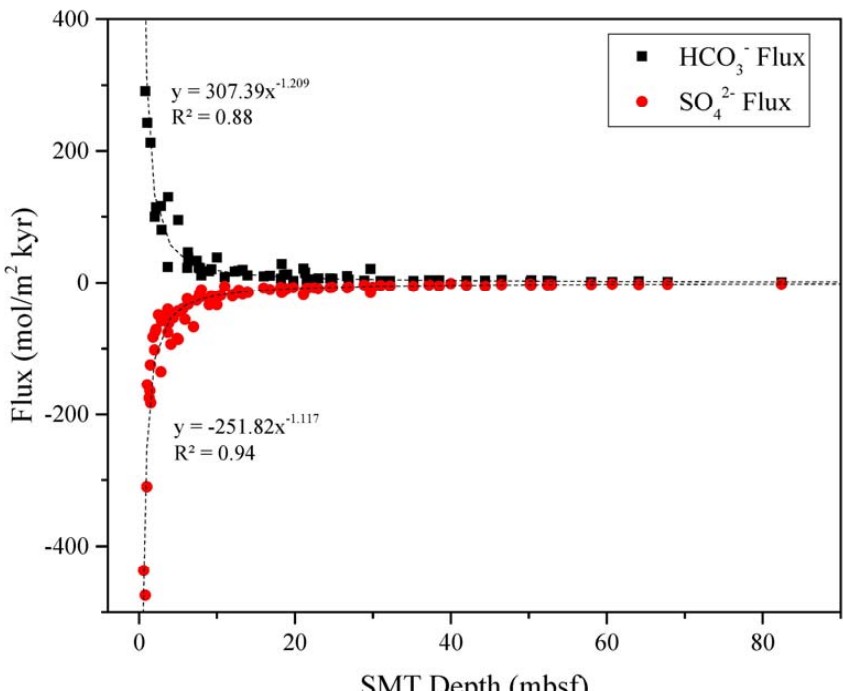















**Figure 16.**

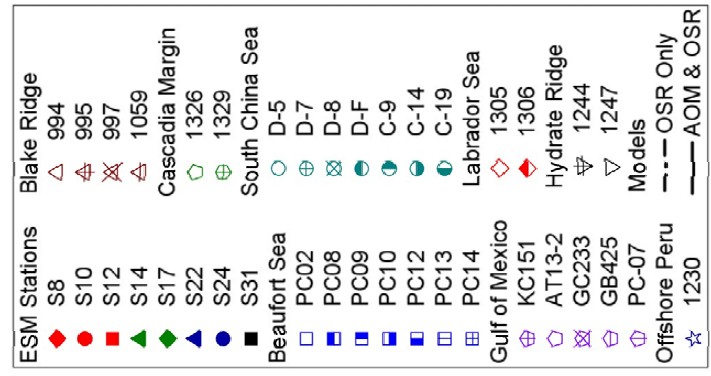

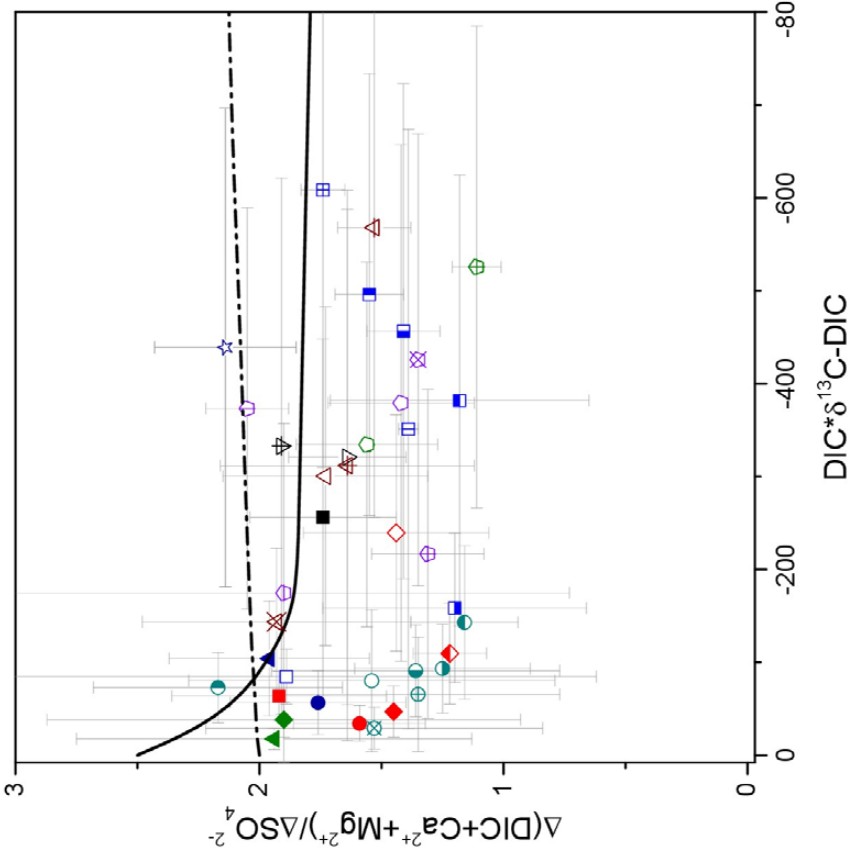




Table 1 Rhizon Efficacy

| Station | Location | | Water Depth | Core Type | Core Length | Total Time from Coring to Sampling | | Rhizon Extraction Time | |
|---|---|---|---|---|---|---|---|---|---|
| | Latitude | Longitude | (m) | | (m) | Average (hrs) | Range (hrs) | Average (hrs) | Range (hrs) |
| 8 | 75° 09' 11.4912" N | 179° 52' 23.0952" E | 524 | MC | 0.16 | 0.92 | (0.57 - 1.15) | 0.47 | (0.08 - 1.21) |
| 9 | 75° 03' 24.2166" N | 179° 49' 13.497" W | 446 | MC | 0.29 | 1.13 | (0.87 - 1.26) | 0.91 | (0.19 - 1.47) |
| 13 | 76° 11' 10.7772" N | 179° 16' 42.102" W | 1118 | MC | 0.17 | 1.40 | (1.20 - 1.68) | 0.40 | (0.20 - 0.68) |
| 14 | 76° 21' 10.3926" N | 176° 27' 39.9954" E | 733 | MC | 0.20 | 2.51 | (0.83 - 4.42) | 1.76 | (0.08 - 3.67) |
| 16 | 76° 30' 43.2540" N | 176° 37' 55.1670" E | 1023 | MC | 0.21 | 2.81 | (1.87 - 3.70) | 1.44 | (0.51 - 2.33) |
| 18 | 76° 24' 32.8680" N | 173° 52' 44.9178" E | 349 | MC | 0.33 | 2.11 | (1.32 - 2.68) | 1.21 | (0.42 - 1.78) |
| 21 | 77° 34' 56.4918" N | 163° 17' 36.7326" E | 159 | MC | 0.24 | 4.02 | (3.45 - 4.37) | 0.73 | (0.17 - 1.08) |
| 22 | 78° 13' 25.9572" N | 164° 25' 36.3612" E | 367 | MC | 0.32 | 0.73 | (0.68 - 0.75) | 0.38 | (0.33 - 0.41) |
| 23 | 78° 39' 51.7206" N | 165° 01' 57.8352" E | 522 | MC | 0.32 | 1.48 | (0.67 - 1.93) | 1.24 | (0.35 - 1.85) |
| 24 | 78° 48' 00.1080" N | 165° 22' 55.8552" E | 982 | MC | 0.32 | 1.79 | (0.85 - 2.18) | 1.34 | (0.52 - 1.67) |
| 25 | 79° 13' 34.6362" N | 152° 40' 32.2284" E | 101 | MC | 0.25 | 0.97 | (0.83 - 1.17) | 0.63 | (0.51 - 0.83) |
| 26 | 79° 44' 31.6782" N | 154° 23' 20.3244" E | 378 | MC | 0.37 | 1.83 | (0.80 - 2.80) | 1.53 | (0.50 - 2.51) |
| 27 | 79° 39' 52.6824" N | 154° 07' 34.5324" E | 276 | MC | 0.26 | 1.65 | (1.35 - 2.18) | 1.30 | (1.01 - 1.83) |
| 28 | 79° 55' 10.3384" N | 154° 21' 12.7578" E | 1145 | MC | 0.27 | 1.53 | (0.85 - 2.27) | 1.10 | (0.42 - 1.83) |
| 29 | 81° 20' 33.9750" N | 141° 46' 31.6668" E | 899 | MC | 0.23 | 1.55 | (1.05 - 2.05) | 1.17 | (0.67 - 1.67) |
| 31 | 79° 55' 13.4070" N | 143° 09' 53.0994" E | 1157 | MC | 0.39 | 2.38 | (1.80 - 3.38) | 2.08 | (1.50 - 3.08) |
| 32 | 85° 08' 28.2582" N | 151° 35' 24.5220" E | 837 | MC | 0.33 | 4.38 | (2.38 - 5.05) | 3.33 | (1.33 - 4.01) |
| Multicore Average and Range | | | | | 0.27 | 1.95 | (0.57 - 5.05) | 1.24 | (0.08 - 4.01) |
| 8 | 75° 08' 06.3342" N | 179° 51' 05.9004" E | 515 | PC | 6.42 | 18.56 | (4.81 - 20.85) | 15.22 | (1.78 - 17.5) |
| 10 | 75° 30' 12.6462" N | 179° 05' 59.265" W | 1000 | GC | 3.99 | 11.01 | (3.38 - 12.38) | 10.31 | (1.83 - 11.83) |
| 12 | 75° 00' 57.3114" N | 179° 45' 09.9900" E | 384 | PC | 8.43 | 21.44 | (20.27 - 21.77) | 17.95 | (16.42 - 19.15) |
| 14 | 76° 22' 04.9146" N | 176° 25' 56.9670" E | 737 | GC | 2.75 | 14.58 | (6.55 - 17.11) | 9.00 | (5.22 - 10.01) |
| 17 | 76° 27' 52.6248" N | 176° 43' 25.7628" E | 977 | PC | 6.37 | 19.62 | (18.12 - 19.62) | 16.58 | (15.08 - 16.58) |
| 18 | 76° 24' 41.7240" N | 173° 47' 17.6454" E | 351 | GC | 1.95 | 7.72 | (3.02 - 14.37) | 6.40 | (1.73 - 12.92) |
| 22 | 78° 13' 22.5336" N | 164° 27' 42.6306" E | 364 | PC | 6.45 | 19.79 | (13.5 - 25.17) | 17.71 | (11.42 - 23.08) |
| 24 | 78° 47' 48.9186" N | 165° 21' 59.5080" E | 964 | GC | 4.05 | 12.25 | (10.94 - 14.19) | 5.89 | (4.83 - 7.42) |
| 28 | 79° 55' 28.0302" N | 154° 23' 44.7180" E | 1143 | GC | 5.23 | 12.78 | (3.33 - 14.33) | 6.98 | (5.58 - 16.63) |
| 29 | 81° 17' 57.6816" N | 141° 46' 57.1794" E | 824 | GC | 4.66 | 17.43 | (9.25 - 18.02) | 13.38 | (4.72 - 14.12) |
| 31 | 79° 54' 53.4270" N | 143° 14' 00.4488" E | 1120 | PC | 8.07 | 11.95 | (3.90 - 18.07) | 9.25 | (1.33 - 15.50) |
| 33 | 84° 16' 29.5422" N | 148° 44' 07.1484" E | 886 | GC | 3.59 | 12.20 | (9.57 - 19.07) | 8.55 | (5.92 - 15.42) |
| 33 | 84° 16' 55.5368" N | 148° 38' 48.3102" E | 888 | PC | 6.24 | 11.06 | (7.55 - 17.88) | 9.48 | (5.82 - 16.25) |
| Gravity/Piston Core Average and Range | | | | | 5.25 | 14.65 | (3.02 - 25.17) | 11.28 | (1.33 - 23.08) |





Table 2 Rhizon Flow Rates

| Station | Flow Rate (mL/hr) | | | Flow Rate Decrease per meter |
|---|---|---|---|---|
| | Average | Min | Max | (mL/hr/m) |
| 8 | 25.28 | 8.33 | 37.50 | 243.06 |
| 9 | 11.36 | 5.16 | 26.82 | 59.21 |
| 13 | 23.68 | 14.63 | 33.33 | 70.36 |
| 14 | 8.31 | 0.55 | 24.00 | 130.30 |
| 16 | 8.31 | 0.55 | 24.00 | 79.37 |
| 18 | 10.77 | 3.93 | 26.40 | 70.23 |
| 21 | 13.00 | 2.77 | 40.00 | 201.71 |
| 22 | 26.82 | 25.00 | 30.00 | 18.52 |
| 23 | 13.52 | 5.41 | 28.57 | 77.22 |
| 24 | 9.51 | 6.00 | 19.35 | 49.46 |
| 25 | 16.24 | 12.00 | 20.00 | 36.36 |
| 26 | 8.85 | 4.00 | 20.00 | 49.23 |
| 27 | 8.09 | 5.45 | 10.00 | 21.65 |
| 28 | 11.80 | 5.45 | 24.00 | 74.18 |
| 29 | 10.36 | 6.00 | 16.50 | 58.33 |
| 31 | 5.16 | 3.24 | 6.67 | 10.07 |
| 32 | 5.21 | 3.75 | 11.25 | 23.44 |
| Multicore Average | 12.72 | 6.60 | 23.43 | 74.86 |
| 8 | 1.13 | 0.29 | 5.61 | 0.84 |
| 10 | 1.35 | 0.38 | 5.45 | 1.33 |
| 12 | 0.53 | 0.17 | 0.67 | 0.04 |
| 14 | 1.19 | 1.00 | 1.92 | 0.14 |
| 17 | 0.61 | 0.36 | 0.73 | 0.02 |
| 18 | 2.76 | 0.08 | 6.35 | 2.83 |
| 22 | 0.59 | 0.22 | 0.88 | 0.11 |
| 24 | 1.71 | 1.25 | 2.07 | 0.20 |
| 28 | 1.74 | 0.70 | 3.00 | 0.70 |
| 29 | 0.80 | 0.64 | 2.12 | 0.35 |
| 31 | 2.03 | 0.65 | 7.50 | 0.19 |
| 33 | 1.26 | 0.65 | 1.69 | 0.14 |
| 33 | 1.11 | 0.62 | 1.69 | 0.13 |
| Gravity/Piston Core Average | 1.29 | 0.54 | 3.05 | 0.54 |






Table 3 QA/QC Results

| Analysis | Sample Type | Number | Result |
|---|---|---|---|
| Alkalinity | Spiked | 15 | PE = 1.53% |
| Alkalinity | Duplicate | 8 | PD = 1.30% |
| $\delta^{13}$C-DIC | Seawater Standard | 2 | 0.23‰ and 0.32‰ |
| $\delta^{13}$C-DIC | Blind Field Duplicate | 4 | PD = 22.98% |
| $\delta^{13}$C-DIC | Field Blank | 1 | No Result |
| $\delta^{13}$C-DIC | Duplicate | 10 | PD = 14.70% |
| Metals | Spiked | 51 | RSD = 2.55% (Ba), 2.17% (Ca), 1.53% (Fe), 0.77% (Mg), 1.73% (Mn), 1.88% (S), and 1.42% (Sr) |
| Metals | Blind Field Duplicate | 11 | PD = 2.56% (Ba), 3.77% (Ca), 5.81% (Fe), 2.68% (Mg), 3.07% (Mn), 0.71% (S), and 3.79% (Sr) |
| Metals | Field Blank | 2 | BDL |
| Phosphate | VKI Standard | 2 | PE = 1.28% and 2.69% |
| Ammonia | VKI Standard | 2 | PE = 2.40% and 6.25% |

Notes:    PE = Percent Error

PD = Percent Difference

RSD = Relative Standard Deviation

BDL = Below Detection Limit













Table 4 - Reported and Calculated Fluxes

| Ocean | Location | Water Depth (m) | SMT Depth (mbsf) | $SO_4^{2-}$ Flux (mol/m²kyr) | Alkalinity Flux (mol/m²kyr) | $\delta^{13}C$ at SMT (‰) |
|---|---|---|---|---|---|---|
| Arctic | Beaufort Sea - Cape Halkett[a,b] | 280 | 1.06 | -154.8 | 242.6 | -21.5 |
| Arctic | Beaufort Sea - Cape Halkett[a,b] | 342 | 1.47 | -124.7 | 212.3 | -20.2 |
| Arctic | Beaufort Sea - Cape Halkett[a,b] | 1005 | 3.73 | -44.2 | 130.3 | -18.2 |
| Arctic | Beaufort Sea - Cape Halkett[a,b] | 1458 | 6.29 | -27.4 | 46.3 | -19.7 |
| Arctic | East Siberian Slope | 349 | 61 | -1.8 | 1.7 | -- |
| Arctic | East Siberian Slope | 367 | 25 | -6.9 | 6.3 | -- |
| Arctic | East Siberian Slope | 384 | 64 | -2.4 | 2.3 | -- |
| Arctic | East Siberian Slope | 524 | 35 | -5.6 | 2.8 | -- |
| Arctic | East Siberian Slope | 733 | 58 | -2.1 | 1.5 | -- |
| Arctic | East Siberian Slope | 977 | 58 | -2.1 | 1.6 | -- |
| Arctic | East Siberian Slope | 964 | 23 | -9.2 | 6.8 | -- |
| Arctic | East Siberian Slope | 1000 | 52 | -3.3 | 3.3 | -- |
| Arctic | East Siberian Slope | 1143 | 44 | -5.1 | 3.5 | -- |
| Arctic | East Siberian Slope | 1120 | 14 | -13.9 | 11.3 | -- |
| Atlantic | New Jersey Continental Slope[q,i] | 912 | 28.9 | -3.3 | 3.6‡ | -- |
| Atlantic | Blake Ridge[q,p] | 1293 | 50.3 | -3.4 | 3.8‡ | -- |
| Atlantic | Blake Ridge[q,p] | 1798 | 26.9 | -6.6 | 4.9‡ | -- |
| Atlantic | Blake Ridge[q,x] | 2567 | 42.0 | -3.8 | 3.5‡ | -- |
| Atlantic | Blake Ridge[q,x] | 2641 | 24.5 | -7.6 | 6.9‡ | -- |
| Atlantic | Blake Ridge[q,x] | 2777 | 21.7 | -8.3 | 5.4‡ | -- |
| Atlantic | Blake Ridge[q,x] | 2770 | 22.5 | -7.8 | 4.7‡ | -- |
| Atlantic | Blake Ridge[q,x] | 2798 | 21.5 | -8.7 | 4.4‡ | -- |
| Atlantic | Blake Ridge[q,p] | 2985 | 9.3 | -20.0 | 20.4‡ | -- |
| Atlantic | Blake Ridge[q,p] | 3481 | 12.3 | -17.1 | 17.0‡ | -- |
| Atlantic | Blake Ridge[q,p] | 4040 | 16.8 | -10.5 | 10.8‡ | -- |
| Atlantic | Gulf of Mexico - Keathley Canyon[w] | 1300 | 9 | -33‡ | 17‡ | -49.6 |
| Atlantic | Gulf of Mexico - Atwater Valley[w] | 1300 | 0.1 | -2901 | -- | -- |
| Atlantic | Gulf of Mexico - Atwater Valley[w] | 1300 | 0.1 | -2901 | -- | -- |
| Atlantic | Gulf of Mexico - Atwater Valley[w] | 1300 | 0.6 | -437 | -- | -- |
| Atlantic | Gulf of Mexico - Atwater Valley[w] | 1300 | 7 | -67 | -- | -46.3 |
| Atlantic | Amazon Fan[q,v,y] | 3191 | 37.2 | -3.2 | 4.1‡ | -39.8 |
| Atlantic | Amazon Fan[q,v,y] | 3474 | 6.2 | -24.6 | 22.7‡ | -47.5 |
| Atlantic | Amazon Fan[q,v,y] | 3704 | 3.7 | -40.3 | 24.3‡ | -49.6 |
| Atlantic | Western Africa[q,z] | 426 | 12.8 | -12.5 | 18.2‡ | -- |
| Atlantic | Western Africa[q,z] | 738 | 52.9 | -3.1 | 2.9‡ | -- |
| Atlantic | Western Africa[q,z] | 1280 | 21.3 | -12.0 | 15.6‡ | -19.8 |
| Atlantic | Western Africa[q,z] | 1402 | 18.3 | -14.9 | 28.3‡ | -- |





| Atlantic | Western Africa[q,z] | 1713 | 38.5 | -5.1 | 4.1[‡] | -- |
|---|---|---|---|---|---|---|
| Atlantic | Western Africa[q,z] | 2179 | 26.7 | -7.8 | 10.4[‡] | -- |
| Atlantic | Western Africa[q,z] | 2382 | 21.1 | -18.1 | 21.8[‡] | -- |
| Atlantic | Western Africa[q,z] | 2995 | 29.7 | -14.9 | 20.9[‡] | -- |
| Atlantic | Argentine Basin[l] | 1228 | 10.5 | -19.1 | -- | -- |
| Atlantic | Argentine Basin[l] | 1492 | 12 | -20.2 | -- | -- |
| Atlantic | Argentine Basin[l] | 1568 | 4.9 | -84.6 | -- | -- |
| Atlantic | Argentine Basin[l] | 1789 | 5.9 | -55.6 | -- | -- |
| Atlantic | Argentine Basin[l] | 3247 | 10 | -21.8 | -- | -- |
| Atlantic | Argentine Basin[l] | 3167 | 14 | -14.7 | -- | -- |
| Atlantic | Argentine Basin[l] | 3542 | 3.7 | -75.4 | -- | -- |
| Atlantic | Argentine Basin[l] | 3551 | 5.6 | -39.9 | -- | -- |
| Atlantic | Argentine Basin[l] | 3551 | 4.1 | -93.3 | -- | -- |
| Atlantic | Argentine Basin[l] | 3623 | 5 | -43.1 | -- | -- |
| Atlantic | Argentine Basin[l] | 4280 | 5.1 | -43.5 | -- | -- |
| Atlantic | Argentine Basin[l] | 4799 | 12 | -17.9 | -- | -- |
| Indian | Oman[q,1] | 591 | 50.2 | -2.2 | 1.1[‡] | -- |
| Indian | Oman[q,1] | 804 | 46.5 | -2.8 | 4.4[‡] | -- |
| Indian | Oman[q,1] | 1423 | 82.4 | -1.8 | 0.8[‡] | -- |
| Pacific | Bering Sea[p,2] | 1008 | 6.3 | -32.8 | 37.8 | -25.1 |
| Pacific | Cascadia[q,u,2] | 959 | 9.0 | -23.6 | -- | -23.8 |
| Pacific | Cascadia[q,u,2] | 1322 | 7.9 | -21.3 | -- | -30.8 |
| Pacific | Cascadia[q,u,2] | 1828 | 2.5 | -49.0 | -- | -33.9 |
| Pacific | Cascadia - Hydrate Ridge[o] | 834 | 8 | -10.9 | 11.3 | -19.6 |
| Pacific | Cascadia - Hydrate Ridge[o] | 850 | 7.65 | -22.3 | 23.2 | -30.2 |
| Pacific | Cascadia - Hydrate Ridge[o] | 871 | 7.4 | -26.6 | 33.4 | -24.9 |
| Pacific | Cascadia - Hydrate Ridge[g] | 896 | 7.8 | -16 | 22 | -22.5 |
| Pacific | Umitaka Spur[h] | 900 | 2.2 | -71 | 114 | -- |
| Pacific | Umitaka Spur[h] | 947 | 2.9 | -58 | 80 | -- |
| Pacific | Umitaka Spur[h] | 1034 | 2.0 | -102 | 100 | -- |
| Pacific | Japan Sea[s,4] | 901 | 10 | -33.6 | 38.4[‡] | -- |
| Pacific | California Margin[q,5] | 955 | 13.3 | -17.3 | 19.6[‡] | -- |
| Pacific | California Margin[q,5] | 1564 | 19.0 | -9.3 | 12.8[‡] | -- |
| Pacific | California Margin[q,5] | 1926 | 31.0 | -4.3 | 3.1[‡] | -- |
| Pacific | Nankai Trough[q,6] | 1741 | 32.2 | -4.9 | 3[‡] | -- |
| Pacific | Nankai Trough[s,6] | 2997 | 11.0 | -5.6 | 8.7[‡] | -- |
| Pacific | Nankai Trough[q,6] | 3020 | 18.2 | -7.0 | 6.4[‡] | -- |
| Pacific | Santa Barbara[k] | 587 | 1.3 | -175.2 | -- | -- |
| Pacific | Soledad[k] | 542 | 1 | -310.3 | -- | -- |



| | | | | | | |
|---|---|---|---|---|---|---|
| Pacific | Pescadero[k] | 408 | 1.4 | -164.3 | -- | -- |
| Pacific | Magdalena[k] | 600 | 1.5 | -182.5 | -- | -- |
| Pacific | Alfonso[k] | 713 | 0.8 | -474.5 | -- | -- |
| Pacific | Costa Rica Margin[q,7] | 3306 | 16.0 | -8.1 | 9.6[‡] | -- |
| Pacific | Costa Rica Margin[q,7] | 4177 | 19.8 | -7.5 | 3.1[‡] | -- |
| Pacific | Costa Rica Margin[q,7] | 4311 | 18.6 | -12.3 | 12.4[‡] | -- |
| Pacific | Peru Margin[s,8] | 161 | 30 | -6.9 | -- | -- |
| Pacific | Peru Margin[t,9] | 427 | 40 | -1.2 | -- | -25.4 |
| Pacific | Peru Margin[t,9] | 5086 | 9 | -25.0 | -- | -13.2 |
| Pacific | Chilean Coast[c] | 586 | 5.55 | -22.9 | -- | -- |
| Pacific | Chilean Coast[c] | 723 | 0.33 | -362.0 | -- | -- |
| Pacific | Chilean Coast[c] | 980 | 2.92 | -45.3 | -- | -- |
| Pacific | Chilean Coast[c] | 768 | 10.11 | -13.3 | -- | -- |
| Pacific | New Zealand - Porangahau Ridge[f] | 1900-2150 | 12.8 | -11.4 | -- | -31.4 |
| Pacific | New Zealand - Porangahau Ridge[f] | 1900-2150 | 4.4 | -53.3 | -- | -31.6 |
| Pacific | New Zealand - Porangahau Ridge[f] | 1900-2150 | 3.6 | -50.5 | -- | -31.4 |
| Pacific | New Zealand - Porangahau Ridge[f] | 1900-2150 | 2.1 | -74.2 | -- | -33.4 |
| Pacific | New Zealand - Porangahau Ridge[f] | 1900-2150 | 3.8 | -61.5 | -- | -35.0 |
| Pacific | New Zealand - Porangahau Ridge[f] | 1900-2150 | 1.8 | -82.6 | -- | -48.8 |
| Pacific | New Zealand - Hikurangi[b,d] | 350 | 39.5 | 5[‡] | 7.3[‡] | -- |
| Pacific | New Zealand - Hikurangi[b,d] | 332 | 12.9 | 19.3[‡] | 13.6[‡] | -- |
| Pacific | New Zealand - Hikurangi[b,d] | 98 | 0.87 | 192.1[‡] | 160.9[‡] | -- |
| Pacific | New Zealand - Hikurangi[b,d] | 285 | 3.64 | 65.2[‡] | 59.6[‡] | -- |
| Southern Ocean | Antarctic - Cumberland Bay[n] | 237 | 5.03 | -86 | 95 | -25.4 |
| Southern Ocean | Antarctic - Cumberland Bay[n] | 260 | 0.80 | -539 | 291 | -23.5 |
| Southern Ocean | Antarctic - Cumberland Bay[n] | 275 | 2.80 | -135 | 116 | -15.5 |











Appendix 1.
These models follow Chatterjee et al., (2011) and Malinverno and Pohlman, (2011). The first
model assumes the only sulfate reduction taking place is through OSR. Carbon fractionation
through OSR is set at $\alpha = 1.01$ from zero at the seafloor. Sulfate is completely consumed at 10m
with a constant porosity of 70%. Diffusion is calculated by Equation 1 where the diffusivity in
sediment is (Iverson and Jørgensen, 1993)

$$D_S = \frac{D_O}{(1 + n(1 - \varphi))} \tag{9}$$

Diffusion in seawater ($D_o$) for sulfate is $0.56*10^{-5}$ cm$^2$/s (Iverson and Jørgensen, 1993) and
$0.60*10^{-5}$ cm$^2$/s for bicarbonate (Li and Gregory, 1974). The saturation factor ($n$) was assumed to
be 3 for clay/silt sediments, and the sedimentation rate was set at an arbitrary 25 cm/kyr. The
conceptual framework for the second model is set to include both OSR and AOM. A SMT is set
at five meters below the seafloor while sulfate reduction takes place at the surface. Carbon
fractionation through AOM is set at $\alpha = 1.0175$. Both downward diffusing sulfate and upward
methane fluxes are set at 120 mol/m$^2$-kyr. The $\delta^{13}$C-CH$_4$ of the upward diffusing methane is set
at -70‰, but an additional flux of DIC set at 20‰ is added from below the SMT.