# Peer review of "Biogeosciences Discuss., doi:10.5194/bg-2016-308, 2016 Manuscript under review for journal Biogeosciences Published: 9 August 2016 © Author(s) 2016. CC-BY 3.0 License."

_Biogeosciences, 2016_

## Referee Comment (RC1) · Anonymous Referee #1 · 13 Sep 2016

Methane (CH4) fluxes on the East Siberian Margin (ESM), as the authors call the area, has indeed attracted much attention from the scientific community lately and caused a storm of debate. I agree that many aspects of this topic need more study because the topic itself is rather novel and has only been under investigation for a little over a decade. I also share the authors' opinion that our understanding of some processes and mechanisms controlling CH4 cycling in this area requires a great deal of improvement. I believe that the authors of this manuscript could have contributed to this topic with new data; they clearly pointed out in the text that this topic was a primary goal for the SWERUS-3 expedition onboard IB Oden which was accomplished in 2014. However, this ms rather demonstrates that the current state of knowledge

of pore water biogeochemistry in particular areas of the ocean is very incomplete; a great deal of effort will be required in order to improve our understanding of the relationship between sulfur and carbon cycling in the Arctic. The authors of the ms come to the following conclusions: 1) Based on interpretation of the pore water profiles, they found no evidence for upwardly diffusing CH4. 2) Based on these data, they strongly suggested that gas hydrates do not occur on the slopes of the ESM. 3) They claimed that previous investigators who suggested that hydrate deposits exist in the Arctic shelf/slope based on results of their investigations were simply wrong.

First of all, I do not understand why, when reporting low CH4 concentrations and the relationship between CH4 and sulfate dynamics in the pore water, the authors did not measure the concentrations of either parameter. Is it not logical to measure CH4 and sulfate in pore water if one is going to report "low methane concentrations in the sediments"? These are rather routine measurements. The authors referred to other researchers in their ms to present supportive arguments, but none of these referenced studies avoided taking measurements. In addition, the authors of this ms speculate about the particulate organic carbon (POC) and OC content of sediment, but did not measure either parameter. OC content of sediments should be reported as a number of different carbon stocks, not just POC. Second, I do not understand why the presence or absence of CH4, either in the sediments or in the water column in this area, should be necessarily connected to the existence or non-existence of hydrates. Are hydrates the only possible source of CH4 in the Arctic shelf/slope? I believe not; hydrates could be only a tiny fraction of the source, because the hydrate stability zone (HSZ) created by P/T conditions could compose only a small fraction of the sedimentary drape (a few hundred meters), while the sedimentary drape could be a few kilometers thick. Third, the purpose of this massive manuscript is not clear to me. This paper is flooded with equations and details devoted to methods, but mathematics, first of all should be applicable; then, the accuracy of mathematics does not aid in interpreting the inconclusive data. Below are my comments on some aspects of this ms. A more detailed look would be as long as the ms itself, because nearly every page of this ms would benefit

from clarification. The methodology chosen by the authors of this ms and their level of understanding of the processes they were trying to investigate are my greatest concern. Biogeochemists working in the marine ecosystems have already gained some understanding of the fact that biogeochemical processes associated with diagenetic transformation of organic matter under anaerobic conditions in marine sediments are very complex microbe-mediated processes. These processes involve microorganisms from various physiological groups: aerobic and anaerobic saprophylic and cellulose-degrading bacteria, sulfate reducers, methanogens, denitrifiers, and methylotrophs. Transformation of organic matter is a multi-stage process: primary anaerobes decompose polymeric compounds to monomers, which, in turn, serve as a substrate for fermentation agents and gas-producing bacteria. A general conclusion is that the major fraction of OC preserved in the sediments is oxidized to $CO_2$ by the sulfate-reducing bacteria (SRB) and that 2 moles of OC are oxidized for every mole of sulfate reduced: $4H(CH_2)_n COO^- + (3n + 1)SO_4^{2-} + H_2O \rightarrow (4n + 4)HCO_3^- + (3n + 1)HS^- + OH^- + nH^+$. When acetate is oxidized completely, the atomic ratio of OC oxidized to sulfate-S reduced is 2 : 1. However, as 'n' increases, the C: S ratio changes; the ratio between the reactants could be different because it depends on the varying nature of the organic matter (Lerman 1982). This is because most of the photosynthate is not immediately available for oxidation; only the low molecular weight (LMW) fraction of dissolved OC (DOC) is rapidly oxidized by SRB, while the high molecular weight (HMW) fraction of POC, which usually increases with depth, is refractory. There are severe restrictions on microbial activity other than substrate availability, including that SR as a biotic process may be more strongly coupled to mineralogy (Ivanov et al., 1989). The knowledge that has been accumulated by scientists so far is very limited and only applicable to those particular ecosystems which were investigated beyond the Arctic. The most reliable method to trace the course of sulfate reduction in sediments uses radioactive sulfate (35S). By the use of this method it was shown that most reduced 35S-sulfate was in pyrite and organic sulfur (Lein et al., 1982). The relationship between sulfur and carbon cycling in the Arctic marine systems is even more complicated, because

the relationships between the sites of primary production and the sites to which organic matter is translocated and deposited, including organic matter delivered to the shelf/slope from surrounding land, are difficult to establish both qualitatively and quantitatively. A recently published review of CH4 emissions from the seafloor in the Arctic Ocean underscored that the role of SRB in the anaerobic oxidation of methane (AOM) is unclear and the ecology of AOM communities, particularly for high-latitude environments, is not well understood. For that reason, predicting CH4 fluxes, especially those related to hydrate dissociation, remains highly speculative (James et al., 2016). This is because CH4 is transported within the sediments in two different ways: as a dissolved phase (by diffusion or advection) or as free gas (ebullition). Free gas is inaccessible to microbes, which depend on a diffusive transmembrane gas transport. This means that release of free gas through the sediments might not leave any traces in the pore water (see Fig.5 in James et al., 2016). Moreover, recently published observational data show that in the Arctic environment, for example in the Alaskan Beaufort Sea continental margin sediments, substantial (30-500 $\mu$M) concentrations of sulfate can remain below the sulfate-methane transition zone (SMTZ) although mass balance cannot explain the source of sulfate below the SMTZ. In addition, sulfate reduction and anaerobic oxidation of CH4 can occur throughout the methanogenic zone. Experimental data indicated decoupling of sulfate reduction and AOM and competition between sulfate reducers and methanogens for substrates, suggesting that the classical redox cascade of electron acceptor utilization based on Gibbs energy yields does not always hold even in diffusion-dominated systems (Treude et al., 2014). Although they vigorously referred to Nauhaus et al. (2002) as a proxy-establishing experiment, the authors did not give this work any critical assessment. If they had done so, they would have definitely questioned the claim that methanotrophic communities associated with SRB oxidize CH4 anaerobically in a 1:1 ratio to sulfate reduction. How that could be possible if the reported 4-5-fold increase in H2S production (accumulated over 80 days!) was accompanied by an increase in CH4 concentration of 3 orders of magnitude (from 0.01 to 15.8 mM)? Besides, rates of SR were so small (0.5-3.0 $\mu$M/d-1) compared to

the concentrations of sulfate (103-1.55 $\mu$ÐIJ) that the question arises: How could this little change be reliably measured (without using the 35S method, which they did not) and related to AOM? Not to mention that this effect has no applicability to the Arctic Ocean. Another concern is this: How representative of the area are these data? Only four short transects consisting of 16 stations are presented; each transect is based on data from 2-6 stations. Data from only 2-4 stations represent all core depths. Core lengths vary from 1.95 to 8.43 m (mean length 5.25 m). Eight of the 16 stations are only represented by the very uppermost layers (from 0.16 to 0.39 m) of sediment collected by the multi-corer. These shortest parts are the most valuable as they represent the least disturbed environment, but they are too short to constitute any sort of conclusive data regarding CH4 cycling in the sediments. I can only guess at how the authors succeeded in dividing these tiny cores into numerous parts, each 0.2-0.3 m in length, and accumulated enough data to compare these cores with one of two idealistic schemes to characterize the specific dynamics of processes occurring over a sediment depth of 100 m (Fig.1). Data obtained by other types of sampling (piston/gravity coring) should be treated and interpreted very cautiously as the cores are not only severely disturbed during the coring process, but also chemically altered as they are extracted from the sea floor and lifted onto the ship. Finally, the authors plotted water concentration profiles along each transect collectively (!) using colors and symbol types which make it virtually impossible to distinguish between these symbols, making interpretation of the data sets very difficult. From this, it follows that the authors assumed complete uniformity of processes occurring not only in the observed settings located tens of kilometers apart from each other, but also over the entire slope area! This is despite the fact that CH4 fluxes on the East Siberian Arctic Shelf (ESAS), which could be associated with CH4 releases from decaying hydrates, have been reported to vary by orders of magnitude within much smaller scales (Shakhova et al., 2015). I see a clear discrepancy between the basic assumptions made by the authors and the methodology used to test these assumptions. The authors assumed CH4 was being released from destabilizing hydrates, most likely via bubbles and the convective flow of geofluids. Despite that,

all equations used for estimates refer to the diffusive transport of CH4 and other substances in the sediments. This is understandable; they used what was available. The problem is that the mathematics associated with diffusive transport cannot be used to describe the release of free gas from decaying hydrates. When assuming CH4 release from gas hydrates, one should realize that hydrates convert to free gas; the released gas travels upward much faster than diffusion occurs, through very efficient gas migration paths (chimneys etc.). In most cases, ascending CH4 can avoid oxidation in a few ways. 1) Because free gas resulting from hydrate decay is over pressured, it builds up a gas front; this disturbs sediment layering, creating the characteristic marks of gas release (pockmarks etc.). 2) Only CH4 dissolved in pore water is reachable by microbial communities; CH4 released as free gas (ebullition) is not consumable by microbes. 3) AOM rates are only remarkable as compared to rates of modern methanogenesis, because all synergetic processes should be energetically efficient for all members of the microbial community, including SRB, methanogens and methanotrophs, etc. Finally, the authors of the ms used three assumptions to explain their findings. Their first assumption is that bottom seawater on the slope north of Siberia is warming, leading to hydrate destabilization. There are no reports of increased bottom water temperatures along the slope of the Arctic during either the last glacial cycle (Cronin et al., 2012) or the Holocene (Biastoch et al., 2011; Dmitrenko et al., 2011; James et al., 2016). All papers published so far project the response of the hydrate inventory to possible future climate change in the Arctic. The paper of Stranne et al., (2016) the authors refer to assumes a linear rise in ocean bottom water temperatures of 3°C over the coming 100 years. This speculative warming of the Arctic is intentionally set higher than in other studies (<2°C by Biastoch et al., 2011; <1°C by Kretschmer et al., 2015) while modeling assumptions contradict the existing hydrological data (Biastoch et al., 2011; Dmitrenko et al., 2011; James et al., 2016). Their second assumption is the quintessential statement that "Implicit of this finding is that sediments sequences along the ESM lack gas hydrates" following the authors' speculations about why predictions of hydrates on the ESM are so markedly wrong. The authors then suggest that: 1) the significant sea-ice concentration on the ESM diminishes net primary production (NPP); 2) the extremely broad continental shelf prevents accumulation of terrestrial organic-rich sediments; and 3) sediment accumulation is highly variable, so organic matter can be consumed during intervals of low deposition. None of these explanations is true. It was recently shown that the total OC (TOC) content in the ESAS/ESM sediments measured along the transect spanning more than 800 km from the Lena River mouth to the shelf (2000–3000 m water depth) varied between ∼2 % at shallow water depths and 0.8% in deeper water (Bröder et al., 2016). In addition, TOC values and general patterns, which reflect fractions of terrigenous OC reaching the slope (based on biomarkers), were within the same range as those measured for the North American Arctic margin (Stein and Fahl, 2000, 2004; Goni et al., 2013). For comparison, an average value for the continental slope of the Gulf of Mexico, where large storage of CH4 hydrates has been proven to exist, is 0.8% ±0.2 (Gulf of Mexico Hydrate Research Consortium). Moreover, according to Arrigo and van Dijken (2011), the total annual NPP over the Arctic Ocean exhibited a statistically significant increase by 20% between 1998 and 2009, due mainly to increases in both the extent of open water (+27%) and the duration of the open water season (+45 days). Most importantly, increases in NPP over the 12 year study period were largest in the eastern Arctic Ocean, most notably in the Siberian (+135%) sector. It is interesting that the authors themselves confirmed that: 1) environmental conditions on the ESM are highly conducive for gas hydrates; 2) hydrate occurrence in the other areas of the Arctic, where hydrates were predicted, was confirmed by hydrate recovery; and 3) all the models developed by generations of geologists to predict hydrates in the Arctic used the same assumptions. If the authors agree that these statements are true, they failed to be critical of their own work, which is based on a handful of inconclusive data obtained on a single expedition, groundless methodology, and a few erroneous assumptions. Instead of casting doubt on the results of others, I would suggest that the authors question their own results and make a greater effort to accumulate clear, interpretable data. I believe I have made it quite clear that there is a huge discrepancy between the results presented by the authors

and the far-reaching conclusions they are trying to support with these data. I see no way to support publication of this manuscript in its current state.

---

## Author Comment (AC1) · 22 Sep 2016

We thank the referee for the time and energy to write a lengthy review. However, while some comments seem reasonable, we find the overall review largely unfair and non-constructive. In Sweden, a well-known adage is "som fan läser bibeln", which describes how someone reads and misinterprets text so as to fit his/her views or intentions. We state this because, for the most part, the review is a string of comments that are either not germane to the primary subject of the manuscript or misguided in thought.

We note that there is no criticism on the importance of the topic, no criticism on the writing, no criticism on the data generated, and no criticism on the novelty of the work. Instead the criticisms, in a general sense, are:

[Figure]

(1) The manuscript does not contain additional data, even though it is one of the most comprehensive pore water chemistry data sets within a single effort generated to date in any region, let alone from a previously virtually unexplored section of the Arctic Ocean;

(2) The pore water results cannot be used to understand methane abundance in shallow sediment, although numerous papers in multiple regions, including in the Arctic offshore Alaska, convincingly demonstrate the opposite (eg. Borowski et al., 1996; Jørgensen et al., 2001; Torres and Kastner, 2009; Treude et al., 2014; etc.)

(3) The primary interpretation and conclusion conflicts with previous speculations in the region, although no pertinent information to the problem exists beyond the current work.

We also note that at least one criticism comes from "out of nowhere." Page C4 the reviewer berates the authors for "vigorous" referencing of a study (Nauhaus et al., 2002) that we neither cite nor discuss in the MS.

We thus suggest that such a review arrives because the results and interpretations disagree with pre-conceived but wholly unconstrained concepts concerning methane on the continental SLOPE north of Siberia. We actually chased the research, in a very similar style to that done offshore Alaska, fully expecting strong pore water evidence for high methane concentrations across this region. However, the data absolutely do not support this.

We elaborate on the above with point-by-point responses below. We can and will correct certain portions of the manuscript as necessary. In the end, however, we largely disagree with the contents and tone of this review, and trust that the editors will also realize these issues.

We welcome additional clarifications from the referee, should this person wish to expound. We put considerable time and energy into generating the data and writing the

manuscript, and stand by the main body of work, including the general interpretations.

Sincerely,

Clint Miller, and co-authors

Referee Comments (#) with Direct Responses (**).

**However, this ms rather demonstrates that the current state of knowledge of pore water biogeochemistry in particular areas of the ocean is very incomplete; a great deal of effort will be required in order to improve our understanding of the relationship between sulfur and carbon cycling in the Arctic.**

** We do not follow this comment. We strongly suggest that the community knows a great deal about pore water chemistry, especially in regions that contain gas hydrate (Borowski et al., 2000; Torres et al., 2004; Treude et al., 2005; Dickens and Snyder, 2009; Coffin et al., 2013) albeit some of the details remain incomplete. This is stressed in the MS.

**The authors of the ms come to the following conclusions: 1) Based on interpretation of the pore water profiles, they found no evidence for upwardly diffusing CH4. 2) Based on these data, they strongly suggested that gas hydrates do not occur on the slopes of the ESM. 3) They claimed that previous investigators who suggested that hydrate deposits exist in the Arctic shelf/slope based on results of their investigations were simply wrong.**

** This can and should be clarified further in the text. We strongly suggest that WIDESPREAD gas hydrates do not occur on the slopes of the ESM as speculated by previous papers. Here, it is important to stress, as discussed in the manuscript, that prior to our work, there is essentially no information on the topic from the studied area, and all has been conjectural. Our results and interpretations DO NOT CONFLICT with any previous data or direct results from the region that we are aware of.

**First of all, I do not understand why, when reporting low CH4 concentrations and**

the relationship between CH4 and sulfate dynamics in the pore water, the authors did not measure the concentrations of either parameter. Is it not logical to measure CH4 and sulfate in pore water if one is going to report "low methane concentrations in the sediments"? These are rather routine measurements.

** Not all measurements could be generated given the limitations of the expedition and subsequent funding. However, we did measure the S concentrations of pore water, and know these are representative of sulfate, because we checked for hydrogen sulfide as well as measured dissolved barium. Moreover, as the referee almost assuredly realizes, there are problems with generating quantitative dissolved methane profiles in marine sediment because of degassing associated with changes in pressure and temperature.

** Here, it is absolutely crucial to realize that, as stressed in the text and at least to our knowledge, no region with significant methane in moderately shallow sediment (< 500 m) has high dissolved S/dissolved sulfate in pore waters near the seafloor, as well as other certain chemistry documented here. While not stated in the MS, this also includes localized areas of high advection and methane venting.

**The authors referred to other researchers in their ms to present supportive arguments, but none of these referenced studies avoided taking measurements.**

** In fact, a significant fraction of the research referenced was partially generated by one of the authors (Dickens). It should be noted that the measurements were not avoided in some means to hide information, but rather that we know how methane exists and cycles in many regions, and the most prudent means of tackling the problem at a first-order level over an immense area from an ice-breaker is to generate numerous detailed pore water profiles (Borowski et al., 1999; Snyder et al., 2007; Hu et al., 2015).

**In addition, the authors of this ms speculate about the particulate organic carbon (POC) and OC content of sediment, but did not measure either parameter. OC content of sediments should be reported as a number of different carbon stocks, not just POC.**

** This is an odd comment because, as abundantly clear from numerous studies (Borowski et al., 1996; Dickens, 2000; Hensen et al., 2003; Geprags et al., 2016), the abundance of methane in shallow sediment on continental slopes does not depend on the current supply of POC, but rather on the integrated input of POC over long time intervals (e.g., hundreds of thousands to million year time scales), which we cannot assess without drilling. We are not sure what the referee means by different stocks. Basically, the data could be generated, but it is irrelevant.

**I do not understand why the presence or absence of CH4, either in the sediments or in the water column in this area, should be necessarily connected to the existence or non-existence of hydrates. Are hydrates the only possible source of CH4 in the Arctic shelf/slope? I believe not; hydrates could be only a tiny fraction of the source, because the hydrate stability zone (HSZ) created by P/T conditions could compose only a small fraction of the sedimentary drape (a few hundred meters), while the sedimentary drape could be a few kilometers thick.**

** We do not understand this comment. We think that it belies some misunderstanding on how methane occurs on continental slopes in general, as well as misreading of the MS. First off, gas hydrates are not a source of methane, but one phase of methane in open and dynamic systems, where methane carbon can exist as dissolved gas, free gas, and gas hydrate.

** We cannot link shallow water chemistry profiles to methane abundance at truly deep depths, and we did not do so in the MS. However, truly deep methane cannot exist as gas hydrates, for reasons of P/T conditions.

** On the other hand, the presence of gas hydrate in the upper few hundreds of meters of sediment is absolutely related to total methane concentrations in pore space, which are linked to shallow sediment through diffusion or in some cases advection.

** We thought that these points were clear in the literature as well as in the MS, but could rephrase things to clarify with some guidance.
**Third, the purpose of this massive manuscript is not clear to me. This paper is flooded with equations and details devoted to methods, but mathematics, first of all should be applicable; then, the accuracy of mathematics does not aid in interpreting the inconclusive data.**

** We do not understand this comment. We thought the purpose was very clear: we know how pore water chemistry profiles look above gas hydrate systems at numerous locations around the world, and we know how to interpret them at a basic level; we generated such profiles in the region of interest; the pore water profiles do not conform to those at any region where significant methane occurs in shallow sediment nor our understanding as to why such profiles arise.

**The methodology chosen by the authors of this ms and their level of understanding of the processes they were trying to investigate are my greatest concern. Biogeochemists working in the marine ecosystems have already gained some understanding of the fact that biogeochemical processes associated with diagenetic transformation of organic matter under anaerobic conditions in marine sediments are very complex microbe-mediated processes. These processes involve microorganisms from various physiological groups: aerobic and anaerobic saprophylic and cellulose- degrading bacteria, sulfate reducers, methanogens, denitrifiers, and methylotrophs. Transformation of organic matter is a multi-stage process: primary anaerobes decom- pose polymeric compounds to monomers, which, in turn, serve as a substrate for fer- mentation agents and gas-producing bacteria. A general conclusion is that the major fraction of OC preserved in the sediments is oxidized to $CO_2$ by the sulfate-reducing bacteria (SRB) and that 2 moles of OC are oxidized for every mole of sulfate reduced: $4H(CH_2)_n COO^- + (3n + 1)SO_4^{2-} + H_2O \rightarrow (4n + 4)HCO_3^- + (3n + 1)HS^- + OH^- + nH^+$. When acetate is oxidized completely, the atomic ratio of OC oxidized to sulfate-S re- duced is 2 : 1. However, as 'n' increases, the C: S ratio changes; the ratio between the reactants could be different because it depends on the varying nature of the organic matter (Lerman 1982). This is because most of the photosynthate is not immediately available**

for oxidation; only the low molecular weight (LMW) fraction of dissolved OC (DOC) is rapidly oxidized by SRB, while the high molecular weight (HMW) fraction of POC, which usually increases with depth, is refractory. There are severe restrictions on microbial activity other than substrate availability, including that SR as a biotic process may be more strongly coupled to mineralogy (Ivanov et al., 1989). The knowledge that has been accumulated by scientists so far is very limited and only applicable to those particular ecosystems which were investigated beyond the Arctic.

\*\* We do not know how to respond to this comment, as it mostly does not pertain to our MS, and also belies faulty logic. At a root level, and as best as we can ascertain, the comment suggests that biogeochemical processes are so complex that the community cannot obtain overall net chemical reactions and flow of carbon from pore water chemistry. If we correctly understand the comment, we then return an obvious question: how and why can the community measure similar pore water chemistry profiles at myriad locations and see basic commonalities (e.g., the absence of sulfate above sites with the presence of significant methane below), irrespective of the specifics and microbiology involved?

**The most reliable method to trace the course of sulfate reduction in sediments uses radioactive sulfate (35S). By the use of this method it was shown that most reduced 35S-sulfate was in pyrite and organic sulfur (Lein et al., 1982). The relationship between sulfur and carbon cycling in the Arctic marine systems is even more complicated, because the relationships between the sites of primary production and the sites to which organic matter is translocated and deposited, including organic matter delivered to the shelf/slope from surrounding land, are difficult to establish both qualitatively and quantitatively.**

\*\* We do not know how to respond to this comment, as it mostly does not pertain to our MS, it begins with a statement for which we disagree, and it does not make sense in its entirety.

**A recently published review of CH4 emissions from the seafloor in the Arctic Ocean underscored that the role of SRB in the anaerobic oxidation of methane (AOM) is unclear and the ecology of AOM communities, particularly for high-latitude environments, is not well understood. For that reason, predicting CH4 fluxes, especially those related to hydrate dissociation, remains highly speculative (James et al., 2016). This is because CH4 is transported within the sediments in two different ways: as a dissolved phase (by diffusion or advection) or as free gas (ebullition). Free gas is inaccessible to microbes, which depend on a diffusive transmembrane gas transport. This means that release of free gas through the sediments might not leave any traces in the pore water (see Fig.5 in James et al., 2016).**

** We do not know how to respond to this comment, as it mostly does not pertain to our MS, but we know the topics very well.

** The referenced MS (James et al., 2016), which we were unaware of when submitting our MS but have read since, by no means conflicts with our interpretations. These authors clearly indicate that AOM is a dominant process above methane-charged systems at steady-state conditions, and should impact pore water S/sulfate gradients (e.g., the very Fig. 5 that the referee emphasizes).

** Far more crucially, we are not concerned in our MS as to how methane would escape the seafloor via ENHANCED gas hydrate dissociation in the future, but whether significant methane exists in shallow sediment on the SLOPE of the ESM in the first place (i.e., at present-day), especially in the form of gas hydrate.

** We entirely agree with the comment that predicting the fate of gas hydrate dissociation on this margin (or indeed, any margin) in the future (or past) is highly speculative, as it depends on several factors, as also stressed by, for example Dickens (EPSL, 2003) or Stranne et al. (G3, 2016). But our MS does not discuss this aspect.

** We absolutely disagree with the comment that passage of free gas through sediment does not leave traces in the pore water. The paper and figures by James et al.

(2006) by no means suggest this concept, and rightfully so. Pore waters in areas where methane advects from below at high rates, such as along faults and fractures (their Fig. 5), have truly different chemistry than seawater and anything in our results.

** All the above stated, we are more than happy to include and reference the paper by James et al. (2016).

**Moreover, recently published observational data show that in the Arctic environment, for example in the Alaskan Beaufort Sea continental margin sediments, substantial (30-500 $\mu$M) concentrations of sulfate can remain below the sulfate-methane transition zone (SMTZ) although mass balance cannot explain the source of sulfate below the SMTZ.**

** This is an odd comment. First, 30 to 500 $\mu$M (=0.03 to 0.5 mM) is not substantial, compared to the $\sim$28 mM in typical seawater. Second, there are at least three known reasons for this occurrence. (1) Pore water contamination, (2) hydrogen sulfide oxidation, and (3) barite dissolution. Third, mass balance always applies.

**In addition, sulfate reduction and anaerobic oxidation of CH4 can occur throughout the methanogenic zone. Experimental data indicated decoupling of sulfate reduction and AOM and competition between sulfate reducers and methanogens for substrates, suggesting that the classical redox cascade of electron acceptor utilization based on Gibbs energy yields does not always hold even in diffusion-dominated systems (Treude et al., 2014).**

** Yes, this may true at a detailed and microscopic level, as pointed out by numerous authors, but not at any macroscopic level, at least that we are aware of, excepting odd environments (e.g., brines). The true beauty of pore water chemistry in the deep-sea marine environment is the remarkable consistency of multiple constituents linked to an array of environments. To restate from above, and in the text, all pore waters in methane-charged systems on continental slopes that we are aware of have certain commonalities – none that are seen in any of the pore waters generated in this study.

**Although they vigorously referred to Nauhaus et al. (2002) as a proxy-establishing experiment, the authors did not give this work any critical assessment. If they had done so, they would have definitely questioned the claim that methanotrophic communities associated with SRB oxidize CH4 anaerobically in a 1:1 ratio to sulfate reduction. How that could be possible if the reported 4-5-fold increase in H2S production (accumulated over 80 days!) was accompanied by an increase in CH4 concentration of 3 orders of magnitude (from 0.01 to 15.8 mM)? Besides, rates of SR were so small (0.5-3.0 $\mu$M/d-1) compared to the concentrations of sulfate (103-1.55 $\mu$ÃŘIJ) that the question arises: How could this little change be reliably measured (without using the 35S method, which they did not) and related to AOM?**

** We have absolutely no clue from where this comment derives. We neither cite nor discuss the paper by Nauhaus et al. (2002). (Seriously, at this point and given previous comments, we are wondering if the referee even read our MS!) The nominal 1:1 ratio of methane and sulfate consumption across the SMT comes from numerous studies of pore water chemistry gradients at numerous locations once the chemistry gradients are placed into a flux domain (rather than simply concentration gradients). By no means does this argument hinge on the experimental results of Nauhaus et al. (2002), which incidentally claim such a 1:1 relationship. Should the referee have fault with the Nauhaus et al. (2002) paper, one might question the design of a laboratory experiment where the net process is examined over a very short time interval in a modified environment after collecting sediment from depth and under different P/T conditions.

**Not to mention that this effect has no applicability to the Arctic Ocean.**

** We are not sure how to address. The referee seems to have a view that physical chemistry and biochemistry in the Arctic Ocean are somehow special, so that basics and inferences gained from elsewhere around the world do not apply.

** Here, it is especially important to note the paper by Coffin et al. (2013), as already highlighted in our MS. These authors characterized pore water chemistry in short sediment cores above sequences with known gas hydrates along the shelf and slope of the Beaufort Sea (Arctic), very much as done in our MS. As predicted, they observed shallow SMTs indicative of a strong diffusive methane flux. As very much obvious in our work, their pore water profiles contrast with those from the slopes off northern Siberia.

**Another concern is this: How representative of the area are these data? Only four short transects consisting of 16 stations are presented; each transect is based on data from 2-6 stations. Data from only 2-4 stations represent all core depths. Core lengths vary from 1.95 to 8.43 m (mean length 5.25 m). Eight of the 16 stations are only represented by the very uppermost layers (from 0.16 to 0.39 m) of sediment collected by the multi-corer. These shortest parts are the most valuable as they represent the least disturbed environment, but they are too short to constitute any sort of conclusive data regarding CH4 cycling in the sediments. I can only guess at how the authors succeeded in dividing these tiny cores into numerous parts, each 0.2-0.3 m in length, and accumulated enough data to compare these cores with one of two idealistic schemes to characterize the specific dynamics of processes occurring over a sediment depth of 100 m (Fig.1). Data obtained by other types of sampling (piston/gravity coring) should be treated and interpreted very cautiously as the cores are not only severely disturbed during the coring process, but also chemically altered as they are extracted from the sea floor and lifted onto the ship.**

** This comment does not make sense. First, the fact that the pore water profiles give nice, detailed gradients in multiple species, demonstrably indicates that the cores have minimal disturbance. We can add core photos to our already long paper, if desired, to further emphasize this point.

** Second, the proposition that the uppermost part of a core is the least disturbed and most important to understanding processes is flat out wrong. This is because of the nature of coring, which tends to disturb (or in many cases not recover) the top few cms, and because of bioturbation and reoxidation.

** Third, the methodology for how the cores were sampled is detailed at length in the manuscript.

** Fourth, the link between shallow pore water chemistry and moderately deep methane abundance is well established, as we thought clearly articulated in the literature for almost 20 years and in the MS. Here, again, we stress the differences in pore water chemistry between the Beaufort Sea (Coffin et al., 2013) and the ESM. The pore waters of shallow sediment in the Beaufort Sea predictably support a high upward methane flux.

**Finally, the authors plotted water concentration profiles along each transect collectively (!) using colors and symbol types which make it virtually impossible to distinguish between these symbols, making interpretation of the data sets very difficult.**

** On this matter, we welcome commentary, because we remain unsure how else to express the huge data set. As explained in the text, it seemed to us somewhat overwhelming to plot the chemistry at every site independently, or alternatively every species analyzed at multiple sites independently.

**From this, it follows that the authors assumed complete uniformity of processes occurring not only in the observed settings located tens of kilometers apart from each other, but also over the entire slope area! This is despite the fact that CH4 fluxes on the East Siberian Arctic Shelf (ESAS), which could be associated with CH4 releases from decaying hydrates, have been reported to vary by orders of mag- nitude within much smaller scales (Shakhova et al., 2015).**

** We did not assume complete uniformity of processes. In fact, the MS goes into great detail explaining the range of processes that relate to the pore water chemistry – processes that have been well documented along many continental slopes.

**I see a clear discrepancy between the basic assumptions made by the authors and the methodology used to test these assumptions. The authors assumed CH4 was**

being released from destabilizing hydrates, most likely via bubbles and the convective flow of geofluids.

** We do not understand this comment. The referee has leapt well beyond anything discussed in the MS. As stated above, and clearly in the MS, we discuss the lack of evidence for methane in shallow sediment and, by inference, gas hydrates on the slopes north of Siberia. Our MS has little bearing on how gas hydrates would be destabilized and how methane would be released.

** However, this comment seemingly belies a misunderstanding as to how carbon and methane cycle in sediment on continental slopes. In moderately shallow (< 500 m) sediment sequences with gas hydrate, there should be, at steady-state conditions, gas hydrate formation, gas hydrate dissociation and gas hydrate dissolution all co-occurring (see, for example, Dickens, EPSL, 2003). The pore water gradients between the top occurrence of gas hydrate in sediment and the seafloor arise, at a basic level, because of gas hydrate formation and gas hydrate dissolution; more specifically, where methane concentration gradients intersect the 2-phase gas hydrate-dissolved gas equilibrium curve.

**Despite that, all equations used for estimates refer to the diffusive transport of CH4 and other sub- stances in the sediments. This is understandable; they used what was available. The problem is that the mathematics associated with diffusive transport cannot be used to describe the release of free gas from decaying hydrates. When assuming CH4 release from gas hydrates, one should realize that hydrates convert to free gas; the released gas travels upward much faster than diffusion occurs, through very efficient gas migration paths (chimneys etc.). In most cases, ascending CH4 can avoid oxidation in a few ways. 1) Because free gas resulting from hydrate decay is over pressured, it builds up a gas front; this disturbs sediment layering, creating the characteristic marks of gas re- lease (pockmarks etc.). 2) Only CH4 dissolved in pore water is reachable by microbial communities; CH4 released as free gas (ebullition) is not consumable by microbes. 3) AOM rates are only remarkable as compared to rates**

of modern methanogenesis, be- cause all synergetic processes should be energetically efficient for all members of the microbial community, including SRB, methanogens and methanotrophs, etc.

** As noted above, this belies a wholesale misunderstanding as to how carbon and methane cycle in sediment on continental slopes. In most locations with gas hydrate, the vast majority of methane generated in the sediment ultimately (i.e., long time scales) escapes back to the ocean through diffusion. The assumption that methane-carbon above gas hydrates only returns to the ocean as free gas is entirely incorrect.

**Finally, the authors of the ms used three assumptions to explain their findings. Their first assumption is that bottom seawater on the slope north of Siberia is warming, leading to hydrate destabilization. There are no reports of increased bottom water temperatures along the slope of the Arctic during either the last glacial cycle (Cronin et al., 2012) or the Holocene (Biastoch et al., 2011; Dmitrenko et al., 2011; James et al., 2016). All papers published so far project the response of the hydrate inventory to possible future climate change in the Arctic.**

**This comment is false. Nowhere in our MS do we assume that bottom seawater is warming. Rather, we point out that the region is of great interest because bottom water might warm in the future.

**The paper of Stranne et al., (2016) the authors refer to assumes a linear rise in assumes a linear rise in ocean bottom water temperatures of 3◦C over the coming 100 years. This speculative warming of the Arctic is intentionally set higher than in other studies (<2◦ C by Biastoch et al., 2011; <1◦ C by Kretschmer et al., 2015) while modeling assumptions contradict the existing hydrological data (Biastoch et al., 2011; Dmitrenko et al., 2011; James et al., 2016).**

** Following from our commentary above, none of this is assumed in current MS.

**Their second assumption is the quintessential statement that "Implicit of this finding**

is that sediments sequences along the ESM lack gas hydrates" following the authors' speculations about why predictions of hydrates on the ESM are so markedly wrong.

** This is not an assumption, but rather a direct consequence of our results. The pore water chemistry profiles strongly indicate the lack of significant methane concentrations in the upper few hundred meters of sediment; given P/T conditions for gas hydrate, this absolutely implies an absence of gas hydrate.

** We can reword this if needed.

**The authors then suggest that: 1) the significant sea-ice concentration on the ESM diminishes net primary production (NPP); 2) the extremely broad continental shelf prevents accumulation of terrestrial organic-rich sediments; and 3) sediment accumulation is highly variable, so organic matter can be consumed during intervals of low deposition. None of these explanations is true. It was recently shown that the total OC (TOC) content in the ESAS/ESM sediments measured along the transect spanning more than 800 km from the Lena River mouth to the shelf (2000–3000 m water depth) varied between âĹij2 % at shallow water depths and 0.8% in deeper water (Broder et al., 2016). In addition, TOC values and general patterns, which reflect fractions of terrigenous OC reaching the slope (based on biomarkers), were within the same range as those measured for the North American Arctic margin (Stein and Fahl, 2000, 2004; Goni et al., 2013). For comparison, an average value for the continental slope of the Gulf of Mexico, where large storage of CH4 hydrates has been proven to exist, is 0.8% ±0.2 (Gulf of Mexico Hydrate Research Consortium). Moreover, according to Arrigo and van Dijken (2011), the total annual NPP over the Arctic Ocean exhibited a statistically significant increase by 20% between 1998 and 2009, due mainly to increases in both the extent of open water (+27%) and the dura- tion of the open water season (+45 days). Most importantly, increases in NPP over the 12 year study period were largest in the eastern Arctic Ocean, most notably in the Siberian (+135%) sector.**

** While interesting, none of this is relevant. This is because, for high methane con-

centrations to exist in the upper few hundreds of meters on the slope, it is past carbon burial (i.e., not recent) that matters.

** We fully admit that we are somewhat perplexed by our findings, given our pre-conceived notions and past speculations. Hence, this portion of the manuscript is speculative. We can rewrite if guidance is given.

**It is interesting that the authors themselves confirmed that: 1) environmental conditions on the ESM are highly conducive for gas hydrates; 2) hydrate occurrence in the other areas of the Arctic, where hydrates were predicted, was confirmed by hydrate recovery; and 3) all the models developed by generations of geologists to predict hydrates in the Arctic used the same assumptions.**

** We would agree with this statement, if logical qualifiers were added. The environmental conditions on the ESM are highly conducive for gas hydrates IF AND ONLY IF THERE IS SUFFICENT METHANE; hydrate occurrence in the other areas of the Arctic, where hydrates were predicted, was confirmed by hydrate recovery AND BY PORE WATER CHEMISTRY IN SHALLOW CORES; all the models developed by generations of geologists to predict hydrates in the Arctic used the same assumptions WHICH ENTIRELY INFER A SOURCE OF CARBON TO PRODUCE METHANE.

** The referee is purposely ignoring two crucial facts, both discussed at length in the MS: (1) all previous works hinge on an assumption of significant methane in shallow sediment; and (2) NO pertinent data to the problem exists beyond our current work.

**If the authors agree that these statements are true, they failed to be critical of their own work, which is based on a handful of inconclusive data obtained on a single expedition, groundless methodology, and a few erroneous assumptions. Instead of casting doubt on the results of others, I would suggest that the authors question their own results and make a greater effort to accumulate clear, interpretable data. I believe I have made it quite clear that there is a huge discrepancy between the results presented by the authors and the far-reaching conclusions they are trying to support with these data. I**

see no way to support publication of this manuscript in its current state.

** On this matter, and as should be even more clear, we entirely disagree.

―――――――――――――――――――――――

---

## Referee Comment (RC2) · Anonymous Referee #2 · 1 Dec 2016

This study, for the first time, presents geochemical data from the Eastern Siberian Margin, an area of the Arctic that is predicted to hold significant amounts of gas hydrates in the sediment. Intriguingly, the authors find no evidence for the presence of methane in the ESM sediments (asides from one site on Lomonosov Ridge) based on a range of pore-water profiles, including total dissolved sulfur and bicarbonate concentrations, and the carbon isotope composition of DIC. These profiles should show evidence of the occurrence of geochemical processes associated with the production and consumption of methane, e.g., AOM, if gas hydrates where occurring in sediment.

General comments

Overall, this study represents an important contribution to the still very limited geo-

chemical data available from this part of the global ocean. I agree with the authors that from all we (sedimentary biogeochemists) have learned it is justified to assume that geochemical profiles, especially sulfate, alkalinity, bicarbonate (and $\delta$13C-DIC), should show clear evidence of microbially mediated processes associated with methane production (or more accurately evidence for very high organic turnover rates) and methane consumption, especially sulfate reduction coupled to AOM. It is very intriguing that these signals are absent from the profiles present here. It is somewhat unfortunate that the authors did not conduct any direct methane analyses (and somewhat surprising giving their initial aim of the research expedition) but given the other detailed geochemical data, this is not a major issue. I feel like the transects are very well spaced out thus allowing for the conclusion that the ESM is predominately gas hydrate-free. The somewhat lengthy methods section reflects the cautious and tedious job the authors have done during sample collection and analysis thus giving confidence that the provided data is of very high quality. While the methods applied in this study are not new and the lack of methane results in what can be viewed as rather "boring" profiles, it is the underlying implication of not having gas hydrates in this area despite previous predictions that makes this study so important.

However, several aspects detailed below require major revisions to strengthen the geochemical framework of this study.

Additionally, the manuscript can be shortened significantly. There are several sections that are repetitions (see next section) and to me the very detailed discussion on the fidelity of the data obtained by the Rhizone sampling technique, albeit warranted, distracts from the main message of the manuscript. Along the same lines, the authors have included several plots, e.g., Figures 13, 14 and 16 that do not help conveying the main message of the study and are irrelevant. Figures 13 and 14 are not even mentioned in the Discussion section. Figure 13c is not mentioned in the manuscript at all?!

Specific comments:

[Figure]

Figures 1 and 3: I suggest placing both maps next to each other in one figure (ie., Fig.1a and 1b). This would make it much easier for the reader to find out where the sampling sites are located relative to predicted gas hydrate occurrence.

Figure 4: This is a nice picture but does not convey any important information. Given the total number of figures in this manuscript, I suggest deleting it.

Figure 6-9: These figures are hard to read. I suggest plotting each core in a specific figure in a different color rather than all data in one figure in the same color. There is very limited discussion/comparison of the ACEX data; why plot it then?

Lines 187-242 vs. lines 616-657: The sections are basically saying the same thing with a few additional points in the latter, discussion section. I suggest removing lines 616-657 and taking the few "new points" that are mentioned here and adding them to the background section. I found it tiring to read the same "intro to reading pore-water profiles" twice.

"Rhizone experiments": These are very helpful experiments that install additional confidence in this comparably novel sampling technique. With that being said the description of these experiments, including the results and discussion of the results take up a lot of space and distract from the main story of the manuscript. I suggest moving all of this into a supplementary material section. This would include the experiment description (line 310), section 4.3, the discussion sections 5.1 and 5.2 and Figures 5 and 11 (and maybe 12 if the authors think that the porosity-rhizone aspect could also be trimmed), Tables 1 and 2. Basically, all we need to know is what is in the short summary in lines 606-614. The reader can be referred to the supplementary material for the detailed experiments.

Dissolved hydrogen sulfide "analysis": It seems like the authors did not actually do any sulfide analyses but just "visually" observed whether white precipitates were forming when zinc acetate was added. To me this is not an appropriate "analysis" to detect hydrogen sulfide. This is especially important since the authors did not do any sulfate

analyses but only analyzed total dissolved sulfur and –based on their visual "analysis" of the sampling vials- assumed that no hydrogen sulfide was present and the total sulfur only reflects sulfate. I strongly suggest doing at least a few hydrogen sulfide analyses with the Cline method, for example of the samples from deeper layers especially on the cores from Lomonosov Ridge, to confirm the absence of hydrogen sulfide.

Lines 176-178- Microbial processes at cold seafloor temperatures: I disagree with the authors here. There are plenty of studies that have shown that organic carbon turnover rates or "bacterial degradation" in high latitude environments are/can be as high as in mid-latitude or tropical environments. For example:

Glud et al., 1998: Benthic mineralization and exchange in Arctic sediments (Svalbard, Norway) Arnosti et al., 2005: Anoxic carbon degradation in Arctic sediments: Microbial transformations of complex substrates

Carbon isotope sections: Generally, the sections discussing the carbon isotope system, e.g., processes associated with carbon isotope fractionation, the discussion of the carbon isotope data etc. is very weak and needs more clarification. Also, it is incorrect to present equations (1) and (8) with 12C and state that it indicates "depletion in 13C". As such, the equations written just present the reaction of one organic molecule containing 12C to bicarbonate which of course also has to contain 12C. Please take the notations out.

Line 227-229: This needs to be expanded and maybe clarified. Both the Holler and the Yoshinaga references are discussing carbon isotope fractionation during AOM. As stated here, the authors only consider the original 13C-depleted value of the CH4 in explaining the light DIC formed. Additionally consider: Alperin, M.J., Reeburgh, W.S., Whiticar, M.J., 1988. Carbon and hydrogen isotope fractionation resulting from anaerobic methane oxidation. Glob. Biogeochem. Cycles 2, 278–288. Martens, C.S., Albert, D.B., Alperin, M.J., 1999. Stable isotope tracing of anaerobic methane oxidation in the gassy sediments of Eckernforde Bay, German Baltic Sea. Am. J. Sci. 299, 586–

610. And for the first part, asides from Paull et al., a reference such as Whiticar, M.J., 1999. Carbon and hydrogen isotope systematics of bacterial formation and oxidation of methane. Chem. Geol. 161, 291–314.

Line 681-687: Similar to the previous carbon isotope section, there is some more detail needed here. For example, carbon isotope fractionation during organoclastic sulfate reduction needs to be discussed. The Chatterjee reference (which should be 2011 not 2001) is insufficient here.

Line 706: "almost necessarily implies CH4 oxidation.. ". This statement needs an explanation and the appropriate literature. . .

Results section: When you list what the concentrations were, they are in past tense, when you describe what the reader sees in the graph, this is in present tense.

Lines 508-519, Figure 14: This is a nice exercise but I am wondering why this is included? I could not find any reference to this approach/figure in the discussion section. If it is not relevant to your discussion-delete! Or add a section in the Discussion part that evaluates the plot.

Lines 728-733 and elsewhere: I disagree with this general interpretation. Many of the collected cores also show decreases in sulfur concentration which point to the occurrence of organoclastic sulfate reduction, and you interpret the delta13C-DIC profiles as being imprinted by this process! While the dissolved Mn profiles can be interpreted as reflecting dissimilatory manganese oxide reduction, there has been a lot of recent work discussing the –somewhat intriguing- manganese biogeochemistry of Arctic ocean sediments, including evidence for dissolved manganese profiles reflecting diagenetic remobilization of Mn and diffusion from deeper sediment intervals. I suggest preparing this section with more caution. For reference: März et al. 2011: Manganese-rich brown layers in Arctic Ocean sediments: Composition, formation mechanisms, and diagenetic overprint (and references therein).

Line 735-737: This section is somewhat incorrect as well. What Mn and Fe is consumed ? I assume you are now referring to Mn- and Fe-oxides. I suggest: 1) making it clear that dissolved Mn and Fe are produced during dissimilatory Mn- and Fe-oxide reduction; 2) highlighting that the reason for the decline in concentrations are consumption processes (assuming steady state you would otherwise expect constant pore-water values below the current reaction zone), which likely include the reaction of Fe with hydrogen sulfide, and interactions of Fe with Mn-oxides. (Again the sedimentary Mn story may be more complicated; see comment above); 3) stepping back from the idea that there is "complete consumption of Fe and Mn". If you are referring to the oxides, then especially in the case of Fe it is the very reactive (towards H2S) iron (oxyhydr)oxide phases that are being reduced (see Canfield et al., 1992: The reactivity of sedimentary iron minerals towards sulfide) but there is without a doubt no "complete Fe consumption"!

Section 5.7 and Figure 16: In this form, I find the plot misleading and somewhat irrelevant (or not providing any new helpful information). First, as you have discussed, sites with methanogenesis and AOM are characterized by much higher DIC concentrations and much lighter delta13C-DIC values than sites lacking these processes. If you multiply these two, of course you get more negative values at the AOM sites. Second, I am not sure what you are actually plotting as ˆDIC here ? You state that other authors have used the concentrations at the seafloor and the SMT. What do you do for your data where there is no SMT? Third, in line 760 you state "two basic models help explain the relationships in Figure 16." However, you are in the following section only discussing the C:S ratios, including their relative changes with depth (as you are interpreting them from the mudline downward using the changes in DIC*delta13C-DIC as an alternative measure for depth). Why then do such a crossplot? On a side note – why is the ratio for the OSR model increasing past 2:1? Because the DIC reflects additional bicarbonate production by dissimilatory Mn and Fe oxide reduction rather than only from sulfate reduction ? Fourth, in line 747 you are stating that "a flux of HCO3- from below the SMT can augment the DIC produced…Thus, changes in alkalinity relative to sulfate

often exceed 1:1...". Now the conclusion from your model/plot is that -line 768-769- "..CH4 charged locations with migrating DIC must have C:S molar ratios in excess of 1:1...". So what have we learned? It would be honest to also mention the studies by Snyder et al., 2007 and Wehrmann et al. 2011 (Coupled organic and inorganic carbon cycling in the deep subseafloor sediment of the northeastern Bering Sea Slope (IODP Exp. 323)) in lines 740-750 who used fluxes instead of concentrations.

Lines 808-816: I suggest expanding this section, and maybe including relevant literature to support the different hypotheses, even if it means speculating. The finding that CH4 is low in the sediment in this part of the Arctic is the essential message of this study; the major question that arises is why? Do the ACEX studies provide any clues that would support any of your hypotheses? Lines 817-820 need more details and references as well!

Discussion section: the Ba and Sr data are not discussed.

Technical corrections:

Line 17:...methane (CH4)...

Line 27: replace "nutrient" with "phosphate and ammonium"...Also, the "nutrient data" does not provide evidence for the dominance of metal oxide reduction but evidence for very low organic carbon turnover rates. Please re-phrase.

Line 35:...substantial amounts of CH4. (or something similar)

Line 44:...in the form of gas hydrates,

Line 79/80: Please re-phrase. Methane is not "reacting with sulfate". Obviously this is still debated but a term like "sulfate reduction coupled to the anaerobic oxidation of methane" or "sulfate reduction-coupled AOM" is more appropriate or rephrase to "microbes utilize methane..." or so.

Line 84: "Where CH4 flux to...

Line 96-100: I suggest deleting these sentences. First, giving the total number of samples etc. is a little too much detail for the intro. Second, putting a "conclusion" sentence here, seems confusing (this is not the abstract).

Lines 150-157: Change all [] to ()

Line 152: Limited information on what?

Line 193: I don't think the Schulz, 2000 reference is appropriate here. I suggest Boudreau (1997) and Iversen and Jørgensen (1993) instead.

Line 241: Delete summary sentence.

Line 273: Remove ; at end.

Line 314: Table 4?

Line 338: Should be Table 3.

Line 340: ..dissolved sulfur and metal concentrations. . .

Line 342: HNO3

Lines 353-381: Please shorten these sections. These are very common methods and you can reference the appropriate literature. We don't need to know exactly how much of which chemical you weight in etc.

Line 389: Can you find a better title for this section than "Generalities"?

Line 390: Table 1?

Line 405-412: Move to methods section.

Line 422-428, section 4.3: As outlined above, I suggest moving this to an supplementary material section.

Line 459: I am not sure a "decrease" can "change"

Line 477: Replace "faster" (time component) with " displayed a steeper decrease" or so.

Line 479: Replace "sulfate" with "sulfur"

Line 480: I don't see where the 0.98 comes from.

Line 482 etc.: I suggest taking the "nutrient" term out. As you discuss, you are considering phosphate and ammonium as mineralization products. Instead of the discussion in Lines 483-485, why not just say "..the mineralization products...."

Line 621: Replace symbol.

Line 633: I am not sure this is correct. A concave-down sulfate concentration profile usually implies on-going organoclastic sulfate reduction above the SMT. Otherwise you get a linear profile driven by diffusion of sulfate from the sediment-water interface to the SMT.

Lines 635-637, 637-639, 639-641: These sentences need references.

Line 650 etc.: Do you actually calculate the methane fluxes somewhere? If so, how were methane fluxes calculated? What was taken into consideration? What if organoclastic sulfate reduction is occurring in close vicinity above the SMT, ie, your upward methane flux would then not be equal to the downward sulfate flux (at a 1:1 ratio). Where is the methane flux data?

Line 671: "...imply a $SO_4^{2-}$ flux.."

Line 674: 6.8 mol/m2

Line 687-688: Ok, it has a different ratio...and ? I am not sure you mention this here?

Line 706: 43.54‰

Line 708-709: I don't think that this is an "issue" but as you point out earlier it is very common to only observe hydrogen sulfide very close to the SMT. Nonetheless, if "none

was detected" what do you conclude from that (ie, here please insert a short discussion on the reaction of hydrogen sulfide with dissolved iron and iron oxides, pyrite formation etc)?

Line 724: manganese oxide reduction, iron oxide reduction; also denitrification and nitrate reduction ???

Line 742: "The idea. . ." There is a word missing here.

Line 779: Can you find a better title than "Explanations"?

---

## Author Comment (AC2) · 1 Dec 2016

We thank this referee for a very thoughtful, constructive and detailed review. It is an exemplary review, which, of course, will take us some time to fully address. We generally agree with the commentary, and will promptly buy this person a round should they ever in the future identify themselves.
* * *

---

## Author Response (AR1)

Response to associate editor request for major revisions

Low methane concentrations in sediment along the continental slope north of Siberia: Inference from pore water geochemistry; MS No.: bg-2016-308; Clint M. Miller et al.

Specific changes requested by editor: Referee/Editor Comments (*italic font*) with direct responses (**bold font**).

How your pore-water profiles are effective in ruling out bubble-mediated methane transport.

As expressed in the MS, sites with bubble-mediated CH4 transport have truly different chemistry that bears no similarity to those observed during SWERUS-C3 Leg 2. Additionally, Section 5.6 describes how, given sufficient permeability and time, CH4 charged sediments show connectivity of pore water chemistry over hundreds of meters. Thus, CH4 ebullition near our coring locations is unlikely.

The authors agree that advection between transects should be discussed more completely. Therefore, we have added discussion focusing on the following:

- 1. No major physiographic provinces exist between transects. All major sedimentary regions within the field area are included within the transects.
- 2. All observed large-scale gas hydrate accumulations with bubble-mediated CH₄ transport also have significant CH₄ diffusion. This is because sediment sequences with gas hydrate have gas hydrate formation, gas hydrate dissociation, and gas hydrate dissolution all co-occurring. The pore water gradients between the top of the gas hydrate stability zone and the seafloor occur due to steady-state formation and dissolution.

Therefore, it is unlikely that widespread gas hydrate accumulations exist and are somehow only venting in small localized regions.

Be more precise about your geographical coverage visavi earlier work. Avoid "East Siberian Margin" and instead describe your study area as something like the slope and rise sediments off the Chukchi and East Siberian Sea. When referring to earlier work on the shelf system, describe that as Laptev and East Siberian Sea shelves. These two systems are very different and should not be lumped together. The study area is now referred to as the, "slope and rise sediments off the Chukchi and East Siberian Sea (CESS)" throughout the MS.

I agree with both reviewers that your paper can be substantially reduced in length (by up to 1/3-1/2).

The MS has been extensively condensed, and several sections (4.2, 4.3, 5.1, and 5.3), figures (4, 5, 11, and 12), and tables (1 and 2) have been removed. Additionally, every section has been reduced especially sections 2.2 and 3.1-3.4.

**Specific comments from Referee 2:**

Figures 1 and 3: I suggest placing both maps next to each other in one figure (ie., Fig. 1a and 1b). This would make it much easier for the reader to find out where the sampling sites are located relative to predicted gas hydrate occurrence. Figure 4: This is a nice picture but does not convey any important information. Given the total number of figures in this MS, I suggest deleting it.

The authors agree with both points, however combining Figures 1 and 3 so that both are readable is challenging. Therefore, the caption has been imbedded in Figure 1, and the symbol description of Figure 3 (now Figure 2) is in prose. Figure 4 has been deleted from the MS.

Figure 6-9: These figures are hard to read. I suggest plotting each core in a specific figure in a different color rather than all data in one figure in the same color. There is very limited discussion/comparison of the ACEX data; why plot it then?

We regret these figures are difficult to read. Given the extremely large dataset over this vast region, it is difficult to clearly present results concisely. It seemed to us somewhat overwhelming to plot the chemistry at every site independently, or alternatively every species analyzed at multiple sites independently. The authors have tried many different plotting methods including plotting each core in a different color. This style did not improve figure readability, and removes the color distinction carried over into Figure 8 (which is the most important figure in this group). Instead, we have chosen to increase panel, symbol, and line widths while minimizing white space. The ACEX data has been removed, as suggested, and the legends are imbedded within the panels. Hopefully, this improves readability without lengthening the paper.

Lines 187-242 vs. lines 616-657: The sections are basically saying the same thing with a few additional points in the latter, discussion section. I suggest removing lines 616-657 and taking the few "new points" that are mentioned here and adding them to the background section. I found it tiring to read the same "intro to reading pore-water profiles" twice.

**The entire section 5.3 "Reading the Pore Water Profiles" has now been removed.**

"Rhizone experiments": These are very helpful experiments that install additional confidence in this comparably novel sampling technique. With that being said the description of these experiments, including the results and discussion of the results take up a lot of space and distract from the main story of the MS. I suggest moving all of this into a supplementary material section. This would include the experiment description (line 310), section 4.3, the discussion sections 5.1 and 5.2 and Figures 5 and 11 (and maybe 12 if the authors think that the porosity-rhizone aspect could also be trimmed), Tables 1 and 2. Basically, all we need to know is what is in the short summary in lines 606-614. The reader can be referred to the supplementary material for the detailed experiments.

Rhizons have been subject to debate leading the some misunderstanding in their applicability to marine settings. The authors believe these experiments provide some much needed clarification to Rhizon sampling fidelity, but agree with the reviewer that this section distracts from the primary purpose to the MS. Therefore, lines 310-315, sections 4.2, 4.3, 5.1, 5.2, Figures 5, 11, 12, Tables 1, and 2 have been edited and moved to supplementary materials.

Dissolved hydrogen sulfide "analysis": It seems like the authors did not actually do any sulfide analyses but just "visually" observed whether white precipitates were forming when zinc acetate was added. To me this is not an appropriate "analysis" to detect hydrogen sulfide. This is especially important since the authors did not do any sulfate analyses but only analyzed total dissolved sulfur and based on their visual "analysis" of the sampling vials- assumed that no hydrogen sulfide was present and the total sulfur only reflects sulfate. I strongly suggest doing at least a few hydrogen sulfide analyses with the Cline method, for example of the samples from deeper layers especially on the cores from Lomonosov Ridge, to confirm the absence of hydrogen sulfide.

**The term "analysis" may be confusing. This section has been reworded to describe "visual inspection" of ZnS precipitate. Unfortunately, pore water sulfide analyses are not possible.**

Lines 176-178- Microbial processes at cold seafloor temperatures: I disagree with the authors here. There are plenty of studies that have shown that organic carbon turnover rates or "bacterial degradation" in high latitude environments are/can be as high as in mid-latitude or tropical environments. For example: Glud et al., 1998: Benthic mineralization and exchange in Arctic sediments (Svalbard, Norway) Arnosti et al., 2005: Anoxic carbon degradation in Arctic sediments: Microbial transformations of complex substrates.

Here, we do not state that bacterial degradation is lower in high latitude than lower latitudes. Rather, line 176 states that burial "might" be enhanced by colder temperatures. This idea is quite logical given our understanding of bacterial processes at different temperatures, and has been discussed in the literature previously (Ex. Darby et al., 1989; Max and Lowrie, 1993). We provide no evidence either way, but simply supply this as a possibility. To make this abundantly clear we have reworded line 168.

Carbon isotope sections: Generally, the sections discussing the carbon isotope system, e.g., processes associated with carbon isotope fractionation, the discussion of the carbon isotope data etc. is very weak and needs more clarification. Also, it is incorrect to present equations (1) and (8) with 12C and state that it indicates "depletion in 13C". As such, the equations written just present the reaction of one organic molecule containing 12C to bicarbonate which of course also has to contain 12C. Please take the notations out.

The subscript notations have been removed from both equations. See the following comment regarding improving the carbon isotope discussion.

*Line 227-229: This needs to be expanded and maybe clarified. Both the Holler and the Yoshinaga references are discussing carbon isotope fractionation during AOM. As stated here, the authors only*

consider the original 13C-depleted value of the CH₄ in explaining the light DIC formed. Additionally consider: Alperin, M.J., Reeburgh, W.S., Whiticar, M.J., 1988. Carbon and hydrogen isotope fractionation resulting from anaerobic methane oxidation. Glob. Biogeochem. Cycles 2, 278–288. Martens, C.S., Albert, D.B., Alperin, M.J., 1999. Stable isotope tracing of anaerobic methane oxidation in the gassy sediments of Eckernforde Bay, German Baltic Sea. Am. J. Sci. 299, 586–610. And for the first part, asides from Paull et al., a reference such as Whiticar, M.J., 1999. Carbon and hydrogen isotope systematics of bacterial formation and oxidation of methane. Chem. Geol. 161, 291–314.

We consider DIC 13C depletion at to result from a variety of factors in CH4 charged sediments including: fractionation during AOM, fractionation during organoclastic sulfate reduction (OSR) and other bacterially mediated reactions, differential diffusion of 12CH4 and 13CH4 from deep sediments, as well as the light CH4 input from below. The authors thought this was clear, however this entire section has been reworded to clarify.

Line 681-687: Similar to the previous carbon isotope section, there is some more detail needed here. For example, carbon isotope fractionation during organoclastic sulfate reduction needs to be discussed. The Chatterjee reference (which should be 2011 not 2001) is insufficient here.

Indeed, the authors interpret the observed 13C depletion as fractionation during OSR and other bacterially mediated reactions. This section has been reworded to clarify this point.

*Line 706: "almost necessarily implies CH*4 oxidation.. ". This statement needs an explanation and the appropriate literature. . .

The  $\delta^{13}$ C-DIC values are comparable to a great many published results from CH4 charged sediments. Additionally, these results imply CH4 oxidation because no other process can realistically create <-40‰  $\delta^{13}$ C-DIC values. This section has been rewritten to make this clear.

*Results section: When you list what the concentrations were, they are in past tense, when you describe what the reader sees in the graph, this is in present tense.*

Discussion of concentrations are now in present tense.

Lines 508-519, Figure 14: This is a nice exercise but I am wondering why this is included? I could not find any reference to this approach/figure in the discussion section. If it is not relevant to your discussiondelete! Or add a section in the Discussion part that evaluates the plot.

Deviations from the Redfield ratio in marine environments may be caused by different organic matter sources (terrigenous?) than primary productivity. Given this MS's results differ markedly from previously assumptions regarding past organic matter turnover; this exercise seems particularly germane! These results, however, are not enough by themselves to show organic source, but simply imply the terrigenous component may be important. This section has been rewritten, and a short paragraph has been added in the discussion section to explain this figure more completely.

Lines 728-733 and elsewhere: I disagree with this general interpretation. Many of the collected cores also show decreases in sulfur concentration which point to the occurrence of organoclastic sulfate reduction, and you interpret the delta13C-DIC profiles as being imprinted by this process! While the dissolved Mn profiles can be interpreted as reflecting dissimilatory manganese oxide reduction, there has been a lot of recent work discussing the –somewhat intriguing- manganese biogeochemistry of Arctic Ocean sediments, including evidence for dissolved manganese profiles reflecting diagenetic remobilization of Mn and diffusion from deeper sediment intervals. I suggest preparing this section with more caution. For reference: März et al. 2011: Manganese rich brown layers in Arctic Ocean sediments: Composition, formation mechanisms, and diagenetic overprint (and references therein).

The first author was unware of März et al. (2011), and thanks the reviewer for this comment. Indeed, the reviewer is correct the Mn profiles in this MS may be partially affected by diagenetic remobilization of Mn below the sampled intervals. The above section has been altered to discuss this possibility.

Line 735-737: This section is somewhat incorrect as well. What Mn and Fe is consumed? I assume you are now referring to Mn- and Fe-oxides. I suggest: 1) making it clear that dissolved Mn and Fe are produced during dissimilatory Mn- and Feoxide reduction; 2) highlighting that the reason for the decline in concentrations are consumption processes (assuming steady state you would otherwise expect constant pore-water values below the current reaction zone), which likely include the reaction of Fe with hydrogen sulfide, and interactions of Fe with Mn-oxides. (Again the sedimentary Mn story may be more complicated; see comment above); 3) stepping back from the idea that there is "complete consumption of Fe and Mn". If you are referring to the oxides, then especially in the case of Fe it is the very reactive (towards H2S) iron (oxyhydr)oxide phases that are being reduced (see Canfield et al., 1992: The reactivity of sedimentary iron minerals towards sulfide) but there is without a doubt no "complete Fe consumption"!

**See the above comment response. This section has been reworded to reflect the possible importance of Mn remobilization in these sediments. The portion on "complete" consumption has been removed.**

Section 5.7 and Figure 16: In this form, I find the plot misleading and somewhat irrelevant (or not providing any new helpful information). First, as you have discussed, sites with methanogenesis and AOM are characterized by much higher DIC concentrations and much lighter delta13C-DIC values than sites lacking these processes. If you multiply these two, of course you get more negative values at the AOM sites. Second, I am not sure what you are actually plotting as *DIC* here ? You state that other authors have used the concentrations at the seafloor and the SMT. What do you do for your data where there is no SMT? Third, in line 760 you state "two basic models help explain the relationships in Figure 16." However, you are in the following section only discussing the C:S ratios, including their relative changes with depth (as you are interpreting them from the mudline downward using the changes in DIC\*delta13C-DIC as an alternative measure for depth). Why then do such a crossplot? On a side note – why is the ratio for the OSR model increasing past 2:1? Because the DIC reflects additional bicarbonate production by dissimilatory Mn and Fe oxide reduction rather than only from sulfate reduction? Fourth, in line 747 you are stating that "a flux of HCO3- from below the SMT can augment the DIC produced. . .Thus, changes in alkalinity relative to sulfate often exceed 1:1...". Now the conclusion from your model/plot is that -line 768-769-"..CH4 charged locations with migrating DIC must have C:S molar ratios in excess of 1:1...". So what have we learned? It would be honest to also mention the studies by Snyder et al., 2007 and Wehrmann et al. 2011 (Coupled organic and inorganic carbon cycling in the deep subseafloor sediment of the northeastern Bering Sea Slope (IODP Exp. 323)) in lines 740-750 who used fluxes instead of concentrations.

As quoted above, line 748 in the previously submitted MS discusses the upward DIC flux common to sites with high CH4 concentrations. This flux is often ignored, but has been shown to broadly affect

both solute concentrations and isotopic values in CH4 charged sediments (Dickens and Snyder, 2009; Chatterjee et al., 2011). An improved section 5.5 clarifies the signatures of AOM and OSR for which this flux is a strong component. Importantly, the x-axis of Figure 16 is NOT simply a result of high DIC concentrations, but rather shows a very large continuum of values in supposedly similar environments which do not follow the 1:1 and 2:1 ratios many authors use. Additionally, plotting the sites from this MS versus locations with high CH4 flux clearly juxtaposes our results.

Specifically from above: The DIC question is irrelevant because all locations other than results from this MS have SMTs. The two models were intended to expand the above concept, but appear to be confusing. We are therefore, removing them from the MS. The Snyder and Wehrmann references have been added.

Lines 808-816: I suggest expanding this section, and maybe including relevant literature to support the different hypotheses, even if it means speculating. The finding that  $CH_4$  is low in the sediment in this part of the Arctic is the essential message of this study; the major question that arises is why? Do the ACEX studies provide any clues that would support any of your hypotheses? Lines 817-820 need more details and references as well! Discussion section: the Ba and Sr data are not discussed.

**This section has been expanded with references.**

Technical corrections: Line 17:...methane (CH4)...

**Fixed**

*Line 27: replace "nutrient" with "phosphate and ammonium". . .Also, the "nutrient data" does not provide evidence for the dominance of metal oxide reduction but evidence for very low organic carbon turnover rates. Please re-phrase.*

Line 35:... substantial amounts of CH4. (or something similar); Line 44:... in the form of gas hydrates,

**Fixed**

Line 79/80: Please re-phrase. Methane is not "reacting with sulfate". Obviously this is still debated but a term like "sulfate reduction coupled to the anaerobic oxidation of methane" or "sulfate reduction-coupled AOM" is more appropriate or rephrase to "microbes utilize methane..." or so.

Line 84: "Where CH4 flux to. . .

**Fixed**

Line 96-100: I suggest deleting these sentences. First, giving the total number of samples etc. is a little too much detail for the intro. Second, putting a "conclusion" sentence here, seems confusing (this is not the abstract).

Lines 150-157: Change all [] to ()

**Fixed**

Line 152: Limited information on what?

**Geologic**

Line 193: I don't think the Schulz, 2000 reference is appropriate here. I suggest Boudreau (1997) and Iversen and Jørgensen (1993) instead.

**Changed**

Line 241: Delete summary sentence; Line 273: Remove ; at end; Line 314: Table 4? Line 338: Should be Table 3. Line 340: ..dissolved sulfur and metal concentrations. . . Line 342: HNO3 All Fixed

*Lines 353-381: Please shorten these sections. These are very common methods and you can reference the appropriate literature. We don't need to know exactly how much of which chemical you weight in etc.*

**This section has been significantly shortened**

Line 389: Can you find a better title for this section than "Generalities"?

**Yes**

Line 390: Table 1?; Line 405-412: Move to methods section.

**Fixed**

*Line 422-428, section 4.3: As outlined above, I suggest moving this to an supplementary material section.*

**This section has been moved.**

Line 459: I am not sure a "decrease" can "change"

**Changed to "is most pronounced."**

Line 477: Replace "faster" (time component) with " displayed a steeper decrease" or so; Line 479: Replace "sulfate" with "sulfur"

**Fixed**

Line 480: I don't see where the 0.98 comes from.

**The ratio to change in alkalinity to sulfate.**

Line 482 etc.: I suggest taking the "nutrient" term out. As you discuss, you are considering phosphate and ammonium as mineralization products. Instead of the discussion in Lines 483-485, why not just say "..the mineralization products. . .."

**"Nutrients" removed and line 483 reworded.**

Line 621: Replace symbol.

**Fixed**

Line 633: I am not sure this is correct. A concave-down sulfate concentration profile usually implies ongoing organoclastic sulfate reduction above the SMT. Otherwise you get a linear profile driven by diffusion of sulfate from the sediment-water interface to the SMT.

Line 633 has been reworded.

Lines 635-637, 637-639, 639-641: These sentences need references.

**References added.**

Line 650 etc.: Do you actually calculate the methane fluxes somewhere? If so, how were methane fluxes calculated? What was taken into consideration? What if organoclastic sulfate reduction is occurring in close vicinity above the SMT, ie, your upward methane flux would then not be equal to the downward sulfate flux (at a 1:1 ratio). Where is the methane flux data?

We do not understand this comment. We infer little to no CH4 in the sediments, therefore we cannot calculate CH4 flux.

Line 671: "...imply a SO42- flux.."; Line 674: 6.8 mol/m2

**Fixed**

Line 687-688: Ok, it has a different ratio. . . and ? I am not sure you mention this here?

Section 5.5 goes into detail explaining the importance of this ratio.

Line 706: 43.54‰

**Fixed**

Line 708-709: I don't think that this is an "issue" but as you point out earlier it is very common to only observe hydrogen sulfide very close to the SMT. Nonetheless, if "none was detected" what do you conclude from that (ie, here please insert a short discussion on the reaction of hydrogen sulfide with dissolved iron and iron oxides, pyrite formation etc)?

**This section has been reworded.**

*Line 724: manganese oxide reduction, iron oxide reduction; also denitrification and nitrate reduction ???; Line 742: "The idea. . ." There is a word missing here.*

**All fixed**

Line 779: Can you find a better title than Explanations"?

How about "Possible Explanations for Methane Absence?"

**Response to Referee 1:**

The authors thank Reviewer 1, and wish to make some general comments before discussing individual criticisms. Unfortunately, this review is difficult to read, and many comments are not germane to the original MS. In order to explain our overall response, we first summarize this review:

1. The MS does not contain additional data, even though it is one of the most comprehensive pore water chemistry data sets within a single effort generated to date in any region, let alone from a previously virtually unexplored section of the Arctic Ocean. 2. The pore water results cannot be used to understand CH₄ abundance, although numerous papers in multiple regions, including the Arctic offshore Alaska, convincingly demonstrate the opposite (eg. Borowski et al., 1996; Jørgensen et al., 2001; Torres and Kastner, 2009; Treude et al., 2014; etc.).

3. The primary interpretation and conclusion conflicts with previous speculations in the region, although no pertinent information to the problem exists beyond the current work.

Additionally, in certain parts, the reviewer appears to be discussing a different MS than ours. Page C4, for example, the reviewer criticizes the authors for "vigorous" referencing of a study (Nauhaus et al., 2002) that we neither cite nor discuss.

**Specific Comments from Referee 1:**

However, this ms rather demonstrates that the current state of knowledge of pore water biogeochemistry in particular areas of the ocean is very incomplete; a great deal of effort will be required in order to improve our understanding of the relationship between sulfur and carbon cycling in the Arctic.

We do not follow this comment. The community has published volumes of detailed research about pore water chemistry in regions that contain gas hydrate (Borowski et al., 2000; Torres et al., 2004; Treude et al., 2005; Dickens and Snyder, 2009; Coffin et al., 2013). This is stressed throughout the updated MS, and where possible, we have added additional discussion and references.

The authors of the ms come to the following conclusions: 1) Based on interpretation of the pore water profiles, they found no evidence for upwardly diffusing CH4. 2) Based on these data, they strongly suggested that gas hydrates do not occur on the slopes of the ESM. 3) They claimed that previous investigators who suggested that hydrate deposits exist in the Arctic shelf/slope based on results of their investigations were simply wrong.

This point has been clarified in the updated text. We strongly suggest WIDESPREAD gas hydrates do not occur as previously speculated. It is important to note our results and interpretations DO NOT CONFLICT with any previous data or direct results from the region. First of all, I do not understand why, when reporting low CH4 concentrations and the relationship between CH4 and sulfate dynamics in the pore water, the authors did not measure the concentrations of either parameter. Is it not logical to measure CH4 and sulfate in pore water if one is going to report "low methane concentrations in the sediments"? These are rather routine measurements.

Not all measurements could be generated given the limitations of the expedition and subsequent funding. However, we did measure the S concentrations of pore water, and know these are representative of SO42-, because we checked for H2S as well as measured dissolved Ba2+. Moreover, there are problems with generating quantitative dissolved CH4 profiles in marine sediment because of degassing associated with changes in pressure and temperature.

Here, it is absolutely crucial to realize that, as stressed in the text and at least to our knowledge, no region with significant  $CH_4$  in moderately shallow sediment (< 500 m) has high dissolved S/dissolved  $SO_4^{2^-}$  in pore waters near the seafloor, as well as other certain chemistry documented here. This is now stated in the updated MS, including discussion of localized areas of high advection and  $CH_4$  venting.

The authors referred to other researchers in their ms to present supportive arguments, but none of these referenced studies avoided taking measurements.

In fact, a significant fraction of the research referenced was partially generated by one of the authors (Dickens). It should be noted that the measurements were not avoided in some means to hide information, but rather that we know how CH4 exists and cycles in many regions, and the most prudent means of tackling the problem at a first-order level over an immense area from an ice-breaker is to generate numerous detailed pore water profiles (Borowski et al., 1999; Snyder et al., 2007; Hu et al., 2015).

In addition, the authors of this ms speculate about the particulate organic carbon (POC) and OC content of sediment, but did not measure either parameter. OC content of sediments should be reported as a number of different carbon stocks, not just POC. This comment is unclear. As numerous studies have shown (Borowski et al., 1996; Dickens, 2000; Hensen et al., 2003; Geprags et al., 2016), the abundance of CH₄ in shallow sediment on continental slopes does not depend on the current supply of POC, but rather on the integrated input of POC over long time intervals (e.g., hundreds of thousands to million year time scales), which cannot be assessed without drilling. This last point has been added to the updated MS.

Second, I do not understand why the presence or absence of  $CH_4$ , either in the sediments or in the water column in this area, should be necessarily connected to the existence or non-existence of hydrates. Are hydrates the only possible source of  $CH_4$  in the Arctic shelf/slope? I believe not; hydrates could be only a tiny fraction of the source, because the hydrate stability zone (HSZ) created by P/T conditions could compose only a small fraction of the sedimentary drape (a few hundred meters), while the sedimentary drape could be a few kilometers thick.

The authors do not understand this comment. Gas hydrates are not a source of CH4, but rather, one phase of CH4 in open and dynamic systems, where CH4 carbon can exist as dissolved gas, free gas, and gas hydrate.

This project cannot link shallow water chemistry profiles to CH4 abundance at truly deep depths, and the MS does not attempt to do so. However, deep CH4 cannot exist as gas hydrates, for reasons of P/T conditions. On the other hand, the presence of gas hydrate in the upper few hundreds of meters of sediment is absolutely related to total CH4 concentrations in pore space, which are linked to shallow sediment through diffusion or in some cases advection. This point is clarified in the updated discussion section.

Third, the purpose of this massive MS is not clear to me. This paper is flooded with equations and details devoted to methods, but mathematics, first of all should be applicable; then, the accuracy of mathematics does not aid in interpreting the inconclusive data.

We do not understand this comment. The purpose, as outlined in the MS, is very clear: we know how pore water chemistry profiles look above gas hydrate systems at numerous locations around the world, and we know how to interpret them at a basic level; we generated such profiles in the region of interest; the pore water profiles do not conform to those at any region where significant CH4 occurs in shallow sediment nor our understanding as to why such profiles arise.

However, we agree the MS can be shortened significantly. The revised MS has moved essentially all of the Rhizon discussion to supplementary materials, deleted several sections, figures, and tables, and streamlined much of the text including the methods section. In total, almost 200 lines were deleted.

Below are my comments on some aspects of this ms. A more detailed look would be as long as the ms itself, because nearly every page of this ms would benefit from clarification. The methodology chosen by the authors of this ms and their level of understanding of the processes they were trying to investigate are my greatest concern. Biogeochemists working in the marine ecosystems have already gained some understanding of the fact that biogeochemical processes associated with diagenetic transformation of organic matter under anaerobic conditions in marine sediments are very complex microbe-mediated processes. These processes involve microorganisms from various physiological groups: aerobic and anaerobic saprophylic and cellulose degrading bacteria, sulfate reducers, methanogens, denitrifiers, and methylotrophs. Transformation of organic matter is a multi-stage process: primary anaerobes decompose polymeric compounds to monomers, which, in turn, serve as a substrate for fermentation agents and gas-producing bacteria. A general conclusion is that the major fraction of OC preserved in the sediments is oxidized to CO2 by the sulfate-reducing bacteria (SRB) and that 2 moles of OC are oxidized for every mole of sulfate reduced: 4H(CH2)n COO- + (3n + 1)SO42-+ H2O!(4n + 4)HCO3- + (3n + 1)HS- + OH- + nH+. When acetate is oxidized completely, the atomic ratio of OC oxidized to sulfate-S reduced is 2 : 1. However, as 'n' increases, the C: S ratio changes; the ratio between the reactants could be different because it depends on the varying nature of the organic matter (Lerman 1982). This is because most of the photosynthate is not immediately available for oxidation; only the low molecular weight (LMW) fraction of dissolved OC (DOC) is rapidly oxidized by SRB, while the high molecular weight (HMW) fraction of POC, which usually increases with depth, is refractory. There are severe restrictions on microbial activity other than substrate availability, including that SR as a biotic process may be more strongly coupled to mineralogy (Ivanov et al., 1989). The knowledge that has been accumulated by scientists so far is very limited and only applicable to those particular ecosystems which were investigated beyond the Arctic.

The authors are perplexed by this comment, as most of it does not pertain to our MS, and it seems to belie faulty logic. At a basic level the referee appears to think biogeochemical processes are so complex that the community cannot obtain overall net chemical reactions and flux of carbon from pore water chemistry.

If we are interpreting the comment correctly, we then return an obvious question: how and why can the community measure similar pore water chemistry profiles at myriad locations and see basic commonalities (e.g., the absence of SO42- above sites with the presence of significant CH4 below), irrespective of the specifics and microbiology involved? No change has been made to the MS regarding this comment.

The most reliable method to trace the course of sulfate reduction in sediments uses radioactive sulfate (35S). By the use of this method it was shown that most reduced 35S-sulfate was in pyrite and organic sulfur (Lein et al., 1982). The relationship between sulfur and carbon cycling in the Arctic marine systems is even more complicated, because the relationships between the sites of primary production and the sites to which organic matter is translocated and deposited, including organic matter delivered to the shelf/slope from surrounding land, are difficult to establish both qualitatively and quantitatively.

We do not know how to respond to this comment, as it mostly does not pertain to our MS. It begins with a statement for which we disagree, and it does not make sense in its entirety. No change has been made to the MS regarding this comment.

A recently published review of CH4 emissions from the seafloor in the Arctic Ocean underscored that the role of SRB in the anaerobic oxidation of methane (AOM) is unclear and the ecology of AOM communities, particularly for high-latitude environments, is not well understood. For that reason, predicting CH4 fluxes, especially those related to hydrate dissociation, remains highly speculative (James et al., 2016). This is because CH4 is transported within the sediments in two different ways: as a dissolved phase (by diffusion or advection) or as free gas (ebullition). Free gas is inaccessible to microbes, which depend on a diffusive transmembrane gas transport. This means that release of free gas through the sediments might not leave any traces in the pore water (see Fig.5 in James et al., 2016).

The authors are unsure how to respond to this comment because it mostly is irrelevant to the topic of our MS. Crucially, we are not concerned how CH4 would escape the seafloor via ENHANCED gas hydrate dissociation in the future, but whether significant CH4 exists in shallow sediment on the SLOPE in the now. We hope this point is clear in the MS.

Additionally, the referenced MS (James et al., 2016) does not conflict with our interpretations. These authors clearly indicate that AOM is a dominant process above methane-charged systems at steady-state conditions, and should impact pore water SO42- gradients (e.g., the very Fig. 5 that the referee emphasizes).

The comment that passage of free gas through sediment does not leave traces in pore water is simply incorrect. The paper and figures by James et al. (2006) by no means suggest this concept, and rightfully so. Pore waters in areas where CH4 advects from below at high rates, such as along faults and fractures (Fig. 5), have truly different chemistry than seawater and anything in our results. The updated Figure 12 emphasizes some of these differences.

Moreover, recently published observational data show that in the Arctic environment, for example in the Alaskan Beaufort Sea continental margin sediments, substantial (30-500  $\mu$ M) concentrations of sulfate can remain below the sulfate-methane transition zone (SMTZ) although mass balance cannot explain the source of sulfate below the SMTZ.

This comment seems to betray a basic misunderstanding of SO42-, CH4 fluxes, and mass balance. SO42- concentrations 0.03-0.5 mM are not "substantial" compared to the ~28 mM in typical seawater. Second, there are at least three known reasons for SO42- below the SMT: 1. Pore water contamination, 2. H2S oxidation, and 3. Barite dissolution. Third, mass balance always applies.

Additionally, the Beaufort Sea results we assume the referee is mentioning (Coffin et al., 2013), show the exact type of shallow pore water profiles which this MS argues would occur above gas hydrates – and do not in our results! No changes have been made to the MS regarding this comment.

In addition, sulfate reduction and anaerobic oxidation of CH4 can occur throughout the methanogenic zone. Experimental data indicated decoupling of sulfate reduction and AOM and competition between

sulfate reducers and methanogens for substrates, suggesting that the classical redox cascade of electron acceptor utilization based on Gibbs energy yields does not always hold even in diffusion-dominated systems (Treude et al., 2014).

Yes, this may true at a detailed and microscopic level, as pointed out by numerous authors, but not at any macroscopic level, at least that we are aware of, excepting odd environments (e.g., brines). The true beauty of pore water chemistry in the deep-sea marine environment is the remarkable consistency of multiple constituents linked to an array of environments. To restate from above, and in the text, all pore waters in methane-charged systems on continental slopes that we are aware of have certain commonalities – none that are seen in any of the pore waters generated in this study. This point has been emphasized in the updated discussion section.

Although they vigorously referred to Nauhaus et al. (2002) as a proxy-establishing experiment, the authors did not give this work any critical assessment. If they had done so, they would have definitely questioned the claim that methanotrophic communities associated with SRB oxidize  $CH_4$  anaerobically in a 1:1 ratio to sulfate reduction. How that could be possible if the reported 4-5-fold increase in H2S production (accumulated over 80 days!) was accompanied by an increase in CH4 concentration of 3 orders of magnitude (from 0.01 to 15.8 mM)? Besides, rates of SR were so small (0.5-3.0  $\mu$ M/d-1) compared to the concentrations of sulfate (103-1.55  $\mu$ DIJ) that the question arises: How could this little change be reliably measured (without using the 35S method, which they did not) and related to AOM?

This is perhaps the most confusing part of this review. We neither cite nor discuss the paper by Nauhaus et al. (2002). Indeed, the first author had not even read this work prior to submittal. We are forced to conjecture the referee is confusing our MS with another. This would explain a number of seemingly inexplicable comments which do not pertain to our text (ex. the referee statement that we "assumed CH4 was being released from destabilizing hydrates, most likely via bubbles and the convective flow of geofluids"). No change has been made to the MS regarding this comment.

Not to mention that this effect has no applicability to the Arctic Ocean.

We are not sure how to address. The referee seems to have a view that physical chemistry and biochemistry in the Arctic Ocean are somehow special, so that basics and inferences gained from elsewhere around the world do not apply.

Here, it is especially important to note the paper by Coffin et al. (2013), as already highlighted in our MS. These authors characterized pore water chemistry in short sediment cores above sequences with known gas hydrates along the shelf and slope of the Beaufort Sea (Arctic), very much as done in our MS. As predicted, they observed shallow SMTs indicative of a strong diffusive methane flux. As obvious in our work, their pore water profiles contrast with those from the slopes off northern Siberia. This point has been emphasized in the updated MS.

Another concern is this: How representative of the area are these data? Only four short transects consisting of 16 stations are presented; each transect is based on data from 2-6 stations. Data from only 2-4 stations represent all core depths. Core lengths vary from 1.95 to 8.43 m (mean length 5.25 m). Eight of the 16 stations are only represented by the very uppermost layers (from 0.16 to 0.39 m) of sediment collected by the multi-corer. These shortest parts are the most valuable as they represent the least disturbed environment, but they are too short to constitute any sort of conclusive data regarding CH4 cycling in the sediments. I can only guess at how the authors succeeded in dividing these tiny cores into numerous parts, each 0.2-0.3 m in length, and accumulated enough data to compare these cores with one of two idealistic schemes to characterize the specific dynamics of processes occurring over a sediment depth of 100 m (Fig.1). Data obtained by other types of sampling (piston/gravity coring) should be treated and interpreted very cautiously as the cores are not only severely disturbed during the coring process, but also chemically altered as they are extracted from the sea floor and lifted onto the ship.

This comment does not make sense. First, the fact that the pore water profiles give nice, detailed gradients in multiple species, demonstrably indicates that the cores have minimal disturbance. We can add core photos to our already long paper, if desired, to further emphasize this point.

Second, the proposition that the uppermost part of a core is the least disturbed and most important to understanding processes is flat out wrong. This is because of the nature of coring, which tends to disturb (or in many cases not recover) the top few cms, and because of bioturbation and reoxidation. Third, the methodology for how the cores were sampled is detailed at length in the manuscript. Indeed, the reviewer criticized the authors earlier for the length of this section, and now claims to only "guess" at how this was accomplished. No changes have been made to the MS regarding this comment.

Finally, the authors plotted water concentration profiles along each transect collectively (!) using colors and symbol types which make it virtually impossible to distinguish between these symbols, making interpretation of the data sets very difficult.

We agree these figures are difficult to read. Given the extremely large dataset over this vast region, it is difficult to clearly present results concisely. It seemed to us somewhat overwhelming to plot the chemistry at every site independently, or alternatively every species analyzed at multiple sites independently. In order to improve readability, we have increased panel, symbol, and line widths while minimizing white space. The ACEX data has been removed to limit clutter, and the legends are inside the panels. Hopefully, this improves readability without lengthening the paper.

From this, it follows that the authors assumed complete uniformity of processes occurring not only in the observed settings located tens of kilometers apart from each other, but also over the entire slope area! This is despite the fact that CH4 fluxes on the East Siberian Arctic Shelf (ESAS), which could be associated with CH4 releases from decaying hydrates, have been reported to vary by orders of magnitude within much smaller scales (Shakhova et al., 2015).

We did not assume complete uniformity of processes. In fact, the MS goes into great detail explaining the range of processes that relate to the pore water chemistry -- processes that have been well documented along many continental slopes. The authors see no reason to add to this already lengthy section.

I see a clear discrepancy between the basic assumptions made by the authors and the methodology used to test these assumptions. The authors assumed CH4 was being released from destabilizing hydrates, most likely via bubbles and the convective flow of geofluids. We do not understand this comment. The referee is stating things that cannot be found anywhere in the text. Additionally, this comment has leapt well beyond anything discussed in the MS. As stated above, and clearly in the MS, we discuss the lack of evidence for CH4 in shallow sediment. Our MS has little bearing on how gas hydrates would be destabilized and how CH4 would be released. No change has been made to the MS regarding this comment.

Despite that, all equations used for estimates refer to the diffusive transport of CH4 and other substances in the sediments. This is understandable; they used what was available. The problem is that the mathematics associated with diffusive transport cannot be used to describe the release of free gas from decaying hydrates. When assuming CH4 release from gas hydrates, one should realize that hydrates convert to free gas; the released gas travels upward much faster than diffusion occurs, through very efficient gas migration paths (chimneys etc.). In most cases, ascending CH4 can avoid oxidation in a few ways. 1) Because free gas resulting from hydrate decay is over pressured, it builds up a gas front; this disturbs sediment layering, creating the characteristic marks of gas release (pockmarks etc.). 2) Only CH4 dissolved in pore water is reachable by microbial communities; CH4 released as free gas (ebullition) is not consumable by microbes. 3) AOM rates are only remarkable as compared to rates of modern methanogenesis, because all synergetic processes should be energetically efficient for all members of the microbial community, including SRB, methanogens and methanotrophs, etc.

This comment is simply wrong. In most locations with gas hydrate, the vast majority of CH4 generated in the sediment ultimately (i.e., long time scales) escapes back to the ocean through diffusion. The assumption that methane-carbon above gas hydrates only returns to the ocean as free gas is entirely incorrect. The authors have added this discussion to the updated MS.

Finally, the authors of the ms used three assumptions to explain their findings. Their first assumption is that bottom seawater on the slope north of Siberia is warming, leading to hydrate destabilization. There are no reports of increased bottom water temperatures along the slope of the Arctic during either the last glacial cycle (Cronin et al., 2012) or the Holocene (Biastoch et al., 2011; Dmitrenko et al., 2011; James et al., 2016). All papers published so far project the response of the hydrate inventory to possible future climate change in the Arctic. The paper of Stranne et al., (2016) the authors refer to assumes a linear rise in ocean bottom water temperatures of 3C over the coming 100 years. This speculative warming of the Arctic is intentionally set higher than in other studies (<2C by Biastoch et al., 2011; <1C by Kretschmer et al., 2015) while modeling assumptions contradict the existing hydrological data (Biastoch et al., 2011; Dmitrenko et al., 2011; James et al., 2016).

This is another statement that absolutely cannot be found in our MS. Nowhere do we assume bottom seawater is warming, nor are warming temperatures even necessary to have CH₄ in pore waters above gas hydrates. Here, again, we wonder if the referee is thinking of another project. No change has been made to the MS regarding this comment.

Their second assumption is the quintessential statement that "Implicit of this finding is that sediments sequences along the ESM lack gas hydrates" following the authors' speculations about why predictions of hydrates on the ESM are so markedly wrong.

This statement has been reworded to better reflect the conclusion, as stated elsewhere in the MS, of WIDESPREAD gas hydrates. However, this is not an assumption, but rather a direct consequence of our results. The pore water chemistry profiles strongly indicate the lack of significant methane concentrations in the upper few hundred meters of sediment; given P/T conditions for gas hydrate, this absolutely implies an absence of gas hydrate.

**Fascinatingly, prior to this expedition, we did expect widespread gas hydrates.**

The authors then suggest that: 1) the significant sea-ice concentration on the ESM diminishes net primary production (NPP); 2) the extremely broad continental shelf prevents accumulation of terrestrial organic-rich sediments; and 3) sediment accumulation is highly variable, so organic matter can be consumed during intervals of low deposition. None of these explanations is true. It was recently shown that the total OC (TOC) content in the ESAS/ESM sediments measured along the transect spanning more than 800 km from the Lena River mouth to the shelf (2000–3000 m water depth) varied between 2 % at shallow water depths and 0.8% in deeper water (Bröder et al., 2016). In addition, TOC values and general patterns, which reflect fractions of terrigenous OC reaching the slope (based on biomarkers), were within the same range as those measured for the North American Arctic margin (Stein and Fahl, 2000, 2004; Goni et al., 2013). For comparison, an average value for the continental slope of the Gulf of Mexico, where large storage of CH4 hydrates has been proven to exist, is 0.8% ±0.2 (Gulf of Mexico Hydrate Research Consortium). Moreover, according to Arrigo and van Dijken (2011), the total annual NPP over the Arctic Ocean exhibited a statistically significant increase by 20% between 1998 and 2009, due mainly to increases in both the extent of open water (+27%) and the duration of the open water season (+45 days). Most importantly, increases in NPP over the 12 year study period were largest in the eastern Arctic Ocean, most notably in the Siberian (+135%) sector.

While interesting, none of this is relevant. This is because, for high CH4 concentrations to exist in the upper few hundreds of meters on the slope, it is past carbon burial (i.e., not recent) that matters. This point has been clarified in the updated text.

It is interesting that the authors themselves confirmed that: 1) environmental conditions on the ESM are highly conducive for gas hydrates; 2) hydrate occurrence in the other areas of the Arctic, where hydrates were predicted, was confirmed by hydrate recovery; and 3) all the models developed by generations of geologists to predict hydrates in the Arctic used the same assumptions.

We would agree with this statement, if logical qualifiers were added. The environmental conditions on the ESM are highly conducive for gas hydrates IF AND ONLY IF THERE IS SUFFICENT METHANE; hydrate occurrence in the other areas of the Arctic, where hydrates were predicted, was confirmed by hydrate recovery AND BY PORE WATER CHEMISTRY IN SHALLOW CORES; all the models developed by generations of geologists to predict hydrates in the Arctic used the same assumptions WHICH ENTIRELY INFER A SOURCE OF CARBON TO PRODUCE CH4.

The referee is ignoring two crucial facts, both discussed at length in the MS: (1) all previous works hinge on an assumption (not evidence) of significant CH₄ in shallow sediment; and (2) NO pertinent data to the problem exists beyond our current work. These point are clarified in the updated text.

If the authors agree that these statements are true, they failed to be critical of their own work, which is based on a handful of inconclusive data obtained on a single expedition, groundless methodology, and a few erroneous assumptions. Instead of casting doubt on the results of others, I would suggest that the authors question their own results and make a greater effort to accumulate clear, interpretable data. I believe I have made it quite clear that there is a huge discrepancy between the results presented by the authors and the far-reaching conclusions they are trying to support with these data. I see no way to support publication of this MS in its current state. We respectfully disagree with the referee's conclusions.

[revised manuscript text omitted]
 all successions and the succession total alles limiter as in a set Matthew Tale dot success |

408 The first aliquot was used to measure total alkalinity using a Mettler Toledo titrator
409 onboard *IB Oden*. Immediately after collection, pore water was diluted2 mL of pore water were
410 diluted to 40 mL with milli-Q water and autotitrated with 0.005M HCl from the original pH to a

| 411 | <del>pH of 5.4</del> . A total of 15 Fifteen spiked samples and 8 eight duplicates were analyzed onboard for     |     |
|-----|--------------------------------------------------------------------------------------------------------------------------------|-----|
| 412 | quality control. Spiked samples were created by pipetting certified reference material (Batch                                  |     |
| 413 | 135; www.cdiac.ornl.gov/oceans/Dickson_CRM) into milli-Q water. Results for spiked samples                                     |     |
| 414 | and duplicates are reported in Table 1. Bateh 135; CRM) into milli Q water. Results for spiked                                 |     |
| 415 | samples and duplicates are reported in Table 3.                                                                                |     |
| 416 | The second aliquot was used to measure the $\delta^{13}C$ composition of DIC ( $\delta^{13}C_{DIC}$ ). Septum                  |     |
| 417 | sealed glass vials prepared with $\frac{\text{H}_3\text{PO}_4}{100\mu\text{L}}$ of 85% phosphoric acid and flushed with helium |     |
| 418 | were prepared before the expedition. The analysis required approximately 40 $\mu$ g of DIC in each                             |     |
| 419 | pore water sample. Onboard alkalinity measurements were used to estimate the correct volume,                                   |     |
| 420 | and this amount was injected into the vials. Samples were sealed in boxes and refrigerated for the                             |     |
| 421 | remainder of the cruise. Four field duplicates, two seawater standards, and a field blank were                                 |     |
| 422 | collected, stored, and analyzed with the samples. The $\delta^{13}C$ -DIC $\delta^{13}C$ -DIC analyses were performed          |     |
| 423 | on a Gasbench II coupled to a MAT 253 mass spectrometer (both Thermo Scientific) at                                            |     |
| 424 | Stockholm University. The $\delta^{13}C$ -DIC carbon isotope composition of DIC is reported in                                 |     |
| 425 | conventional delta notation relative to Vienna PeeDee Belemnite (VPDB). Results for field                                      |     |
| 426 | duplicates and standards are reported in Table 2 Table 1 . Standard deviation for the analyses of                       | For |
| 427 | $\delta^{13}$ C-DIC was less than 0.1 ‰. $\delta^{13}$ C DIC was less than 0.1 per mille. The results for seawater  |     |
| 428 | standards collected onboard are given in Table 3.                                                                              |     |
| 429 | The third aliquot was used to measure dissolved sulfur and metal concentrations.                                               |     |
| 430 | Approximately 3 mL of pore water were placed into acid washed eryovials. Samples were acid                                     |     |
| 431 | preserved with 10 $\mu$ L ultrapure HNO 3 . Additionally, 11 blind field duplicates and 2 field blanks              |     |
| 432 | were collected and processed in the same manner. Concentrations of Ba, Ca, Fe, Mg, Mn, S, and                                  |     |
| 433 | Sr were determined on an Agilent Vista Pro Inductively Coupled Atomic Emission Spectrometer                                    |     |

| 434 | (ICP-AES) housed in the geochemistry facilities at Rice University. Known standard solutions                                                                                                 |         |                                                                            |
|-----|----------------------------------------------------------------------------------------------------------------------------------------------------------------------------------------------|---------|----------------------------------------------------------------------------|
| 435 | and pore fluid samples were diluted 1:20 with 18-M $\Omega$ water. Scandium was added to both                                                                                                |         |                                                                            |
| 436 | standards and samples to correct for instrumental drift (emission line 361.383 nm). Wavelengths                                                                                              |         |                                                                            |
| 437 | used for elemental analysis followed those indicated by Murray et al. (2000). Following initial                                                                                              |         |                                                                            |
| 438 | analysis, an additional dilution, 1:80 with 18-M $\Omega$ water, was analyzed for Ca, Mg, and S. After                                                                                       |         |                                                                            |
| 439 | every 10 analyses, an International Association of Physical Sciences (IAPSO) standard seawater                                                                                               |         |                                                                            |
| 440 | spiked sample and a blank were examined for quality control. Relative standard deviations                                                                                                    |         |                                                                            |
| 441 | (RSD) from stock solutions are reported in Table 3 Table 1 .                                                                                                                          |         | Formatted: Font: Not Bold                                                  |
| 442 | The fourth aliquot was used to measure dissolved ammonia (NH4 + ) via a colorimetric                                                                                              |         |                                                                            |
| 443 | method similar to that presented by Gieskes et al. (1991) This was earried out shipboard via a                                                                                               |         |                                                                            |
| 444 | colorimetric method similar to that presented by Gieskes et al. (1991). Set volumes ( $100\mu$ L) of                                                                                         |         |                                                                            |
| 445 | pore water were pipetted into 1-em 3 -plastie-cuvettes and diluted with 900 µL of with milli-Q                                                                                    |         |                                                                            |
| 446 | water. Two reagents (100 $\mu$ L of A and 100 $\mu$ L of B) were then pipetted into the cuvettes.                                                                                            |         |                                                                            |
| 447 | Reagent A was prepared by adding- Na3C6H5O7, C6H5OH, and Na2(Fe(CN)5NO) to <del>35 g of</del> |         |                                                                            |
| 448 | trisodium citrate (Na 3 C 6 H 5 O 7 ), 2.7 g of phenol (C 6 H 5 OH), and 0.06 g of sodium nitroprusside                    |         |                                                                            |
| 449 | (Na₂[Fe(CN)₅NO]) to 100 mL of milli-Q water. Reagent B was prepared by NaOH in milli-Q                                                                                                       |         |                                                                            |
| 450 | water and adding NaClO solution. dissolving 1.36 g of sodium hydroxide in 100 mL of milli-Q                                                                                                  |         |                                                                            |
| 451 | water and adding 3 mL sodium hypochlorite (NaClO) solution. After the reagents were added,                                                                                                   |         |                                                                            |
| 452 | sSolutions were mixed, and allowed to react for at least six but not more than 24 hours. Solutions                                                                                           |         |                                                                            |
| 453 | turned various shades of blue, which to relate to $\mathrm{NH_4^+}$ concentration, and which were measured                                                                                   |         |                                                                            |
| 454 | by absorbance at 630 nm on a Hitachi U-1100 spectrophotometer. Five point calibration curves                                                                                                 |         |                                                                            |
| 455 | (0 to 200 µM) were measured before each sample set and corrected using VKI standard (QC                                                                                                      | 1       | Formatted: Default Paragraph Font, Font: Calibri, 11 pt, Pattern: Clear    |
| 456 | RW1; www.eurofins.dkwww.eurofins.dk; Table 31).                                                                                                                                              | 1       | Formatted: Default Paragraph Font, Font: Calibri, 11 pt,
Pattern: Clear |
| I   |                                                                                                                                                                                              |  | Formatted: Font: Not Bold                                                  |

| 4 | 57 | The fifth aliquot was used to measure dissolved phosphate (PO4 3- ) following the method                                                                       |                     |
|---|----|---------------------------------------------------------------------------------------------------------------------------------------------------------------------------|---------------------|
| 4 | 58 | given by Gieskes et al. (1991). The method of preparation also followed that given by Gieskes et                                                                          |                     |
| 4 | 59 | al. (1991). The remainder of the Remaining pore water (generally between 1 and 3mL) was                                                                                   |                     |
| 4 | 60 | added to milli-Q water to a sum of 10 mL. Two reagents were then-added to the solution to react                                                                           |                     |
| 4 | 61 | with PO43-phosphate (200 <math>\mu</math>L of A and B) . Reagent A was prepared by first-making three                                        |                     |
| 4 | 62 | solutions: (NH4)2MoO4, H2SO4, and C8H4K2O12Sb2 • XH2O were added to milli-Q water, and                                                                                    |                     |
| 4 | 63 | the solutions were added dropwise.eight grams of ammonium molybdate ((NH4)2MoO4) were                                                                                     |                     |
| 4 | 64 | added to 80 mL of milli Q water, 50 mL of concentrated sulfuric acid were added to 150 mL of                                                                              |                     |
| 4 | 65 | milli-Q water, and 0.01 g of potassium antimonyl tartrate hydrate (C 8 H 4 K 2 O 12 Sb 2 • XH 2 O) were |                     |
| 4 | 66 | added to 10 mL of milli-Q water. Then, 30 mL of the ammonium molybdate solution were added                                                                                |                     |
| 4 | 67 | to 90 mL of the sulfurie acid solution, and five mL potassium antimonyl tartrate solution was                                                                             |                     |
| 4 | 68 | slowly added dropwise. Reagent B was created with $C_6H_8O_6$ by dissolving 10 g of ascorbic acid                                                                         |                     |
| 4 | 69 | in 50 mL of milli-Q water. After the samples were prepared, reagent A and B were added,                                                                                   |                     |
| 4 | 70 | mixed, and allowed to react for 10 but not more than 30 minutes. Solutions turned various shades                                                                          |                     |
| 4 | 71 | of blue, which to relate realting to PO4 3- concentration, and which were then measured at an                                                           |                     |
| 4 | 72 | absorbance of 880 nm-on the above spectrophotometer. Five point calibration curves (0 to 50                                                                               |                     |
| 4 | 73 | HM) were measured before each sample set and corrected using VKI standard (QC RW1;                                                                                        |                     |
| 4 | 74 | www.eurofins.dk www.eurofins.dk; Table 31 ).                                                                                                                |
Formatt         |
| 4 | 75 | For 352 pore water samples, a sixth aliquot of approximately 2 mL could be collected to                                                                                   | Formatt             |
| 4 | 76 | mix with 200 $\mu$ L of a 2.5% Zn-acetate (Zn(C 2 H 3 O 2 ) 2 ) solution. Given the extremely low                             | Pattern:
Formatt |
| 4 | 77 | solubility of ZnS, when such a solution is added to pore water samples, a white precipitate                                                                               |                     |
| 4 | 78 | should form in the presence of even very low H 2 S concentrations (Cline, 1969; Goldhaber,                                                                     |                     |
| 4 | 79 | 1974).                                                                                                                                                             |                     |
|   |    |                                                                                                                                                                           |                     |

| 480 | A method detection limit (MDL) for each species can be determined by the following                                                                                                          |       |
|-----|---------------------------------------------------------------------------------------------------------------------------------------------------------------------------------------------|-------|
| 481 | equation:                                                                                                                                                                                   |       |
| 482 | $\underline{\qquad}MDL = \left(\frac{C_{High} - C_{Low}}{I_{High} - I_{Low}}\right) 3\sigma_{}, \tag{6}$                                                                                    |       |
| 483 | where C = concentration and I = intensity (counts per second on the ICP-AES). The MDLs were                                                                                                 |       |
| 484 | as follows: Ba = 0.01 <math>\mu</math>M, Ca = 0.08 <math>\mu</math>M, Fe = 5.9 <math>\mu</math>M, Mg = 0.22 <math>\mu</math>M, Mn = 0.24 <math>\mu</math>M, S = 1.2                  |       |
| 485 | $\mu$ M, Sr = 0.01 $\mu$ M. On all plots, for reference, we place dashed lines for values of IAPSO                                                                                          |       |
| 486 | seawater standard (Alkalinity = 2.33 mM, Ba = 0.00 mM, Ca = 10.28 mM, Fe = 0.00 mM, Mg =                                                                                                    |       |
| 487 | $\underline{53.06 \text{ mM}, \text{ Mn} = 0.00 \text{ mM}, \text{ S} = 28.19 \text{ mM}, \text{ Sr} = 0.09 \text{ mM}, \text{ NH}_4 = 0.00 \text{ mM}, \text{ HPO}_4 = 0.00 \text{ mM}).}$ |       |
| 488 | In cases of excess sample, an additional aliquot was collected to test for dissolved hydrogen                                                                                               |       |
| 489 | sulfide. Approximately 2 mL of pore water was placed into a cryovial, and 200 $\mu$ L of a 2.5% Zn-                                                                                         |       |
| 490 | acetate $(Zn(C_2H_3O_2)_2)$ solution was added. Given the extremely low solubility of ZnS, a white                                                                                          |       |
| 491 | precipitate should form in the presence of even very low $H_2S$ concentrations (Cline, 1969;                                                                                                |       |
| 492 | Goldhaber, 1974).                                                                                                                                                                           |       |
| 493 |                                                                                                                                                                                             |       |
| 494 | 4. Results                                                                                                                                                                                  |       |
| 495 | 4.1 GeneralitiesBroad conclusions                                                                                                                                                           |       |
| 496 | With the large number of pore water measurements (Table 1 Tbl. S1) we begin with some                                                                                                       | Forma |
| 497 | generalities regarding-the results. We plot pore water concentration profiles along each transect                                                                                           |       |
| 498 | collectively (Fig. 4-8Figures 6 - 10), irrespective of coring device or water depth, although clear                                                                                         |       |
| 499 | variance in pore water chemistry exists between stations for some dissolved species (e.g., Fe).                                                                                             |       |
| 500 | Most species display "smooth" concentration profiles with respect to sediment depth (Fig.                                                                                                   |       |
| 501 | 4-8 Figures 6 – 10). That is, concentrations of successive samples do not display a high degree of                                                                                   |       |
| 502 | scatter. This is expected for pore water profiles in sediment where diffusion dominates (Froelich                                                                                           |       |

| 503 | et al., 1979; Klump and Martens 1981; Schulz, 2000). However, for someas best seen for                                                                                     |
|-----|----------------------------------------------------------------------------------------------------------------------------------------------------------------------------|
| 504 | dissolved species whose concentrations do not appreciably change over depth (e.g., Ba2+ and                                                                                |
| 505 | Ca 2+ ), scatter exists beyond that predicted from analytical precision. We discuss this in detail in                                                           |
| 506 | the supplementary information.concentrations do not appreciably change over depth (e.g., Ba2+                                                                              |
| 507 | and Ca 2+ ) seatter exists beyond that predicted from analytical precision. This scatter has a weak                                                             |
| 508 | positive correlation with increased sampling time, which can be shown by comparing time to a                                                                               |
| 509 | deviation in concentration (Figure 11). The latter is defined by:                                                                                                          |
| 510 | $\Delta X = X_{Measured} - X_{Predicted}, (6)$                                                                                                                             |
| 511 | where X is the species of interest, and $X_{Predicted}$ is the concentration of X determined from the                                                                      |
| 512 | linear best fit line of a concentration profile.                                                                                                                           |
| 513 |                                                                                                                                                                            |
| 514 | equation:                                                                                                                                                                  |
| 515 | $- MDL = \left(\frac{c_{\mu tgh} - c_{Low}}{t_{\mu tgh} - t_{Low}}\right) 3\sigma ,  (7)$                                                                                  |
| 516 | where C = concentration and I = intensity (counts per second on the ICP-AES). The MDLs were                                                                                |
| 517 | <del>as follows: Ba-= 0.01 µM, Ca-= 0.08 µM, Fe-= 5.9 µM, Mg-= 0.22 µM, Mn-= 0.24 µM, S = 1.2</del>                 |
| 518 | $\mu$ M, Sr = 0.01 $\mu$ M. On all plots, for reference, we place dashed lines for values of IAPSO                                                                         |
| 519 | seawater standard (Alkalinity = 2.33 mM, Ba = 0.00 mM, Ca = 10.28 mM, Fe = 0.00 mM, Mg =                                                                                   |
| 520 | $53.06 \text{ mM}, \text{Mn} = 0.00 \text{ mM}, \text{S} = 28.19 \text{ mM}, \text{Sr} = 0.09 \text{ mM}, \text{NH}_4 = 0.00 \text{ mM}, \text{HPO}_4 = 0.00 \text{ 
[revised manuscript text omitted]

 $\underline{depth} (-\Delta Alkalinity / \Delta S) \text{ is } 0.98 \text{ (Fig. 9a)}. \\ \underline{increase to sulfate decrease} (-\Delta Alkalinity / \Delta S) \\ \underline{is } 0.98 \text{ (Fig. 9a)}. \\ \underline{increase to sulfate decrease} (-\Delta Alkalinity / \Delta S) \\ \underline{is } 0.98 \text{ (Fig. 9a)}. \\ \underline{increase to sulfate decrease} (-\Delta Alkalinity / \Delta S) \\ \underline{is } 0.98 \text{ (Fig. 9a)}. \\ \underline{increase to sulfate decrease} (-\Delta Alkalinity / \Delta S) \\ \underline{is } 0.98 \text{ (Fig. 9a)}. \\ \underline{increase to sulfate decrease} (-\Delta Alkalinity / \Delta S) \\ \underline{is } 0.98 \text{ (Fig. 9a)}. \\ \underline{increase to sulfate decrease} (-\Delta Alkalinity / \Delta S) \\ \underline{is } 0.98 \text{ (Fig. 9a)}. \\ \underline{increase to sulfate decrease} (-\Delta Alkalinity / \Delta S) \\ \underline{is } 0.98 \text{ (Fig. 9a)}. \\ \underline{increase to sulfate decrease} (-\Delta Alkalinity / \Delta S) \\ \underline{increase to sulfate decrease} (-\Delta Alkalinity / \Delta S) \\ \underline{increase to sulfate decrease} (-\Delta Alkalinity / \Delta S) \\ \underline{increase to sulfate decrease} (-\Delta Alkalinity / \Delta S) \\ \underline{increase to sulfate decrease} (-\Delta Alkalinity / \Delta S) \\ \underline{increase to sulfate decrease} (-\Delta Alkalinity / \Delta S) \\ \underline{increase to sulfate decrease} (-\Delta Alkalinity / \Delta S) \\ \underline{increase to sulfate decrease} (-\Delta Alkalinity / \Delta S) \\ \underline{increase to sulfate decrease} (-\Delta Alkalinity / \Delta S) \\ \underline{increase to sulfate decrease} (-\Delta Alkalinity / \Delta S) \\ \underline{increase to sulfate decrease} (-\Delta Alkalinity / \Delta S) \\ \underline{increase to sulfate decrease} (-\Delta Alkalinity / \Delta S) \\ \underline{increase to sulfate decrease} (-\Delta Alkalinity / \Delta S) \\ \underline{increase to sulfate decrease} (-\Delta Alkalinity / \Delta S) \\ \underline{increase to sulfate decrease} (-\Delta Alkalinity / \Delta S) \\ \underline{increase to sulfate decrease} (-\Delta Alkalinity / \Delta S) \\ \underline{increase to sulfate decrease} (-\Delta Alkalinity / \Delta S) \\ \underline{increase to sulfate decrease} (-\Delta Alkalinity / \Delta S) \\ \underline{increase to sulfate decrease} (-\Delta Alkalinity / \Delta S) \\ \underline{increase to sulfate decrease} (-\Delta Alkalinity / \Delta S) \\ \underline{increase to sulfate decrease} (-\Delta Alkalinity / \Delta S) \\ \underline{increase to sulfate decrease} (-\Delta Alkalinity / \Delta S) \\ \underline{increase to sulfate decrease} (-\Delta Alkalinity / \Delta S) \\ \underline{increase to sulfate decrease} (-\Delta Alkalinity / \Delta S) \\ \underline{increase to sulfate decrease} (-\Delta Alkalini$

| 617 | NH4 gradients are as follows: $Tr1 = 43.0 \ \mu\text{M/m}$ , $Tr2 = 17.4 \ \mu\text{M/m}$ , $Tr3 = 69.0 \ \mu\text{M/m}$ , and $Tr4 = 10.0 \ \mu\text{M/m}$ |
|-----|-------------------------------------------------------------------------------------------------------------------------------------------------------------------------------|
| 618 | 29.0 µM/m.                                                                                                                                                             |
| 619 | By contrast, concentrations of dissolved HPO4 2- in our cores typically increase, reach a                                                                          |
| 620 | subsurface maximum, and then decrease (Fig. 4-8). With available data, a more pronounced                                                                                      |
| 621 | maximum generally occurs at stations with relatively shallow water depth. For example, consider                                                                               |
| 622 | the peak in HPO4 2- concentrations at four stations. At the two shallow stations, S12 (384 m) and                                                                  |
| 623 | S22 (367 m) the HPO 4 2- maxima are, 73 $\mu$ M (1.91 m) and 18 $\mu$ M (0.66 m), respectively, but at                                                  |
| 624 | the two deeper stations, S17 (977 m) and S14 (733 m), the HPO 4 2- maxima are only 6.7 $\mu$ M (1.76                                                    |
| 625 | m) and 7.1 µM (2.33 m) respectively. The station on Lomonosov Ridge (S31) has a high in                                                                                       |
| 626 | $\underline{\text{HPO}_4^{2-}}$ concentration of 76 $\mu$ M at 1.02 m below the mudline. In general, stations with more                                                       |
| 627 | pronounced HPO 4 2- maxima also have greater increases in alkalinity with depth.                                                                        |
| 628 | The $NH_4^+$ , $HPO_4^{2^-}$ , and alkalinity profiles relate to one another statistically, although with                                                                     |
| 629 | distinction. All stations have a C:N ratio in pore waters much higher than the canonical Redfield                                                                             |
| 630 | Ratio of 6.625 (Fig. 10). Rather, the concentration relationship of alkalinity and ammonium ion                                                                               |
| 631 | can be expressed by a second order polynomial ( $[NH_4^+] = -0.003[Alk]^2 + 0.105[Alk] - 0.253$ ;                                                                             |
| 632 | Fig. 9b) with an average molar ratio ( $\Delta Alk/\Delta NH_4^+$ ) of 14.7, close to what might be expected for                                                              |
| 633 | degradation of terrestrial organic carbon. Interestingly, this ratio deviates somewhat across                                                                                 |
| 634 | transects, increasing at sites from Tr1, Tr3, Tr2, to the Lomonosov Ridge station. The molar ratio                                                                            |
| 635 | of alkalinity to phosphate ion ( $\Delta$ Alk/ $\Delta$ HPO 4 2- ) averages 55.7 for all stations. This ratio also                                      |
| 636 | generally increases in cores from east to west. With depth, concentrations of dissolved HPO42-                                                                                |
| 637 | typically increase, reach a subsurface maximum, and then decrease (Figure

---

## Author Response (AR2)

Response to associate editor second request for major revisions

Pore water geochemistry along continental slopes north of the East Siberian Sea: Inference of low methane concentrations; MS No.: bg-2016-308; Clint M. Miller et al.

We begin with a synopsis of changes to the original MS (Line numbers in parentheses).

| Original MS | 2nd Revised MS |
|---|---|
| 1. Introduction (65 Lines) | 1. Introduction (59) |
| 2. Background (1) | 2. Background (1) |
| 2.1 East Siberian Margin Geology (25) | 2.1 East Siberian margin geology (18) |
| 2.2 Regional Oceanography (19) | 2.2 Regional oceanography (11) |
| 2.3 Current Speculation on Gas Hydrates in the Arctic (36) | 2.3 Current speculation on gas hydrates in the Arctic (28) |
| 2.4 Pore Water Chemistry Above Methane-Charged Sediment Sequences (57) | 2.4 Pore water chemistry above methane-charged sediment (82) |
| 3. Materials and Methods (1) | 3. Materials and Methods (1) |
| 3.1 The SWERUS-C3 Expedition, Leg 2 (19) | 3.1 SWERUS-C3 Expedition, Leg 2 (14) |
| 3.2 Core material (12) | 3.2 Core material (11) |
| 3.3 Interstitial Water Collection (42) | 3.3 Interstitial water collection (18) |
| 3.4 Interstitial Water Analyses (66) | 3.4 Interstitial water analyses (59) |
| 4. Results (1) | 4. Results (1) |
| 4.1 Generalities (26) | 4.1 General observations (12) |
| 4.2 Porosity and Sampling Time (13) | |
| 4.3 Physiochemical Conditions During Rhizon Sampling (19) | |
| 4.4 Alkalinity and δ13C (17) | 4.2 Alkalinity and δ13C of DIC (15) |
| 4.5 Sulfur and sulfate (13) | 4.3 Sulfur and sulfate (13) |
| 4.6 "Nutrients": Phosphate and Ammonia (38) | 4.4  Ammonia and phosphate (35) |
| 4.7 Metals (22) | 4.5 Metals (22) |
| 5. Discussion (1) | 5. Discussion (1) |
| 5.1 Flow Rates from Rhizons (6) | |
| 5.2 Fidelity of Rhizon Pore Water Measurements (63) | 5 .1 Fidelity of rhizon pore water measurements (21) |
| 5.3 Reading the Pore Water Profiles (42) | |
| 5.4 General Absence of Methane (39) | 5.2 General absence of methane (36) |
| 5.5 Special Case "Lomonosov Ridge Station" (19) | 5.3 Special case: Lomonosov Ridge station (20) |
| 5.6 Other Chemistry (19) | 5.4 Other chemistry (25) |
| 5.7 Signatures of AOM and Organoclastic Sulfate Reduction (39) | 5.5 Signatures of AOM and OSR (31) |
| 5.7 Explanations (42) | 5.6 Possible explanations for widespread absence of gas hydrate and methane (66) |
| 6. Conclusions (22; Total = 784) | 6. Conclusions (18; Total = 617) |

Specific changes requested by editor:

Referee/Editor Comments (*italic font*) with direct responses (**bold font**).

*I therefore return the ms to you and ask that you revise the response letter with clear reference to where in the revised ms the relevant new /revised information can be found (page and line number). It would also facilitate the handling if you next to the clean ms would also provide a ms file where revised/new text is highlighted. Please then also describe clearly in the response letter what the changes are encompassing.*

**The table above provides a clear comparison where major structural changes occurred. In addition to a new "tracked changes" document, the authors include a highlighted MS with explanatory comments (and one without). This response letter provides line numbers of both the original MS and the 2nd revised MS (Note that the 1st revised MS is not generally discussed to avoid unnecessary confusion).**

*In preparing these new files, I ask you to consider the following points for the revisions:*
*1. The geographical scope of the study. This is a reminder from the last round. The title still reads "north of Siberia". The margin north of Siberia spans roughly between 67E-180E (Kara Sea, Laptev Sea and East Siberian Sea). Your cores are roughly from 140E-180E, which is the East Siberian Sea.*

**The new title is "Pore water geochemistry along continental slopes north of the East Siberian Sea: Inference of low methane concentrations." The geographical scope of this study is referred to as "Slopes north of the East Siberian Sea" abbreviated "SNESS." Original MS revised line numbers are: 16, 18, 21, 24, 34, 69, 70, 92, 96, 117, 119, 126, 130, 136, 153, 157, 158, 162, 165, 182, 248, 253, 476, 694, 728, 758, 765, 772, 774, 777, 780, 785, 788, 801, 804, 808, 824, 827, 835, and 838. Germane revised lines in the new MS are: 18, 20, 22-24, 36-37, 71, 73, 92, 95, 113-118, 121, 133, 139, 143, 246, 389, 509, 540, 586, 591, 602, 619, 626, 630, 632, 647, 656, 661, and 664. The Supplementary Information does not discuss geographical extent.**

*2. The inference of dissolved methane concentrations*

*Based on reviewer comments and my own assessment, I also ask you to revise both title and text throughout the ms to reflect that (a) porewater methane is inferred (not directly measured);*

**Both the original and revised MSs refer to pore water methane as "inferred." Abstract lines 24-25 state methane flux is, "inferred from profiles of dissolved sulfate ($SO_4^{2-}$), alkalinity, and the δ13C of dissolved inorganic carbon (DIC)." This is explained in great detail in Section 2.4 (Pore water chemistry above methane-charged sediment) particularly lines 188-231. Additionally, lines 478-489 and 605-617 further explain the details of inference. Indeed, the title of the MS states that methane is inferred.**

*(b) the inferred methane refers to dissolved methane. The latter point is particularly salient as there are ample studies demonstrating bubble-mediated transport of methane from Arctic sediments (i.e., ebullition). As stated in a recent review (James et al., 2016, L&O) "methane gas can bypass microbially mediated oxidation reactions because microbes can only access dissolved methane". I ask you to revise your ms to clearly recognize this aspect.*

**We agree that this in an important point and wish to clarify any confusion. As lines 618-622 explain, "No seafloor features indicative of seafloor CH4 expulsion were found during the bathymetric mapping of SNESS. Nonetheless, it is possible that local CH$_4$ venting, perhaps related to and mediated by bubble transport, could occur away from transects and cores of SWERUS Leg 2. Certainly, the chemistry of advecting fluids toward seafloor features such as mud volcanoes and cold seeps typically differs from the much broader surrounding region."**

**However, within cored locations, neither free gas nor dissolved methane can be inferred. This concept has been vetted in the literature extensively. We continue in lines 624-628, "However, in such cases, even the encompassing area typically has shallow SMTs. Without invoking odd geology, such as an extensive impermeable layer, it is unlikely that significant CH$_4$ exists in shallow sediment across much of SNESS, including as gas hydrate or free gas.**

Here it is stressed that neither gas hydrate nor free gas can exist in sediment on continental slopes without high concentrations of dissolved gas in surrounding pore water."

The passage of free gas through sediment leaves many traces in pore water. Unfortunately, the paper and key figure in James et al. (2006) does not fully convey this concept. Pore waters in sediment where methane also occurs in gas hydrate or free gas phase must be saturated with methane in the dissolved phase, except in some bizarre (perhaps only theoretical) environment where advection is so fast that basic thermodynamics does not apply. This comes from basic physical chemistry and can be readily realized by looking at phase diagrams for methane-water systems. There is a good reason why we state the above, and why to our knowledge this has never been found. One should not invoke methane bubbling up through sediment where surrounding pore waters have little to no methane.

Examples of this concept are found throughout the literature including: Borowski et al., 2001; Aharon and Fu, 2000, 2002; Luff and Wallmann, 2003; Torres et al., 2004; Joye et al., 2004; Claypool et al., 2006; Coffin et al., 2007; Kastner et al., 2008; Hiruta et al., 2009; Hu et al., 2010; Coffin et al., 2014; Hu et al., 2015; and Geprägs et al., 2016.

*I agree with both reviewers that your paper can be substantially reduced in length (by up to 1/3-1/2).*

The revised MS is significantly shorter than the original. The above table shows that three sections (4.2 Porosity and Sampling Time; 4.3 Physiochemical Conditions During Rhizon Sampling; 5.3 Reading the Pore Water Profiles) have been deleted from the MS. The text alone is shorted 22%. Additionally, 4 figures, 2 tables, and 23 references have been deleted.

**Specific comments from Referee 2:**

*Figures 1 and 3: I suggest placing both maps next to each other in one figure (ie.,Fig.1a and 1b). This would make it much easier for the reader to find out where the sampling sites are located relative to predicted gas hydrate occurrence. Figure 4: This is a nice picture but does not convey any important information. Given the total number of figures in this MS, I suggest deleting it.*

**The authors agree with both points, however combining Figures 1 and 3 so that both are readable is challenging. Therefore, the caption has been imbedded in Figure 1, and the symbol description of Figure 3 (now Figure 2) is in prose. Figure 4 has been deleted from the MS.**

*Figure 6-9: These figures are hard to read. I suggest plotting each core in a specific figure in a different color rather than all data in one figure in the same color. There is very limited discussion/comparison of the ACEX data; why plot it then?*

**We regret these figures are difficult to read. Given the extremely large dataset over this vast region, it is difficult to clearly present results concisely. It seemed to us somewhat overwhelming to plot the chemistry at every site independently, or alternatively every species analyzed at multiple sites independently. The authors have tried many different plotting methods including plotting each core in a different color. This style did not improve figure readability, and removes the color distinction carried over into Figure 8 (which is the most important figure in this group). Instead, we have chosen to increase panel, symbol, and line widths while minimizing white space. The ACEX data has been removed, as suggested, and the legends are imbedded within the panels. Hopefully, this improves readability without lengthening the paper.**

*Lines 187-242 vs. lines 616-657: The sections are basically saying the same thing with a few additional points in the latter, discussion section. I suggest removing lines 616-657 and taking the few "new points" that are mentioned here and adding them to the background section. I found it tiring to read the same "intro to reading pore-water profiles" twice.*

**The entire section 5.3 "Reading the Pore Water Profiles" has now been removed. All concepts in 187-242 and 616-657 of the original MS are condensed into lines 162-242 of the revised MS.**

*"Rhizone experiments": These are very helpful experiments that install additional confidence in this comparably novel sampling technique. With that being said the description of these experiments, including the results and discussion of the results take up a lot of space and distract from the main story of the MS. I suggest moving all of this into a supplementary material section. This would include the experiment description (line 310), section 4.3, the discussion sections 5.1 and 5.2 and Figures 5 and 11 (and maybe 12 if the authors think that the porosity-rhizone aspect could also be trimmed), Tables 1 and 2. Basically, all we need to know is what is in the short summary in lines 606-614. The reader can be referred to the supplementary material for the detailed experiments.*

**Rhizons have been subject to debate leading to some misunderstanding in their applicability to marine settings. The authors believe these experiments provide some much needed clarification to Rhizon sampling fidelity, but we agree with the reviewer that this section distracts from the primary purpose to the MS. Therefore, lines 310-315, sections 4.2, 4.3, 5.1, 5.2, Figures 5, 11, 12, Tables 1, and 2 have been edited and moved to supplementary materials. These concepts are briefly covered in lines 455-475 of the revised MS.**

*Dissolved hydrogen sulfide "analysis": It seems like the authors did not actually do any sulfide analyses but just "visually" observed whether white precipitates were forming when zinc acetate was added. To me this is not an appropriate "analysis" to detect hydrogen sulfide. This is especially important since the authors did not do any sulfate analyses but only analyzed total dissolved sulfur and based on their visual "analysis" of the sampling vials- assumed that no hydrogen sulfide was present and the total sulfur only reflects sulfate. I strongly suggest doing at least a few hydrogen sulfide analyses with the Cline method, for example of the samples from deeper layers especially on the cores from Lomonosov Ridge, to confirm the absence of hydrogen sulfide.*

**The term "analysis" may be confusing. This section has been reworded to describe "visual inspection" of ZnS precipitate (Revised MS Lines 29 and 382). Unfortunately, pore water sulfide analyses are not possible.**

*Lines 176-178- Microbial processes at cold seafloor temperatures: I disagree with the authors here. There are plenty of studies that have shown that organic carbon turnover rates or "bacterial degradation" in high latitude environments are/can be as high as in mid-latitude or tropical environments. For example: Glud et al., 1998: Benthic mineralization and exchange in Arctic sediments (Svalbard, Norway) Arnosti et al., 2005: Anoxic carbon degradation in Arctic sediments: Microbial transformations of complex substrates.*

**The original MS did not state that bacterial degradation is lower in high latitudes than lower latitudes. Rather, line 176 states that burial "might" be enhanced by colder temperatures. This idea is quite logical given our understanding of bacterial processes at different temperatures, and has been discussed in the literature previously (Ex. Darby et al., 1989; Max and Lowrie, 1993). However, this is idea is conjecture and has been removed from the revised MS.**

*Carbon isotope sections: Generally, the sections discussing the carbon isotope system, e.g., processes associated with carbon isotope fractionation, the discussion of the carbon isotope data etc. is very weak and needs more clarification. Also, it is incorrect to present equations (1) and (8) with 12C and state that it indicates "depletion in $^{13}C$". As such, the equations written just present the reaction of one organic molecule containing 12C to bicarbonate which of course also has to contain 12C. Please take the notations out.*

**The subscript notations have been removed from both equations (82 and 503). See the following comment regarding improving the carbon isotope discussion.**

*Line 227-229: This needs to be expanded and maybe clarified. Both the Holler and the Yoshinaga references are discussing carbon isotope fractionation during AOM. As stated here,*

*the authors only consider the original [13]C-depleted value of the CH[4] in explaining the light DIC formed. Additionally consider: Alperin, M.J., Reeburgh, W.S., Whiticar, M.J., 1988. Carbon and hydrogen isotope fractionation resulting from anaerobic methane oxidation. Glob. Biogeochem. Cycles 2, 278–288. Martens, C.S., Albert, D.B., Alperin, M.J., 1999. Stable isotope tracing of anaerobic methane oxidation in the gassy sediments of Eckernforde Bay, German Baltic Sea. Am. J. Sci. 299, 586–610. And for the first part, asides from Paull et al., a reference such as Whiticar, M.J., 1999. Carbon and hydrogen isotope systematics of bacterial formation and oxidation of methane. Chem. Geol. 161, 291–314.*

**We consider DIC $^{13}$C depletion to result from a variety of factors in CH$_4$ charged sediments including: fractionation during AOM, fractionation during organoclastic sulfate reduction (OSR) and other bacterially mediated reactions, differential diffusion of $^{12}$CH$_4$ and $^{13}$CH$_4$ from deep sediments, as well as the light CH$^4$ input from below. The authors thought this was clear, however lines 202-208 has been reworded to clarify.**

*Line 681-687: Similar to the previous carbon isotope section, there is some more detail needed here. For example, carbon isotope fractionation during organoclastic sulfate reduction needs to be discussed. The Chatterjee reference (which should be 2011 not 2001) is insufficient here.*

**Indeed, the authors interpret the observed $^{13}$C depletion as fractionation during OSR and other bacterially mediated reactions. Lines 496-503 provide extended detail.**

*Line 706: "almost necessarily implies CH$_4$ oxidation.. ". This statement needs an explanation and the appropriate literature. . .*

**The $\delta^{13}$C-DIC values are comparable to a great many published results from CH$_4$ charged sediments. Additionally, these results imply CH$_4$ oxidation because no other process can realistically create <-40‰ $\delta^{13}$C-DIC values. Lines 496-503 provide extended detail.**

*Results section: When you list what the concentrations were, they are in past tense, when you describe what the reader sees in the graph, this is in present tense.*

**Discussion of concentrations are now in present tense (353-379, 382-393, 403-419, and 432-452)**

*Lines 508-519, Figure 14: This is a nice exercise but I am wondering why this is included? I could not find any reference to this approach/figure in the discussion section. If it is not relevant to your discussion-delete! Or add a section in the Discussion part that evaluates the plot.*

**Deviations from the Redfield ratio in marine environments may be caused by different organic matter sources (terrigenous?) than primary productivity. Given this MS's results differ markedly from previously assumptions regarding past organic matter turnover; this exercise seems particularly germane! These results, however, are not enough by themselves to show organic source, but simply imply the terrigenous component may be important. Lines 420-429 have been rewritten, and a short section (556-559) has been added in the discussion section to explain this figure more completely.**

*Lines 728-733 and elsewhere: I disagree with this general interpretation. Many of the collected cores also show decreases in sulfur concentration which point to the occurrence of organoclastic sulfate reduction, and you interpret the delta$^{13}$C-DIC profiles as being imprinted by this process! While the dissolved Mn profiles can be interpreted as reflecting dissimilatory manganese oxide reduction, there has been a lot of recent work discussing the –somewhat intriguing- manganese biogeochemistry of Arctic Ocean sediments, including evidence for dissolved manganese profiles reflecting diagenetic remobilization of Mn and diffusion from deeper sediment intervals. I suggest preparing this section with more caution. For reference: März et al. 2011: Manganese rich brown layers in Arctic Ocean sediments: Composition, formation mechanisms, and diagenetic overprint (and references therein).*

**The first author was unaware of März et al. (2011), and thanks the reviewer for this comment. Indeed, the reviewer is correct the Mn profiles in this MS may be partially affected by diagenetic remobilization of Mn below the sampled intervals. The above section (543-549) has been altered to discuss this possibility.**

*Line 735-737: This section is somewhat incorrect as well. What Mn and Fe is consumed? I assume you are now referring to Mn- and Fe-oxides. I suggest: 1) making it clear that dissolved Mn and Fe are produced during dissimilatory Mn- and Feoxide reduction; 2) highlighting that the reason for the decline in concentrations are consumption processes (assuming steady state you would otherwise expect constant pore-water values below the current reaction zone), which likely include the reaction of Fe with hydrogen sulfide, and interactions of Fe with Mn-oxides. (Again the sedimentary Mn story may be more complicated; see comment above); 3) stepping back from the idea that there is "complete consumption of Fe and Mn". If you are referring to the oxides, then especially in the case of Fe it is the very reactive (towards H2S) iron (oxyhydr)oxide phases that are being reduced (see Canfield et al., 1992: The reactivity of sedimentary iron minerals towards sulfide) but there is without a doubt no "complete Fe consumption"!*

**See the above comment response. This section (543-549) has been reworded to reflect the possible importance of Mn remobilization in these sediments. The sentence on "complete" consumption (Lines 735-736 of original MS) has been removed.**

*Section 5.7 and Figure 16: In this form, I find the plot misleading and somewhat irrelevant (or not providing any new helpful information). First, as you have discussed, sites with methanogenesis and AOM are characterized by much higher DIC concentrations and much lighter delta$^{13}$C-DIC values than sites lacking these processes. If you multiply these two, of course you get more negative values at the AOM sites. Second, I am not sure what you are actually plotting as ˆDIC here ? You state that other authors have used the concentrations at the seafloor and the SMT. What do you do for your data where there is no SMT? Third, in line 760 you state "two basic models help explain the relationships in Figure 16." However, you are in the following section only discussing the C:S ratios, including their relative changes with depth (as you are interpreting them from the mudline downward using the changes in DIC\*delta$^{13}$C-DIC as an alternative measure for depth). Why then do such a crossplot? On a side note – why is the ratio for the OSR model increasing past 2:1? Because the DIC reflects additional bicarbonate*

*production by dissimilatory Mn and Fe oxide reduction rather than only from sulfate reduction ? Fourth, in line 747 you are stating that "a flux of HCO3- from below the SMT can augment the DIC produced. . .Thus, changes in alkalinity relative to sulfate often exceed 1:1. . .". Now the conclusion from your model/plot is that -line 768-769-"..CH4 charged locations with migrating DIC must have C:S molar ratios in excess of 1:1. . .". So what have we learned? It would be honest to also mention the studies by Snyder et al., 2007 and Wehrmann et al. 2011 (Coupled organic and inorganic carbon cycling in the deep subseafloor sediment of the northeastern Bering Sea Slope (IODP Exp. 323)) in lines 740-750 who used fluxes instead of concentrations.*

**As quoted above, line 748 in the original MS discusses the upward DIC flux common to sites with high CH4 concentrations. This flux is often ignored, but has been shown to broadly affect both solute concentrations and isotopic values in CH4 charged sediments (Dickens and Snyder, 2009; Chatterjee et al., 2011). Improved lines 562-584 clarify the signatures of AOM and OSR for which this flux is a strong component. Importantly, the x-axis of Figure 16 is NOT simply a result of high DIC concentrations, but rather shows a very large continuum of values in supposedly similar environments which do not follow the 1:1 and 2:1 ratios many authors use. Additionally, plotting the sites from this MS versus locations with high CH4 flux clearly juxtaposes our results.**

**Specifically from above: The DIC question is irrelevant because all locations other than results from this MS have SMTs. The two models were intended to expand the above concept, but appear to be confusing. We are therefore, removing them from the MS. The Snyder and Wehrmann statement and references have been added (566-567)**

*Lines 808-816: I suggest expanding this section, and maybe including relevant literature to support the different hypotheses, even if it means speculating. The finding that CH4 is low in the sediment in this part of the Arctic is the essential message of this study; the major question that arises is why? Do the ACEX studies provide any clues that would support any of your hypotheses? Lines 817-820 need more details and references as well! Discussion section: the Ba and Sr data are not discussed.*

**This section (632-649) has been expanded and additional references are added. Conclusions from ACEX are discussed in lines 652-654.**

*Technical corrections:*
*Line 17:. . .methane (CH$_4$). . .*

**Fixed (19)**

*Line 27: replace "nutrient" with "phosphate and ammonium". . .Also, the "nutrient data" does not provide evidence for the dominance of metal oxide reduction but evidence for very low organic carbon turnover rates. Please re-phrase.*

**Fixed (30)**

*Line 35:. . .substantial amounts of CH$_4$. (or something similar); Line 44:. . .in the form of gas hydrates,*

**Fixed (36, 47)**

*Line 79/80: Please re-phrase. Methane is not "reacting with sulfate". Obviously this is still debated but a term like "sulfate reduction coupled to the anaerobic oxidation of methane" or "sulfate reduction-coupled AOM" is more appropriate or rephrase to "microbes utilize methane. . ." or so.*

**Fixed (79)**

*Line 84: "Where CH$_4$ flux to. . .*

**Fixed (86)**

*Line 96-100: I suggest deleting these sentences. First, giving the total number of samples etc. is a little too much detail for the intro. Second, putting a "conclusion" sentence here, seems confusing (this is not the abstract).*

**Fixed (90-98)**

*Lines 150-157: Change all [] to ()*

**Fixed (136-141)**

*Line 152: Limited information on what?*

**Geologic, but this sentence has been deleted.**

*Line 193: I don't think the Schulz, 2000 reference is appropriate here. I suggest Boudreau (1997) and Iversen and Jørgensen (1993) instead.*

**Changed (166)**

*Line 241: Delete summary sentence; Line 314: Table 4?*

**Lines 241 and 314 deleted.**

*Line 273: Remove ; at end;*

**Fixed (268)**

*Line 338: Should be Table 3.*

**Fixed (305)**

*Line 340: ..dissolved sulfur and metal concentrations. . .*

**Fixed (307)**

*Line 342: HNO$_3$*

**Fixed (308)**

*Lines 353-381: Please shorten these sections. These are very common methods and you can reference the appropriate literature. We don't need to know exactly how much of which chemical you weight in etc.*

**This section (319-342) is shortened by 50 lines.**

*Line 389: Can you find a better title for this section than "Generalities"?*

**Yes "General observations" (352)**

*Line 390: Table 1?;*

**Fixed (353)**

*Line 405-412: Move to methods section.*

**Fixed (343-349)**

*Line 422-428, section 4.3: As outlined above, I suggest moving this to an supplementary material section.*

**This section has been moved to Supplementary Information.**

*Line 459: I am not sure a "decrease" can "change"*

**Changed to "is most pronounced" (373)**

*Line 477: Replace "faster" (time component) with " displayed a steeper decrease" or so;*

**Fixed (390)**

*Line 479: Replace "sulfate" with "sulfur"*

**Fixed (388)**

*Line 480: I don't see where the 0.98 comes from.*

**The ratio of change in alkalinity to sulfate.**

*Line 482 etc.: I suggest taking the "nutrient" term out. As you discuss, you are considering phosphate and ammonium as mineralization products.*

**Fixed (395)**

*Instead of the discussion in Lines 483-485, why not just say "..the mineralization products. . .."*

**This line deleted.**

*Line 621: Replace symbol.*

**This line deleted.**

*Line 633: I am not sure this is correct. A concave-down sulfate concentration profile usually implies on-going organoclastic sulfate reduction above the SMT. Otherwise you get a linear profile driven by diffusion of sulfate from the sediment-water interface to the SMT.*

**This line deleted.**

*Lines 635-637, 637-639, 639-641: These sentences need references.*

**These lines deleted.**

*Line 650 etc.: Do you actually calculate the methane fluxes somewhere? If so, how were methane fluxes calculated? What was taken into consideration? What if organoclastic sulfate reduction is occurring in close vicinity above the SMT, ie, your upward methane flux would then not be equal to the downward sulfate flux (at a 1:1 ratio). Where is the methane flux data?*

**We do not understand this comment. We infer little to no CH$_4$ in the sediments, therefore we cannot calculate CH$_4$ flux.**

*Line 671: "...imply a SO$_4$2- flux..";*

**This line deleted.**

*Line 674: 6.8 mol/m$^2$*

**This line deleted.**

*Line 687-688: Ok, it has a different ratio. . .and ? I am not sure you mention this here?*

**This line deleted.**

*Line 706: 43.54‰*

**Fixed (520)**

*Line 708-709: I don't think that this is an "issue" but as you point out earlier it is very common to only observe hydrogen sulfide very close to the SMT. Nonetheless, if "none was detected" what do you conclude from that (ie, here please insert a short discussion on the reaction of hydrogen sulfide with dissolved iron and iron oxides, pyrite formation etc)?*

**Fixed (523-525). We still use the term "issue," but elaborate on the reasons.**

*Line 724: manganese oxide reduction, iron oxide reduction; also denitrification and nitrate reduction ???;*

**Fixed (537-539)**

*Line 742: "The idea. . ." There is a word missing here.*

**Fixed (567)**

*Line 779: Can you find a better title than Explanations"?*

**How about "Possible explanations for widespread absence of gas hydrate and methane" (593)?**

**Response to Referee 1:**

**The authors thank Reviewer 1, and wish to make some general comments before discussing individual criticisms. Unfortunately, this review is difficult to read, and many comments are not germane to the original MS. In order to explain our overall response, we first summarize this review:**

**1. The MS does not contain additional data, even though it is one of the most comprehensive pore water chemistry data sets within a single effort generated to date in any region, let alone from a previously virtually unexplored section of the Arctic Ocean.**

**2. The pore water results cannot be used to understand $CH_4$ abundance, although numerous papers in multiple regions, including the Arctic offshore Alaska, convincingly demonstrate the opposite (eg. Borowski et al., 1996; Jørgensen et al., 2001; Torres and Kastner, 2009; Treude et al., 2014; etc.).**

**3. The primary interpretation and conclusion conflicts with previous speculations in the region, although no pertinent information to the problem exists beyond the current work.**

**Additionally, in certain parts, the reviewer appears to be discussing a different MS than ours. Page C4, for example, the reviewer criticizes the authors for "vigorous" referencing of a study (Nauhaus et al., 2002) that we neither cite nor discuss.**

**Specific Comments from Referee 1:**

*However, this ms rather demonstrates that the current state of knowledge of pore water biogeochemistry in particular areas of the ocean is very incomplete; a great deal of effort will be required in order to improve our understanding of the relationship between sulfur and carbon cycling in the Arctic.*

**We do not follow this comment. The community has published volumes of detailed research about pore water chemistry in regions that contain gas hydrate (Borowski et al., 2000; Torres et al., 2004; Treude et al., 2005; Dickens and Snyder, 2009; Coffin et al., 2013). This is stressed throughout the updated MS.**

*The authors of the ms come to the following conclusions: 1) Based on interpretation of the pore water profiles, they found no evidence for upwardly diffusing $CH_4$. 2) Based on these data, they strongly suggested that gas hydrates do not occur on the slopes of the ESM. 3) They claimed that previous investigators who suggested that hydrate deposits exist in the Arctic shelf/slope based on results of their investigations were simply wrong.*

**The first point is incorrect. The MS clearly states St. 31 has evidence of $CH_4$. The second point has been clarified in the revised MS. We strongly suggest WIDESPREAD gas hydrates do not occur as previously speculated. It is important to note our results and interpretations DO NOT CONFLICT with any previous data or direct results from the region. See lines 36, 72, 593-594, 603-606, 618-622, 629-631, and 671-676.**

*First of all, I do not understand why, when reporting low $CH_4$ concentrations and the relationship between CH4 and sulfate dynamics in the pore water, the authors did not measure the concentrations of either parameter. Is it not logical to measure $CH_4$ and sulfate in pore water if one is going to report "low methane concentrations in the sediments"? These are rather routine measurements.*

**Not all measurements could be generated given the limitations of the expedition and subsequent funding. However, we did measure the S concentrations of pore water, and know these are representative of $SO_4^{2-}$, because we checked for $H_2S$ as well as measured dissolved $Ba^{2+}$. Moreover, there are problems with generating quantitative dissolved $CH_4$ profiles in marine sediment because of degassing associated with changes in pressure and temperature.**

**Here, it is absolutely crucial to realize that, as stressed in the text and at least to our knowledge, no region with significant CH$_4$ in moderately shallow sediment (< 500 m) has high dissolved S/dissolved SO$_4^{2-}$ in pore waters near the seafloor, as well as other certain chemistry documented here. This is now stated in the updated MS, including discussion of localized areas of high advection and CH$_4$ venting (484-489 and 496-512).**

*The authors referred to other researchers in their ms to present supportive arguments, but none of these referenced studies avoided taking measurements.*

**In fact, a significant fraction of the research referenced was partially generated by one of the authors (Dickens). It should be noted that the measurements were not avoided in some means to hide information, but rather that we know how CH$_4$ exists and cycles in many regions, and the most prudent means of tackling the problem at a first-order level over an immense area from an ice-breaker is to generate numerous detailed pore water profiles (Borowski et al., 1999; Snyder et al., 2007; Hu et al., 2015).**

*In addition, the authors of this ms speculate about the particulate organic carbon (POC) and OC content of sediment, but did not measure either parameter. OC content of sediments should be reported as a number of different carbon stocks, not just POC.*

**This comment is unclear. As numerous studies have shown (Borowski et al., 1996; Dickens, 2000; Hensen et al., 2003; Geprags et al., 2016), the abundance of CH$_4$ in shallow sediment on continental slopes does not depend on the current supply of POC, but rather on the integrated input of POC over long time intervals (e.g., hundreds of thousands to million year time scales), which cannot be assessed without drilling. This last point has been added to the updated MS (144-145, 548-549, and 630).**

*Second, I do not understand why the presence or absence of CH$_4$, either in the sediments or in the water column in this area, should be necessarily connected to the existence or non-existence of hydrates. Are hydrates the only possible source of CH$_4$ in the Arctic shelf/slope? I believe not; hydrates could be only a tiny fraction of the source, because the hydrate stability zone (HSZ)*

*created by P/T conditions could compose only a small fraction of the sedimentary drape (a few hundred meters), while the sedimentary drape could be a few kilometers thick.*

**The authors do not understand this comment. Gas hydrates are not a source of $CH_4$, but rather, one phase of $CH_4$ in open and dynamic systems, where $CH_4$ carbon can exist as dissolved gas, free gas, and gas hydrate.**

**This project cannot link shallow water chemistry profiles to $CH_4$ abundance at truly deep depths, and the MS does not attempt to do so. However, deep $CH_4$ cannot exist as gas hydrates, for reasons of P/T conditions. On the other hand, the presence of gas hydrate in the upper few hundreds of meters of sediment is absolutely related to total $CH_4$ concentrations in pore space, which are linked to shallow sediment through diffusion or in some cases advection. This point is clarified in the updated discussion section (605-617).**

*Third, the purpose of this massive MS is not clear to me. This paper is flooded with equations and details devoted to methods, but mathematics, first of all should be applicable; then, the accuracy of mathematics does not aid in interpreting the inconclusive data.*

**We do not understand this comment. The purpose, as outlined in the MS, is very clear: we know how pore water chemistry profiles look above gas hydrate systems at numerous locations around the world, and we know how to interpret them at a basic level; we generated such profiles in the region of interest; the pore water profiles do not conform to those at any region where significant $CH_4$ occurs in shallow sediment nor our understanding as to why such profiles arise.**

**However, we agree the MS can be shortened significantly. The revised MS has moved essentially all of the Rhizon discussion to supplementary materials, deleted several sections, figures, and tables, (described above) and streamlined much of the text including the methods section.**

*Below are my comments on some aspects of this ms. A more detailed look would be as long as the ms itself, because nearly every page of this ms would benefit from clarification. The methodology chosen by the authors of this ms and their level of understanding of the processes they were trying to investigate are my greatest concern. Biogeochemists working in the marine ecosystems have already gained some understanding of the fact that biogeochemical processes associated with diagenetic transformation of organic matter under anaerobic conditions in marine sediments are very complex microbe-mediated processes. These processes involve microorganisms from various physiological groups: aerobic and anaerobic saprophylic and cellulose degrading bacteria, sulfate reducers, methanogens, denitrifiers, and methylotrophs. Transformation of organic matter is a multi-stage process: primary anaerobes decompose polymeric compounds to monomers, which, in turn, serve as a substrate for fermentation agents and gas-producing bacteria. A general conclusion is that the major fraction of OC preserved in the sediments is oxidized to $CO_2$ by the sulfate-reducing bacteria (SRB) and that 2 moles of OC are oxidized for every mole of sulfate reduced: $4H(CH_2)_n COO^- + (3n + 1)SO_4^{2-} + H_2O!(4n + 4)HCO_3^- + (3n + 1)HS^- + OH^- + nH^+$. When acetate is oxidized completely, the atomic ratio of OC oxidized to sulfate-S reduced is 2 : 1. However, as 'n' increases, the C: S ratio changes; the ratio between the reactants could be different because it depends on the varying nature of the organic matter (Lerman 1982). This is because most of the photosynthate is not immediately available for oxidation; only the low molecular weight (LMW) fraction of dissolved OC (DOC) is rapidly oxidized by SRB, while the high molecular weight (HMW) fraction of POC, which usually increases with depth, is refractory. There are severe restrictions on microbial activity other than substrate availability, including that SR as a biotic process may be more strongly coupled to mineralogy (Ivanov et al., 1989). The knowledge that has been accumulated by scientists so far is very limited and only applicable to those particular ecosystems which were investigated beyond the Arctic.*

**The authors are perplexed by this comment, as most of it does not pertain to our MS, and it seems to belie faulty logic. At a basic level the referee appears to think biogeochemical processes are so complex that the community cannot obtain overall net chemical reactions and flux of carbon from pore water chemistry.**

**If we are interpreting the comment correctly, we then return an obvious question: how and why can the community measure similar pore water chemistry profiles at myriad locations and see basic commonalities (e.g., the absence of SO$_4^{2-}$ above sites with the presence of significant CH$_4$ below), irrespective of the specifics and microbiology involved? No change has been made to the MS regarding this comment.**

*The most reliable method to trace the course of sulfate reduction in sediments uses radioactive sulfate (35S). By the use of this method it was shown that most reduced 35S-sulfate was in pyrite and organic sulfur (Lein et al., 1982). The relationship between sulfur and carbon cycling in the Arctic marine systems is even more complicated, because the relationships between the sites of primary production and the sites to which organic matter is translocated and deposited, including organic matter delivered to the shelf/slope from surrounding land, are difficult to establish both qualitatively and quantitatively.*

**We do not know how to respond to this comment, as it mostly does not pertain to our MS. It begins with a statement for which we disagree, and it does not make sense in its entirety. No change has been made to the MS regarding this comment.**

*A recently published review of CH$_4$ emissions from the seafloor in the Arctic Ocean underscored that the role of SRB in the anaerobic oxidation of methane (AOM) is unclear and the ecology of AOM communities, particularly for high-latitude environments, is not well understood. For that reason, predicting CH$_4$ fluxes, especially those related to hydrate dissociation, remains highly speculative (James et al., 2016). This is because CH4 is transported within the sediments in two different ways: as a dissolved phase (by diffusion or advection) or as free gas (ebullition). Free gas is inaccessible to microbes, which depend on a diffusive transmembrane gas transport. This means that release of free gas through the sediments might not leave any traces in the pore water (see Fig.5 in James et al., 2016).*

**The authors are unsure how to respond to this comment because it mostly is irrelevant to the topic of our MS. Crucially, we are not concerned how CH$_4$ would escape the seafloor**

via ENHANCED gas hydrate dissociation in the future, but whether significant $CH_4$ exists in shallow sediment on the SLOPE in the now. We hope this point is clear in the MS.

Additionally, the referenced MS (James et al., 2016) does not conflict with our interpretations. These authors clearly indicate that AOM is a dominant process above methane-charged systems at steady-state conditions, and should impact pore water $SO_4^{2-}$ gradients (e.g., the very Fig. 5 that the referee emphasizes).

The comment that passage of free gas through sediment does not leave traces in pore water is simply incorrect. The paper and figures by James et al. (2006) by no means suggest this concept, and rightfully so. Pore waters in areas where $CH_4$ advects from below at high rates, such as along faults and fractures (Fig. 5), have truly different chemistry than seawater and anything in our results. The updated Figure 12 emphasizes some of these differences.

*Moreover, recently published observational data show that in the Arctic environment, for example in the Alaskan Beaufort Sea continental margin sediments, substantial (30-500 µM) concentrations of sulfate can remain below the sulfate-methane transition zone (SMTZ) although mass balance cannot explain the source of sulfate below the SMTZ.*

This comment seems to betray a basic misunderstanding of $SO_4^{2-}$, $CH_4$ fluxes, and mass balance. $SO_4^{2-}$ concentrations 0.03-0.5 mM are not "substantial" compared to the ~28 mM in typical seawater. Second, there are at least three known reasons for $SO_4^{2-}$ below the SMT: 1. Pore water contamination, 2. $H_2S$ oxidation, and 3. Barite dissolution. Third, mass balance always applies.

Additionally, the Beaufort Sea results we assume the referee is mentioning (Coffin et al., 2013), show the exact type of shallow pore water profiles which this MS argues would occur above gas hydrates – and do not in our results! No changes have been made to the MS regarding this comment.

*In addition, sulfate reduction and anaerobic oxidation of CH$_4$ can occur throughout the methanogenic zone. Experimental data indicated decoupling of sulfate reduction and AOM and competition between sulfate reducers and methanogens for substrates, suggesting that the classical redox cascade of electron acceptor utilization based on Gibbs energy yields does not always hold even in diffusion-dominated systems (Treude et al., 2014).*

**Yes, this may true at a detailed and microscopic level, as pointed out by numerous authors, but not at any macroscopic level, at least that we are aware of, excepting odd environments (e.g., brines). The true beauty of pore water chemistry in the deep-sea marine environment is the remarkable consistency of multiple constituents linked to an array of environments. To restate from above, and in the text, all pore waters in methane-charged systems on continental slopes that we are aware of have certain commonalities – none that are seen in any of the pore waters generated in this study. This point has been emphasized in the updated discussion section (605-617).**

*Although they vigorously referred to Nauhaus et al. (2002) as a proxy-establishing experiment, the authors did not give this work any critical assessment. If they had done so, they would have definitely questioned the claim that methanotrophic communities associated with SRB oxidize CH$_4$ anaerobically in a 1:1 ratio to sulfate reduction. How that could be possible if the reported 4-5-fold increase in H2S production (accumulated over 80 days!) was accompanied by an increase in CH4 concentration of 3 orders of magnitude (from 0.01 to 15.8 mM)? Besides, rates of SR were so small (0.5-3.0 µM/d-1) compared to the concentrations of sulfate (103-1.55 µÐIJ) that the question arises: How could this little change be reliably measured (without using the 35S method, which they did not) and related to AOM?*

**This is perhaps the most confusing part of this review. We neither cite nor discuss the paper by Nauhaus et al. (2002). Indeed, the first author had not even read this work prior to submittal. We are forced to conjecture the referee is confusing our MS with another. This would explain a number of seemingly inexplicable comments which do not pertain to our text (ex. the referee statement that we "assumed CH$_4$ was being released from**

**destabilizing hydrates, most likely via bubbles and the convective flow of geofluids"). No change has been made to the MS regarding this comment.**

*Not to mention that this effect has no applicability to the Arctic Ocean.*

**We are not sure how to address. The referee seems to have a view that physical chemistry and biochemistry in the Arctic Ocean are somehow special, so that basics and inferences gained from elsewhere around the world do not apply.**

**Here, it is especially important to note the paper by Coffin et al. (2013), as already highlighted in our MS. These authors characterized pore water chemistry in short sediment cores above sequences with known gas hydrates along the shelf and slope of the Beaufort Sea (Arctic), very much as done in our MS. As predicted, they observed shallow SMTs indicative of a strong diffusive methane flux. As obvious in our work, their pore water profiles contrast with those from the slopes off northern Siberia. This point has been emphasized in the updated MS (88, 223, 508-509, 623, and 647-649)**

*Another concern is this: How representative of the area are these data? Only four short transects consisting of 16 stations are presented; each transect is based on data from 2-6 stations. Data from only 2-4 stations represent all core depths. Core lengths vary from 1.95 to 8.43 m (mean length 5.25 m). Eight of the 16 stations are only represented by the very uppermost layers (from 0.16 to 0.39 m) of sediment collected by the multi-corer. These shortest parts are the most valuable as they represent the least disturbed environment, but they are too short to constitute any sort of conclusive data regarding CH₄ cycling in the sediments. I can only guess at how the authors succeeded in dividing these tiny cores into numerous parts, each 0.2-0.3 m in length, and accumulated enough data to compare these cores with one of two idealistic schemes to characterize the specific dynamics of processes occurring over a sediment depth of 100 m (Fig.1). Data obtained by other types of sampling (piston/gravity coring) should be treated and interpreted very cautiously as the cores are not only severely disturbed during the coring process, but also chemically altered as they are extracted from the sea floor and lifted onto the ship.*

**This comment does not make sense. First, the fact that the pore water profiles give nice, detailed gradients in multiple species, demonstrably indicates that the cores have minimal disturbance. We can add core photos to our already long paper, if desired, to further emphasize this point.**

**Second, the proposition that the uppermost part of a core is the least disturbed and most important to understanding processes is flat out wrong. This is because of the nature of coring, which tends to disturb (or in many cases not recover) the top few cms, and because of bioturbation and reoxidation.**

**Third, the methodology for how the cores were sampled is detailed at length in the manuscript. Indeed, the reviewer criticized the authors earlier for the length of this section, and now claims to only "guess" at how this was accomplished. No changes have been made to the MS regarding this comment.**

*Finally, the authors plotted water concentration profiles along each transect collectively (!) using colors and symbol types which make it virtually impossible to distinguish between these symbols, making interpretation of the data sets very difficult.*

**We agree these figures are difficult to read. Given the extremely large dataset over this vast region, it is difficult to clearly present results concisely. It seemed to us somewhat overwhelming to plot the chemistry at every site independently, or alternatively every species analyzed at multiple sites independently. In order to improve readability, we have increased panel, symbol, and line widths while minimizing white space. The ACEX data has been removed to limit clutter, and the legends are inside the panels. Hopefully, this improves readability without lengthening the paper.**

*From this, it follows that the authors assumed complete uniformity of processes occurring not only in the observed settings located tens of kilometers apart from each other, but also over the entire slope area! This is despite the fact that CH4 fluxes on the East Siberian Arctic Shelf*

*(ESAS), which could be associated with CH4 releases from decaying hydrates, have been reported to vary by orders of magnitude within much smaller scales (Shakhova et al., 2015).*

**We did not assume complete uniformity of processes. In fact, the MS goes into great detail explaining the range of processes that relate to the pore water chemistry -- processes that have been well documented along many continental slopes. The authors see no reason to add to this already lengthy section.**

*I see a clear discrepancy between the basic assumptions made by the authors and the methodology used to test these assumptions. The authors assumed CH4 was being released from destabilizing hydrates, most likely via bubbles and the convective flow of geofluids.*

**We do not understand this comment. The referee is stating things that cannot be found anywhere in the text. Additionally, this comment has leapt well beyond anything discussed in the MS. As stated above, and clearly in the MS, we discuss the lack of evidence for CH$_4$ in shallow sediment. Our MS has little bearing on how gas hydrates would be destabilized and how CH$_4$ would be released. No change has been made to the MS regarding this comment.**

*Despite that, all equations used for estimates refer to the diffusive transport of CH$_4$ and other substances in the sediments. This is understandable; they used what was available. The problem is that the mathematics associated with diffusive transport cannot be used to describe the release of free gas from decaying hydrates. When assuming CH4 release from gas hydrates, one should realize that hydrates convert to free gas; the released gas travels upward much faster than diffusion occurs, through very efficient gas migration paths (chimneys etc.). In most cases, ascending CH4 can avoid oxidation in a few ways. 1) Because free gas resulting from hydrate decay is over pressured, it builds up a gas front; this disturbs sediment layering, creating the characteristic marks of gas release (pockmarks etc.). 2) Only CH$_4$ dissolved in pore water is reachable by microbial communities; CH$_4$ released as free gas (ebullition) is not consumable by microbes. 3) AOM rates are only remarkable as compared to rates of modern methanogenesis,*

*because all synergetic processes should be energetically efficient for all members of the microbial community, including SRB, methanogens and methanotrophs, etc.*

**This comment is simply wrong. In most locations with gas hydrate, the vast majority of CH₄ generated in the sediment ultimately (i.e., long time scales) escapes back to the ocean through diffusion. The assumption that methane-carbon above gas hydrates only returns to the ocean as free gas is entirely incorrect. The authors have added this discussion to the updated MS (605-617).**

*Finally, the authors of the ms used three assumptions to explain their findings. Their first assumption is that bottom seawater on the slope north of Siberia is warming, leading to hydrate destabilization. There are no reports of increased bottom water temperatures along the slope of the Arctic during either the last glacial cycle (Cronin et al., 2012) or the Holocene (Biastoch et al., 2011; Dmitrenko et al., 2011; James et al., 2016). All papers published so far project the response of the hydrate inventory to possible future climate change in the Arctic. The paper of Stranne et al., (2016) the authors refer to assumes a linear rise in ocean bottom water temperatures of 3C over the coming 100 years. This speculative warming of the Arctic is intentionally set higher than in other studies (<2C by Biastoch et al., 2011; <1C by Kretschmer et al., 2015) while modeling assumptions contradict the existing hydrological data (Biastoch et al., 2011; Dmitrenko et al., 2011; James et al., 2016).*

**This is another statement that absolutely cannot be found in our MS. Nowhere do we assume bottom seawater is warming, nor are warming temperatures even necessary to have CH₄ in pore waters above gas hydrates. Here, again, we wonder if the referee is thinking of another project. No change has been made to the MS regarding this comment.**

*Their second assumption is the quintessential statement that "Implicit of this finding is that sediments sequences along the ESM lack gas hydrates" following the authors' speculations about why predictions of hydrates on the ESM are so markedly wrong.*

**This statement has been reworded to better reflect the conclusion, as stated elsewhere in the MS, of WIDESPREAD gas hydrates. However, this is not an assumption, but rather a direct consequence of our results. The pore water chemistry profiles strongly indicate the lack of significant methane concentrations in the upper few hundred meters of sediment; given P/T conditions for gas hydrate, this absolutely implies an absence of gas hydrate.**

**Fascinatingly, prior to this expedition, we did expect widespread gas hydrates.**

*The authors then suggest that: 1) the significant sea-ice concentration on the ESM diminishes net primary production (NPP); 2) the extremely broad continental shelf prevents accumulation of terrestrial organic-rich sediments; and 3) sediment accumulation is highly variable, so organic matter can be consumed during intervals of low deposition. None of these explanations is true. It was recently shown that the total OC (TOC) content in the ESAS/ESM sediments measured along the transect spanning more than 800 km from the Lena River mouth to the shelf (2000–3000 m water depth) varied between 2 % at shallow water depths and 0.8% in deeper water (Bröder et al., 2016). In addition, TOC values and general patterns, which reflect fractions of terrigenous OC reaching the slope (based on biomarkers), were within the same range as those measured for the North American Arctic margin (Stein and Fahl, 2000, 2004; Goni et al., 2013). For comparison, an average value for the continental slope of the Gulf of Mexico, where large storage of CH4 hydrates has been proven to exist, is 0.8% ±0.2 (Gulf of Mexico Hydrate Research Consortium). Moreover, according to Arrigo and van Dijken (2011), the total annual NPP over the Arctic Ocean exhibited a statistically significant increase by 20% between 1998 and 2009, due mainly to increases in both the extent of open water (+27%) and the duration of the open water season (+45 days). Most importantly, increases in NPP over the 12 year study period were largest in the eastern Arctic Ocean, most notably in the Siberian (+135%) sector.*

**While interesting, none of this is relevant. This is because, for high $CH_4$ concentrations to exist in the upper few hundreds of meters on the slope, it is past carbon burial (i.e., not recent) that matters. This point has been clarified in the updated text (144-145, 548-549, and 630).**

*It is interesting that the authors themselves confirmed that: 1) environmental conditions on the ESM are highly conducive for gas hydrates; 2) hydrate occurrence in the other areas of the Arctic, where hydrates were predicted, was confirmed by hydrate recovery; and 3) all the models developed by generations of geologists to predict hydrates in the Arctic used the same assumptions.*

**We would agree with this statement, if logical qualifiers were added. The environmental conditions are highly conducive for gas hydrates IF AND ONLY IF THERE IS SUFFICENT METHANE; hydrate occurrence in the other areas of the Arctic, where hydrates were predicted, was confirmed by hydrate recovery AND BY PORE WATER CHEMISTRY IN SHALLOW CORES; all the models developed by generations of geologists to predict hydrates in the Arctic used the same assumptions WHICH ENTIRELY INFER A SOURCE OF CARBON TO PRODUCE CH$_4$.**

**The referee is ignoring two crucial facts, both discussed at length in the MS: (1) all previous works hinge on an assumption (not evidence) of significant CH$_4$ in shallow sediment; and (2) NO pertinent data to the problem exists beyond our current work. These point are clarified in the updated text.**

*If the authors agree that these statements are true, they failed to be critical of their own work, which is based on a handful of inconclusive data obtained on a single expedition, groundless methodology, and a few erroneous assumptions. Instead of casting doubt on the results of others, I would suggest that the authors question their own results and make a greater effort to accumulate clear, interpretable data. I believe I have made it quite clear that there is a huge discrepancy between the results presented by the authors and the far-reaching conclusions they are trying to support with these data. I see no way to support publication of this MS in its current state.*

**We respectfully disagree with the referee's conclusions.**